# When Does Closeness in Distribution Imply Representational Similarity? An Identifiability Perspective

**Beatrix M. G. Nielsen**[*,1]  **Emanuele Marconato**[2]

**Andrea Dittadi**[†,3,4,5]  **Luigi Gresele**[†,6]

[1]Technical University of Denmark  [2]University of Trento, Italy  [3]Helmholtz AI, Munich
[4]Technical University of Munich  [5]Munich Center for Machine Learning (MCML)  [6]University of Copenhagen

## Abstract

When and why representations learned by different deep neural networks are similar is an active research topic. We choose to address these questions from the perspective of identifiability theory, which suggests that a measure of representational similarity should be invariant to transformations that leave the model distribution unchanged. Focusing on a model family which includes several popular pre-training approaches, e.g., autoregressive language models, we explore when models which generate distributions that are close have similar representations. We prove that a small Kullback–Leibler divergence between the model distributions does not guarantee that the corresponding representations are similar. This has the important corollary that models with near-maximum data likelihood can still learn dissimilar representations—a phenomenon mirrored in our experiments with models trained on CIFAR-10. We then define a distributional distance for which closeness implies representational similarity, and in synthetic experiments, we find that wider networks learn distributions which are closer with respect to our distance and have more similar representations. Our results thus clarify the link between closeness in distribution and representational similarity.

## 1 Introduction

How to compare and relate the internal representations learned by different deep neural networks is a long-standing and active research topic [3, 30, 33, 38, 45, 49, 65]. Understanding when and why various kinds of *representational similarity* emerge has implications for model stitching [12, 16, 36, 41, 46], knowledge-distillation [53, 58, 73, 74] and interpretability for concept-based and neuro-symbolic models [5, 15, 42], see [61] for a review. A theory of **when similarity occurs** will chiefly depend on **how similarity is measured**. As argued by Bansal et al. [3], a similarity measure "should be invariant to operations that do not modify the 'quality' of the representation, but it is not always clear what these operations are". Indeed, this depends on how one defines the 'quality' of a representation, and previous works have debated the pros and cons of different choices [3, 14, 30, 31, 45, 49, 65, 67].

We choose to address these questions from the perspective of identifiability theory. In identifiability of probabilistic models, representations are called *equivalent* if they result in equal likelihoods, and

---

[*]Correspondence to: bmgi@dtu.dk or beat@itu.dk.
[†]Joint last authors.

39th Conference on Neural Information Processing Systems (NeurIPS 2025).

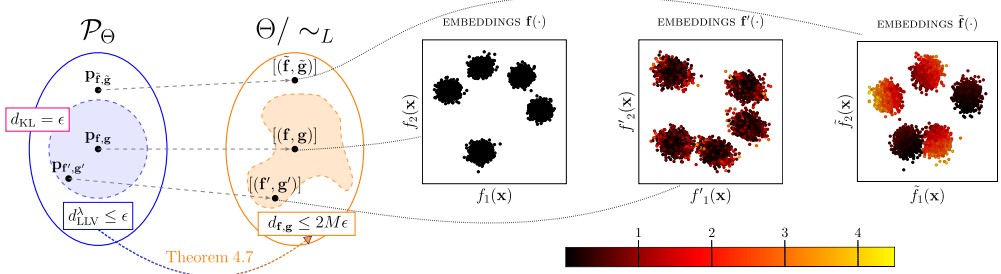

Figure 1: **When closeness in distribution does and does not imply representational similarity**. On the left, we show two distributions $p_{\mathbf{f},\mathbf{g}}, p_{\mathbf{f}',\mathbf{g}'} \in \mathcal{P}_\Theta$ which are closer than $\epsilon$ w.r.t. the distance $d_{\mathrm{LLV}}^\lambda$ (Definition 4.4), as illustrated by the shaded blue ball. We use $[(\mathbf{f},\mathbf{g})] \in \Theta/\sim_L$ to denote the *identifiability class* (Footnote 3) of a $\sim_L$-identifiable model with *embedding* $\mathbf{f}$ and *unembedding* $\mathbf{g}$ (Section 2). Theorem 4.7 implies that the identifiability classes $[(\mathbf{f},\mathbf{g})]$ and $[(\mathbf{f}',\mathbf{g}')]$, within the shaded orange area, will have *similar* representations—i.e., their dissimilarity under $d_{\mathbf{f},\mathbf{g}}$ (Definition 4.6) is bounded above by $2M\epsilon$. We also consider a third distribution $p_{\tilde{\mathbf{f}},\tilde{\mathbf{g}}} \in \mathcal{P}_\Theta$ which, while $\epsilon$-close in $d_{\mathrm{KL}}$ (in magenta), falls outside the blue region and has representations that are very *dissimilar* from those of $[(\mathbf{f},\mathbf{g})]$ and $[(\mathbf{f}',\mathbf{g}')]$, as described by our Theorem 3.1. On the right, we plot the three model embeddings: Taking $\mathbf{f}$ as reference, we find the best linear fit to $\mathbf{f}'$ and $\tilde{\mathbf{f}}$, and then color each of the points according to the residual error (brighter colors denote larger errors). The embeddings $\mathbf{f}'$ are nearly a linear transformation of $\mathbf{f}$, while $\tilde{\mathbf{f}}$ shows substantial deviation—visibly farther from being linearly related.

all models with equivalent representations together form an *identifiability class* [27].[3] Therefore, to preserve the 'quality' of representations, our notion of representational similarity needs to be invariant to precisely those transformations which leave the model likelihood unchanged. We focus on a model family including many popular pre-training approaches, e.g., autoregressive language models [8, 48], contrastive predictive coding [47], as well as standard supervised classifiers. Identifiability results for this model class show that, under a *diversity* condition, equal-likelihood models yield representations which are equal up to certain invertible linear transformations [28, 34, 51, 52].

As a first step toward a theory of representational similarity, we ask: **For what distributional and representational distances is it true that models whose output distributions are close have similar internal representations?** Addressing this requires going beyond classical identifiability, which assumes exact likelihood equality. We do so by proving in Theorem 3.1 that a small Kullback–Leibler (KL) divergence does not guarantee that the corresponding representations are similar—i.e., close to the identifiability class of our model family. As an important corollary, two models which are arbitrarily close to maximizing the data likelihood can still have dissimilar representations (Corollary 3.2). We then introduce a new distributional distance (Definition 4.4) and prove (Theorem 4.7) that closeness in this distance bounds representational dissimilarity (Definition 4.5).

We conduct classification experiments on CIFAR-10 and find substantial representational dissimilarity between some trained models with similarly good performance (Section 5.1). This dissimilarity appears to stem from a mechanism analogous to that identified in our KL divergence theory. We also run synthetic experiments showing that wider neural networks tend to learn distributions closer under our distance and have more similar representations (Section 5.2). This suggests that the inductive biases of larger models may promote representational similarity, something already observed in previous works.[4] Altogether, our findings indicate that the robustness of identifiability results [28, 51, 52] under the KL divergence may require additional assumptions, and that whether distributional closeness implies representational similarity depends critically on the choice of distributional and representational distance.

The **structure and main contributions** of our paper are as follows:

- We prove that small KL divergence between two models from our considered class (Equation (1)) does not guarantee similar representations (Section 3, Theorem 3.1); as a corollary, models with near-maximum data likelihood can have entirely dissimilar representations (Corollary 3.2).

---

[3] For a model family $\Theta$ and an equivalence relation $\sim_L$, $\Theta/\sim_L$ denotes the quotient space whose elements are the *identifiability classes* $[(\mathbf{f},\mathbf{g})] := \{(\mathbf{f}',\mathbf{g}') \in \Theta : (\mathbf{f},\mathbf{g}) \sim_L (\mathbf{f}',\mathbf{g}')\}$ [19, 27].

[4] E.g., Roeder et al. [52]: "as the representational capacity of the model and dataset size increases, learned representations indeed tend towards solutions that are equal up to only a linear transformation."

- We define a distance between probability distributions (Section 4.1) and a dissimilarity measure between representations (Section 4.2), and prove that small distributional distance implies representational similarity up to a specific class of invertible linear transformations (Section 4.3).
- We empirically show that: (1) models trained on CIFAR-10 can exhibit dissimilar representations despite similarly good performance through a mechanism similar to that in the proof of Corollary 3.2 (Section 5.1); (2) on synthetic data, wider networks yield lower mean and variance of our distributional distance, and also have more similar representations (Section 5.2).

## 2 Model Class and Identifiability

We consider input variables $\mathbf{x} \in \mathcal{X}$, which can be real-valued (e.g., images) or discrete (e.g., text strings). We assume that the input data is sampled *i.i.d.* from a distribution $p_{\mathcal{D}}(\mathbf{x})$. Given an input $\mathbf{x}$, we consider the task of defining a conditional distribution over categorical outputs $y \in \mathcal{Y}$, which can be class labels in classification tasks or next-tokens following a context in language modeling.

**Model class.** Following Roeder et al. [52], we consider a model to be a pair of functions $(\mathbf{f}, \mathbf{g}) \in \Theta$, where the codomain of both $\mathbf{f}$ and $\mathbf{g}$ is a vector space $\mathbb{R}^M$, which we will also refer to as the *representation space*. Here, $\Theta$ entails the space of all such pairs that can be generated by arbitrarily large neural networks.[5] We will refer to $\mathbf{f}(\mathbf{x})$ as the model *embedding* and to $\mathbf{g}(y)$ as the *unembedding*. The conditional distribution[6] defined by the model is given by

$$p_{\mathbf{f},\mathbf{g}}(y|\mathbf{x}) = \frac{\exp(\mathbf{f}(\mathbf{x})^\top \mathbf{g}(y))}{\sum_{y' \in \mathcal{Y}} \exp(\mathbf{f}(\mathbf{x})^\top \mathbf{g}(y'))} . \tag{1}$$

We will also indicate the model distribution with $p_{\mathbf{f},\mathbf{g}} \in \mathcal{P}_\Theta$, where $\mathcal{P}_\Theta$ is the space of conditional distributions that can be constructed as in Equation (1) using models in $\Theta$. This model family captures a variety of models for different learning contexts [52] among which: (i) auto-regressive language models like GPT-2 [48] and GPT-3 [8]; (ii) supervised classifiers like energy-based models [28] and models trained with DIET [26]; (iii) pretrained language models like BERT [13]; (iv) self-supervised pretraining for image classification with Contrastive Predictive Coding (CPC) [47].

**Identifiability.** It is useful to fix a pivot input point $\mathbf{x}_0 \in \mathcal{X}$ and a pivot label $y_0 \in \mathcal{Y}$ and consider the displaced embeddings $\mathbf{f}_0(\mathbf{x}) := \mathbf{f}(\mathbf{x}) - \mathbf{f}(\mathbf{x}_0)$ and unembeddings $\mathbf{g}_0(y) := \mathbf{g}(y) - \mathbf{g}(y_0)$. The conditional probability distribution generated by the model can then be rewritten as

$$p_{\mathbf{f},\mathbf{g}}(y|\mathbf{x}) \propto \exp(\mathbf{f}(\mathbf{x})^\top \mathbf{g}(y)) \exp(-\mathbf{f}(\mathbf{x})^\top \mathbf{g}(y_0)) \tag{2}$$

$$= \exp(\mathbf{f}(\mathbf{x})^\top \mathbf{g}_0(y)) . \tag{3}$$

Next, we review *identifiability* of the model class in Equation (1). Identifiability theory characterizes the set of transformations of representations that leave a model's output distribution unchanged; two models are therefore deemed equivalent when one can be obtained from the other by such a transformation. To proceed, we introduce an assumption about our considered models. This condition, also known as *diversity* [28, 34, 52], corresponds to only considering models that *span the whole representation space*, in the sense that there is no proper linear subspace of $\mathbb{R}^M$ containing the embeddings or the unembeddings [43]. Formally, this can be written as:

**Definition 2.1** (Diversity condition). *A model $(\mathbf{f}, \mathbf{g}) \in \Theta$ satisfies the diversity condition if there exists $\mathbf{x}_1, \ldots, \mathbf{x}_M \in \mathcal{X}$ and $y_1, ..., y_M \in \mathcal{Y}$ such that both the displaced embedding vectors $\{\mathbf{f}_0(\mathbf{x}_i)\}_{i=1}^M$ and the displaced unembedding vectors $\{\mathbf{g}_0(y_i)\}_{i=1}^M$ are linearly independent.*

The diversity condition can hold when the total number of unembeddings in $\mathcal{Y}$ is strictly higher than the representation dimension $M$ [52]. For example, in GPT-2 [48], while the representation dimensionality varies in the order of thousands (depending on the model size), the number of tokens in $\mathcal{Y}$ is of the order of ten thousands ($\sim 50k$ tokens), implying that diversity may be easily satisfied. With this assumption, we can prove the *identifiability* of the model in Equation (1) up to invertible linear transformations (a detailed proof can be found in Appendix B):

---

[5]Both $\mathbf{f}$ and $\mathbf{g}$ are therefore treated as nonparametric functions in the following, similar to [28, 34, 43, 52].

[6]We use the term "distribution" generically. In Equation (1), $p_{\mathbf{f},\mathbf{g}}(y|\mathbf{x})$ is a softmax over a finite label set, i.e., a probability *mass* function.

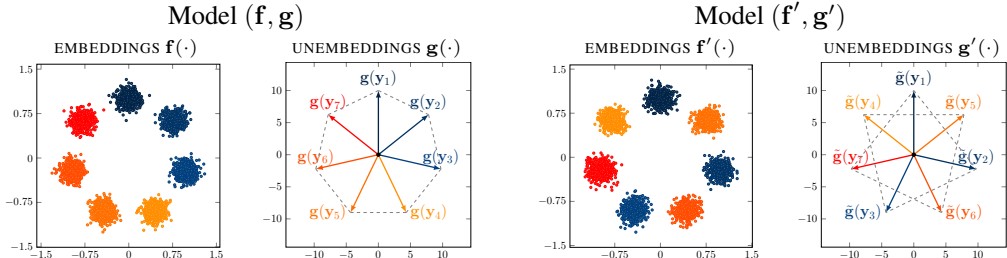

Figure 2: **Two models with small KL divergence but highly dissimilar representations.** We construct two models $(\mathbf{f}, \mathbf{g}), (\mathbf{f}', \mathbf{g}') \in \Theta$ whose representations are related by a non-linear transformation: The embeddings and unembeddings of the $(\mathbf{f}', \mathbf{g}')$ model are constructed by permuting the embedding clusters and the corresponding unembedding vectors of the $(\mathbf{f}, \mathbf{g})$ model. As a result, the nearest unembedding vectors in $\mathbf{g}(\cdot)$ (dashed lines) are mapped away from each other in $\mathbf{g}'(\cdot)$. In Theorem 3.1, we show that as the norm of the unembedding vectors $\rho$ grows for both models, their distributions $p_{\mathbf{f}, \mathbf{g}}$ and $p_{\mathbf{f}', \mathbf{g}'}$ become closer in KL divergence, whereas their representations remain dissimilar—i.e., far from being equal up to a linear transformation, see also Table 1.

**Theorem 2.2** (Linear Identifiability [28, 34, 52]). *Let $(\mathbf{f}, \mathbf{g}), (\mathbf{f}', \mathbf{g}') \in \Theta$, and $(\mathbf{f}, \mathbf{g})$ satisfy the diversity condition (Definition 2.1). Let $\mathbf{L}$ (resp. $\mathbf{L}'$) be the matrix with columns $\mathbf{g}_0(y)$ (resp. $\mathbf{g}_0'(y)$), and let $\mathbf{A} := \mathbf{L}^{-\top} \mathbf{L}'^{\top} \in \mathbb{R}^{M \times M}$. Then:*

$$p_{\mathbf{f}, \mathbf{g}}(y \mid \mathbf{x}) = p_{\mathbf{f}', \mathbf{g}'}(y \mid \mathbf{x}), \ \forall (\mathbf{x}, y) \in \mathcal{X} \times \mathcal{Y} \implies (\mathbf{f}, \mathbf{g}) \sim_L (\mathbf{f}', \mathbf{g}') \tag{4}$$

*where the equivalence relation $\sim_L$ is defined by*

$$(\mathbf{f}, \mathbf{g}) \sim_L (\mathbf{f}', \mathbf{g}') \iff \begin{cases} \mathbf{f}(\mathbf{x}) = \mathbf{A}\mathbf{f}'(\mathbf{x}) \\ \mathbf{g}_0(y) = \mathbf{A}^{-\top}\mathbf{g}_0'(y) \end{cases}. \tag{5}$$

The result of Theorem 2.2 is central to the scope of our analysis because it shows that representations of two models generating the same conditional distribution are related by an invertible linear transformation. This also means that, under the diversity condition, to each probability distribution $p_{\mathbf{f}, \mathbf{g}} \in \mathcal{P}_\Theta$ corresponds a unique (up to equivalence) choice of embeddings and unembeddings. See Fig. 1 for an illustration.

**Representational similarity.** Theorem 2.2 suggests that a natural notion of similarity between representations for our model class should account for the linear equivalence relation in Equation (5). Hereafter, we say two models $(\mathbf{f}, \mathbf{g}), (\mathbf{f}', \mathbf{g}') \in \Theta$ have *equivalent representations* if their embeddings and unembeddings are related by an invertible linear transformation of the kind in Theorem 2.2, i.e., $(\mathbf{f}, \mathbf{g}) \sim_L (\mathbf{f}', \mathbf{g}')$, and we call representations *similar* if they are 'close' to having such a relation—which we will make precise in Section 4.2. We note that several measures of representational similarity [30, 45, 49] are based on Canonical Correlation Analysis (CCA) [22], and are thus invariant to any invertible linear transformation. They thus attain their maximum when applied to representations that are equivalent in the sense above, but also for other invertible linear transformations. We discuss alternative choices of similarity measures in Section 6.

## 3 Models Close in KL Divergence Can Have Dissimilar Representations

Given Theorem 2.2, we next ask whether a similar conclusion holds approximately when the models have *nearly* the same distributions. More precisely, we ask: For a given choice of divergence between distributions, does a small difference in distributions imply representational similarity?

As a natural starting point, we consider the widely used Kullback–Leibler (KL) divergence. Specifically, we quantify the difference between two models by the KL divergence between their conditional distributions, averaged over the data distribution $p_\mathcal{D}(\mathbf{x})$. This is equivalent to the KL divergence between the corresponding joint distributions, assuming both share the marginal $p_\mathcal{D}(\mathbf{x})$. Formally:

$$d_{\mathrm{KL}}(p, p') := \mathbb{E}_{\mathbf{x} \sim p_\mathcal{D}} [D_{\mathrm{KL}}(p(y|\mathbf{x}) \| p'(y|\mathbf{x}))] = D_{\mathrm{KL}}(p(\mathbf{x}, y) \| p'(\mathbf{x}, y)). \tag{6}$$

When $D_{\mathrm{KL}}(p(y|\mathbf{x}) \| p'(y|\mathbf{x})) = 0$ for all $\mathbf{x}$, we obtain models that are $\sim_L$-equivalent, as specified in Theorem 2.2. However, only making the KL divergence small is insufficient to conclude much about the similarity of representations. The following result formalizes this claim (proof in Appendix C.1):

**Theorem 3.1** (Informal). *Let $k = |\mathcal{Y}|$ and assume $k \geq M + 1$. Then $\forall \epsilon > 0$, there exist models $(\mathbf{f}, \mathbf{g}), (\mathbf{f}', \mathbf{g}') \in \Theta$ for which $d_{\mathrm{KL}}(p_{\mathbf{f}, \mathbf{g}}, p_{\mathbf{f}', \mathbf{g}'}) \leq \epsilon$, and whose embeddings are far from being linearly equivalent.[7] More precisely, one can construct a sequence of model pairs, depending smoothly on a real-valued parameter $\rho$, such that, as $\rho \to \infty$, the KL divergence between the models tends to zero while their embeddings remain fixed and not equal up to any linear transformation.*

*Proof sketch.* We here sketch the construction used for $k \geq M + 2$ (see Appendix C.1 for a construction which holds with $k = M + 1$). Let $(\mathbf{f}, \mathbf{g}) \in \Theta$ be a model which has unembeddings with fixed norm, i.e., $\|\mathbf{g}(y)\| = \rho$, for all $y \in \mathcal{Y}$ and $\rho \in \mathbb{R}^+$. Assume the unembeddings have non-zero angles between them and let the unembeddings satisfy diversity (Definition 2.1). For $\mathbf{x} \in \mathcal{X}$, let the embedding $\mathbf{f}(\mathbf{x})$ have strictly largest cosine similarity with one $\mathbf{g}(\hat{y})$, for some $\hat{y} \in \mathcal{Y}$, and such that $\hat{y} = \operatorname{argmax}_{y \in \mathcal{Y}}(p_{\mathbf{f}, \mathbf{g}}(y|\mathbf{x}))$. As a consequence, note that the model $(\mathbf{f}, \mathbf{g})$ also satisfies the diversity condition for the embeddings. Let $(\mathbf{f}', \mathbf{g}') \in \Theta$ be another model which is constructed starting from $(\mathbf{f}, \mathbf{g})$ and considering a permutation, $\pi$, of the label indices such that $\mathbf{g}'(y_i) = \mathbf{g}(y_{\pi(i)})$ for $i = 1, \ldots, k$, with a corresponding shift of the embedding clusters.[8] For every $\mathbf{x} \in \mathcal{X}$, we let $\|\mathbf{f}'(\mathbf{x})\| = \|\mathbf{f}(\mathbf{x})\|$ and let the angle between $\mathbf{f}'(\mathbf{x})$ and $\mathbf{g}'(\hat{y})$ be equal to that between $\mathbf{f}(\mathbf{x})$ and $\mathbf{g}(\hat{y})$. We illustrate this construction in Fig. 2 for $\mathbf{f}(\mathbf{x}), \mathbf{g}(y) \in \mathbb{R}^2$. One can then find a permutation $\pi$ for which $\mathbf{f}(\mathbf{x})$ is not a linear transformation[9] of $\mathbf{f}'(\mathbf{x})$ (thus the KL divergence between $p_{\mathbf{f}, \mathbf{g}}$ and $p_{\mathbf{f}', \mathbf{g}'}$ is non-zero). However, as $\rho \to \infty$, we have that $d_{\mathrm{KL}}(p_{\mathbf{f}, \mathbf{g}}, p_{\mathbf{f}', \mathbf{g}'}) \to 0$, although the embeddings stay constant and not linearly equivalent. $\square$

Theorem 3.1 shows that, even when $d_{\mathrm{KL}}(p_{\mathbf{f}, \mathbf{g}}, p_{\mathbf{f}', \mathbf{g}'}) \leq \epsilon$, there is no guarantee that the models are close to being $\sim_L$-equivalent. The proof constructs a $\rho$-parameterized sequence in which the embeddings remain fixed and not equal up to a linear transformation; consequently, the error of the best linear regression of one embedding set onto the other is strictly positive and stays constant as $\rho \to \infty$, while $d_{\mathrm{KL}} \to 0$.

In Theorem 3.1, we consider the KL divergence between two distributions from our model family. However, when practitioners study representational similarity in trained models, they fit each model to the data distribution—usually by maximum-likelihood estimation—and then compare their representations. We therefore also formulate the following corollary which is closer to the usual setup:

**Corollary 3.2** (Informal). *Let $k = |\mathcal{Y}|$ and assume $k \geq M + 1$. Consider a dataset of $(\mathbf{x}, y)$ pairs, where only one label is assigned to each unique input. Then, we can find two models that obtain an arbitrarily small cross-entropy loss on the data while having representations which are far from being linearly equivalent in the same sense as in Theorem 3.1.*

The proof is in Appendix C.2, and an illustration through synthetic experiments can be found in Appendix F.4. This corollary shows that two models having close to optimal classification loss on training data, possibly also equal, may not possess similar representations at all. Importantly, we find that this situation can arise in practice with discriminative models trained on CIFAR-10 (Section 5), which can have dissimilar representations despite assigning high probability to the same labels.

## 4 When Closeness in Distribution Implies Representational Similarity

In this section, we introduce a distance between probability distributions (Section 4.1) and measure of representational similarity (Section 4.2) and prove that, for our model family (Equation (1)), small values of the former imply high similarity under the latter (Section 4.3).

---

[7]We show in Section 5.3 how the construction from our proof can lead to high dissimilarity both according to $m_{\mathrm{CCA}}$ [49, 52] and under the measure introduced in Definition 4.5.

[8]This is not a permutation of the representation dimensions—i.e., it is not $\mathbf{f}'(\mathbf{x}) = \mathbf{P}\mathbf{f}(\mathbf{x})$ for some permutation matrix $\mathbf{P}$. The permutation of label indices, and corresponding shift of embedding clusters, is in general not a linear transformation of the representations (see Fig. 2); it also changes the model's output distribution, since it alters the rank—by predicted probability—of every label except the highest-probability one for each input $\mathbf{x}$.

[9]In Table 1, we show that one can construct models this way and such that the mean canonical correlation is close to zero, whereas it would be equal to one if their representations were linear transformations of each other.

## 4.1 A Distance Between Probability Distributions

We start by studying what relation holds between models in $\Theta$ which need not have the same probability distributions. Hereafter, we will also consider the variance over input samples and outputs and denote it with $\mathrm{Var}_{\mathbf{x}}[\cdot] := \mathrm{Var}_{\mathbf{x} \sim p_{\mathcal{D}}}[\cdot]$ and $\mathrm{Var}_y[\cdot] := \mathrm{Var}_{y \sim \mathrm{Unif}(\mathcal{Y})}[\cdot]$, respectively. With this in mind, it is useful to introduce the following functions:

$$\psi_{\mathbf{x}}(y_i; p) := \sqrt{\mathrm{Var}_{\mathbf{x}}[\log p(y_i|\mathbf{x}) - \log p(y_0|\mathbf{x})]} \ \text{ and } \ \psi_y(\mathbf{x}_j; p) := \sqrt{\mathrm{Var}_y[\log p(y|\mathbf{x}_j) - \log p(y|\mathbf{x}_0)]} \ . \tag{7}$$

We also denote with $\mathbf{S} \in \mathbb{R}^{M \times M}$ the diagonal matrix with entries $S_{ii} := \psi_{\mathbf{x}}(y_i; p)^{-1}$. We now have everything we need to state a Lemma relating any two model embeddings (the full statement and proof, including the relation between the unembeddings can be found in Appendix E.1):

**Lemma 4.1.** *Let $(\mathbf{f}, \mathbf{g}), (\mathbf{f}', \mathbf{g}') \in \Theta$ be models satisfying the diversity condition (Definition 2.1). Let $\mathbf{L}, \mathbf{L}'$ be defined as in Theorem 2.2. Let $\tilde{\mathbf{A}} := \mathbf{L}^{-\top}\mathbf{S}^{-1}\mathbf{S}'\mathbf{L}'^{\top}$ and $\mathbf{h}_{\mathbf{f}}(\mathbf{x}) := \mathbf{L}^{-\top}\mathbf{S}^{-1}\boldsymbol{\epsilon}_y(\mathbf{x})$, where the $i$-th entry of $\boldsymbol{\epsilon}_y(\mathbf{x})$ is given by*

$$\epsilon_{y_i}(\mathbf{x}) = \frac{\log p_{\mathbf{f},\mathbf{g}}(y_i|\mathbf{x}) - \log p_{\mathbf{f},\mathbf{g}}(y_0|\mathbf{x})}{\psi_{\mathbf{x}}(y_i; p_{\mathbf{f},\mathbf{g}})} - \frac{\log p_{\mathbf{f}',\mathbf{g}'}(y_i|\mathbf{x}) - \log p_{\mathbf{f}',\mathbf{g}'}(y_0|\mathbf{x})}{\psi_{\mathbf{x}}(y_i; p_{\mathbf{f}',\mathbf{g}'})} \ . \tag{8}$$

*Then, we have*

$$\mathbf{f}(\mathbf{x}) = \tilde{\mathbf{A}}\mathbf{f}'(\mathbf{x}) + \mathbf{h}_{\mathbf{f}}(\mathbf{x}) \ . \tag{9}$$

**Remark 4.2.** *In the special case where the two models $(\mathbf{f}, \mathbf{g}), (\mathbf{f}', \mathbf{g}') \in \Theta$ have the same distribution, the error term in Equation (8) vanishes and we recover $(\mathbf{f}, \mathbf{g}) \sim_L (\mathbf{f}', \mathbf{g}')$ (see Corollary E.1).*

We note that the error term in Lemma 4.1 depends entirely on differences of log probabilities. We also see that if $\epsilon_{y_i}(\mathbf{x})$ is a constant (or equivalently, $\mathrm{Var}[\epsilon_{y_i}(\mathbf{x})] = 0$), the embeddings will be invertible linear transformations of each other. This suggests that if we want a distance between distributions which is related to the similarity of the embeddings, we can measure how far this error is from being a constant. We therefore define a distance which includes these weighted differences of log-likelihoods.

To proceed, we restrict our analysis to a subset of distributions that satisfy the following assumption:

**Assumption 4.3.** *We consider probability distributions $p$ such that for sets $\mathcal{X}_{\mathrm{LLV}} \subset \mathcal{X}$ and $\mathcal{Y}_{\mathrm{LLV}} \subset \mathcal{Y}$ containing all labels except one, the following conditions are satisfied: (1) For all $\mathbf{x} \in \mathcal{X}_{\mathrm{LLV}}$ we have that $\log p(y|\mathbf{x}) - \log p(y|\mathbf{x}_0)$ is not constant in $y$ and $p_{\mathcal{D}}(\mathbf{x}) > 0$; and (2) For all $y \in \mathcal{Y}_{\mathrm{LLV}}$ we have that $\log p(y|\mathbf{x}) - \log p(y_0|\mathbf{x})$ is not constant in $\mathbf{x}$.*

Note that this assumption guarantees that, for any such $p$, the terms $\psi_{\mathbf{x}}(y_i; p)$, and $\psi_y(\mathbf{x}_j; p)$ are non-zero for $\mathbf{x}_j \in \mathcal{X}_{\mathrm{LLV}}$ and $y_i \in \mathcal{Y}_{\mathrm{LLV}}$. Under Assumption 4.3, we exclude those probability distributions that assign the same distributions over the labels for all inputs, that is, we need at least one input to result in a different distribution. We also exclude those distributions which assign equal probability to two or more labels for all inputs.

**Definition 4.4** (Log-likelihood variance distance between distributions). *For any two probability distributions $p, p'$ for which there exists a common choice of $\mathcal{X}_{\mathrm{LLV}} \subset \mathcal{X}$ and $\mathcal{Y}_{\mathrm{LLV}} \subset \mathcal{Y}$ such that they both satisfy Assumption 4.3, for a fixed $\lambda \in \mathbb{R}^+$, and by considering the following terms:*

$$t_1 := \max_{y \in \mathcal{Y}_{\mathrm{LLV}} \backslash \{y_0\}} \left\{ \sqrt{\mathrm{Var}_{\mathbf{x}}\left[\frac{\log p(y|\mathbf{x})}{\psi_{\mathbf{x}}(y; p)} - \frac{\log p'(y|\mathbf{x})}{\psi_{\mathbf{x}}(y; p')}\right]}, \sqrt{\mathrm{Var}_{\mathbf{x}}\left[\frac{\log p(y_0|\mathbf{x})}{\psi_{\mathbf{x}}(y; p)} - \frac{\log p'(y_0|\mathbf{x})}{\psi_{\mathbf{x}}(y; p')}\right]} \right\}$$

$$t_2 := \max_{\mathbf{x} \in \mathcal{X}_{\mathrm{LLV}} \backslash \{\mathbf{x}_0\}} \left\{ \sqrt{\mathrm{Var}_y\left[\frac{\log p(y|\mathbf{x})}{\psi_y(\mathbf{x}; p)} - \frac{\log p'(y|\mathbf{x})}{\psi_y(\mathbf{x}; p')}\right]}, \sqrt{\mathrm{Var}_y\left[\frac{\log p(y|\mathbf{x}_0)}{\psi_y(\mathbf{x}; p)} - \frac{\log p'(y|\mathbf{x}_0)}{\psi_y(\mathbf{x}; p')}\right]} \right\}$$

$$t_3 := \max_{y \in \mathcal{Y}_{\mathrm{LLV}} \backslash \{y_0\}} |\psi_{\mathbf{x}}(y; p) - \psi_{\mathbf{x}}(y; p')|, \quad t_4 := \max_{\mathbf{x} \in \mathcal{X}_{\mathrm{LLV}} \backslash \{\mathbf{x}_0\}} |\psi_y(\mathbf{x}; p) - \psi_y(\mathbf{x}; p')| \ ,$$

*the log-likelihood variance (LLV) distance between $p$ and $p'$ is given by*

$$d_{\mathrm{LLV}}^{\lambda}(p, p') := \max \{t_1, t_2, \lambda t_3, \lambda t_4\} \ . \tag{10}$$

In Appendix E.2, we show that $d_{\mathrm{LLV}}^{\lambda}$ is a distance metric between sets of conditional probability distributions.[10] The non-zero weighting constant $\lambda$ is introduced because, as we will show in

---

[10] Specifically, $d_{\mathrm{LLV}}^{\lambda}(p, p')$ is non-negative; zero iff $p = p'$, symmetric; and it satisfies the triangle inequality.

Theorem 4.7, $t_1$ and $t_2$ can be used alone to bound the similarity between model representations, but $t_3$ and $t_4$ need also to be considered to make $d_{\mathrm{LLV}}^\lambda$ a distance metric. In our experiments, we set $\lambda$ to a small non-zero value, so that the $t_1$ and $t_2$ terms dominate (see Appendix E.3 for details).

Note that the log-likelihood variance distance, $d_{\mathrm{LLV}}^\lambda$, requires a choice of input, $\mathcal{X}_{\mathrm{LLV}}$, and label, $\mathcal{Y}_{\mathrm{LLV}}$, sets, and the exact value of the distance depends on this choice when the distributions are not equal. However, $d_{\mathrm{LLV}}^\lambda$ is a metric for any choice satisfying Assumption 4.3. Therefore in the experiments, we draw a sample of possible sets and choose the ones which give the smallest value.

## 4.2 A Dissimilarity Measure Between Representations

Having defined a distance between distributions, we turn to a distance between representations which is related to invertible linear transformations. We define a dissimilarity measure based on the partial least squares (PLS) approach called PLS-SVD [54, 60]. For two random vectors $\mathbf{z}, \mathbf{w}$, let $\mathbf{\Sigma_{zw}}$ denote the covariance matrix whose $(i, j)$-th entry is

$$(\mathbf{\Sigma_{zw}})_{ij} = \mathrm{Cov_{zw}}[z_i, w_j] := \mathbb{E}_{(\mathbf{z}, \mathbf{w}) \sim p_{\mathbf{z}, \mathbf{w}}} \left[ (z_i - \mathbb{E}_{\mathbf{z} \sim p_{\mathbf{z}}}[z_i])(w_j - \mathbb{E}_{\mathbf{w} \sim p_{\mathbf{w}}}[w_j]) \right], \quad (11)$$

where $p_{\mathbf{z}, \mathbf{w}}$ is the joint distribution of $\mathbf{z}, \mathbf{w}$, and $p_{\mathbf{z}}, p_{\mathbf{w}}$ are the respective marginals. PLS-SVD seeks pairs of unit-length vectors $\mathbf{u}_\ell \in \mathbb{R}^{d_Z}$ and $\mathbf{v}_\ell \in \mathbb{R}^{d_W}$ ($\ell = 1, \ldots, r$) that maximize the covariance between the one-dimensional projections $\mathbf{u}_\ell^\top \mathbf{z}$ and $\mathbf{v}_\ell^\top \mathbf{w}$, subject to mutual orthogonality of successive directions. This is equivalent to finding the left and right singular vectors of $\mathbf{\Sigma_{zw}}$ (more about PLS-SVD in Appendix D). Leveraging this procedure by PLS-SVD, we introduce a similarity and a dissimilarity measure:

**Definition 4.5** (PLS SVD distance between vectors). *Let $\mathbf{z}, \mathbf{w}$ be two $M$-dimensional random vectors, and define $\mathbf{z}', \mathbf{w}'$ by standardizing their components: $z_i' = (z_i - \mathbb{E}[z_i])/\mathrm{std}(z_i)$, $w_i' = (w_i - \mathbb{E}[w_i])/\mathrm{std}(w_i)$. Let $\{\mathbf{u}_i\}_{i=1}^M$ and $\{\mathbf{v}_i\}_{i=1}^M$ be the left and right singular vectors of the cross-covariance matrix $\mathbf{\Sigma_{z'w'}}$. We define*

$$m_{\mathrm{SVD}}(\mathbf{z}, \mathbf{w}) := \frac{1}{M} \sum_{i=1}^M \mathrm{Cov_{z'w'}}[\mathbf{u}_i^\top \mathbf{z}', \mathbf{v}_i^\top \mathbf{w}'], \quad d_{\mathrm{SVD}}(\mathbf{z}, \mathbf{w}) := 1 - m_{\mathrm{SVD}}(\mathbf{z}, \mathbf{w}). \quad (12)$$

Since $m_{\mathrm{SVD}}(\mathbf{z}, \mathbf{w}) \leq m_{\mathrm{SVD}}(\mathbf{z}, \mathbf{z}) = 1$, it follows that $d_{\mathrm{SVD}}(\mathbf{z}, \mathbf{w}) \geq 0$ for all random vectors $\mathbf{z}, \mathbf{w}$. We also have that the dissimilarity measure is invariant to rotation followed by dimension-wise scaling (see Appendix E.4): if $m_{\mathrm{SVD}}(\mathbf{z}, \mathbf{w}) = 1$ or, equivalently, $d_{\mathrm{SVD}}(\mathbf{z}, \mathbf{w}) = 0$, then there exist an orthonormal matrix $\mathbf{R}$ and diagonal matrices $\mathbf{S}, \mathbf{S}'$ scaling $\mathbf{z}, \mathbf{w}$ to unit variance such that $\mathbf{z} = \mathbf{S}^{-1} \mathbf{R} \mathbf{S}' \mathbf{w}$.

With this distance between vectors, we can define a dissimilarity between representations for models from our model family by seeing the embeddings and unembeddings as random vectors.

**Definition 4.6** (Representational dissimilarity measure $d_{\mathbf{f}, \mathbf{g}}$). *Let $(\mathbf{f}, \mathbf{g}), (\mathbf{f}', \mathbf{g}') \in \Theta$. Let $\mathbf{L}, \mathbf{L}'$ be defined as in Theorem 2.2. Let $\mathbf{N}$ (resp. $\mathbf{N}'$) be the matrix with columns $\mathbf{f}_0(\mathbf{x})$ (resp. $\mathbf{f}_0'(\mathbf{x})$). The representational dissimilarity measure $d_{\mathbf{f}, \mathbf{g}}$ is defined as follows:*

$$d_{\mathbf{f}, \mathbf{g}}((\mathbf{f}, \mathbf{g}), (\mathbf{f}', \mathbf{g}')) := \max \left\{ d_{\mathrm{SVD}}(\mathbf{L}^\top \mathbf{f}(\mathbf{x}), \mathbf{L}'^\top \mathbf{f}'(\mathbf{x})), d_{\mathrm{SVD}}(\mathbf{N}^\top \mathbf{g}(y), \mathbf{N}'^\top \mathbf{g}'(y)) \right\}. \quad (13)$$

## 4.3 Bounding Representation Dissimilarity via Distributional Distance

We now prove that it is possible to relate a bound on the distance metric $d_{\mathrm{LLV}}^\lambda$ between model distributions (Definition 4.4) to a bound on the dissimilarity measure $d_{\mathbf{f}, \mathbf{g}}$ between model representations (Definition 4.6). For the result below, We denote with $\mathbf{z}_1 := \mathbf{L}^\top \mathbf{f}(\mathbf{x})$, $\mathbf{z}_2 := \mathbf{L}'^\top \mathbf{f}'(\mathbf{x})$, $\mathbf{w}_1 := \mathbf{N}^\top \mathbf{g}(y)$ and $\mathbf{w}_2 := \mathbf{N}'^\top \mathbf{g}'(y)$ (proof in Appendix E.6).

**Theorem 4.7.** *Let $(\mathbf{f}, \mathbf{g}), (\mathbf{f}', \mathbf{g}') \in \Theta$ be two models such that: (1) There exist $\mathcal{X}_{\mathrm{LLV}} \subset \mathcal{X}$ and $\mathcal{Y}_{\mathrm{LLV}} \subset \mathcal{Y}$, consisting of a pivot point and all labels but one, such that all $\mathbf{L}, \mathbf{L}'$ and $\mathbf{N}, \mathbf{N}'$ matrices constructed from these sets are invertible;[11] (2) Both $p_{\mathbf{f}, \mathbf{g}}$ and $p_{\mathbf{f}', \mathbf{g}'}$ satisfy Assumption 4.3 for $\mathcal{X}_{\mathrm{LLV}}$ and $\mathcal{Y}_{\mathrm{LLV}}$; (3) The covariance matrices $\mathbf{\Sigma}_{\mathbf{z}_1 \mathbf{z}_1}, \mathbf{\Sigma}_{\mathbf{w}_1 \mathbf{w}_1}$ and the cross-covariance matrices $\mathbf{\Sigma}_{\mathbf{z}_1 \mathbf{z}_2}, \mathbf{\Sigma}_{\mathbf{w}_1 \mathbf{w}_2}$ are non-singular. Then, for any weighting constant $\lambda \in \mathbb{R}^+$, we have*

$$d_{\mathrm{LLV}}^\lambda(p_{\mathbf{f}, \mathbf{g}}, p_{\mathbf{f}', \mathbf{g}'}) \leq \epsilon \implies d_{\mathbf{f}, \mathbf{g}}((\mathbf{f}, \mathbf{g}), (\mathbf{f}', \mathbf{g}')) \leq 2M\epsilon. \quad (14)$$

---

[11]This is slightly stronger than the diversity condition (Definition 2.1) in the sense that we need diversity to hold for all sets using labels from $\mathcal{Y}_{\mathrm{LLV}}$ and the chosen pivot point.

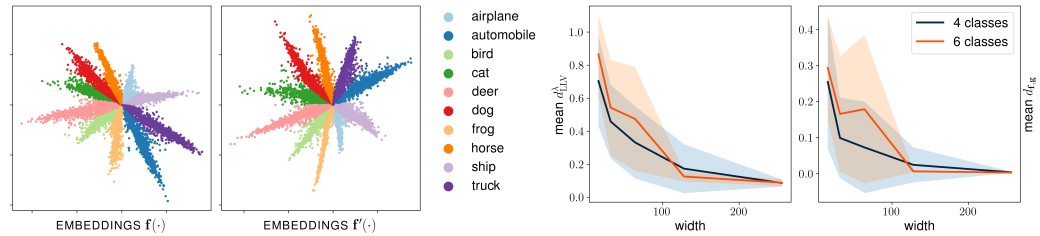

Figure 3: *(Left)* Embedding representations of two models trained on CIFAR-10. Representations for some of the labels are permuted, and $m_{\mathrm{CCA}}(\mathbf{f}(\mathbf{x}), \mathbf{f}'(\mathbf{x})) = 0.55$. *(Right)* Mean $d_{\mathrm{LLV}}^{\lambda}$ and $d_{\mathbf{f},\mathbf{g}}$ vs network width. Shaded area is standard deviation. Both mean and standard deviation decrease as the network width increases.

**Remark 4.8.** *It can be shown (Lemma E.8) that* $d_{\mathrm{SVD}}(\mathbf{L}^{\top}\mathbf{f}(\mathbf{x}), \mathbf{L}'^{\top}\mathbf{f}'(\mathbf{x}))$ *remains constant for models that are linearly equivalent to the members in the expression. That is, for any* $(\mathbf{f}^{*}, \mathbf{g}^{*}) \in \Theta$, *such that* $(\mathbf{f}^{*}, \mathbf{g}^{*}) \sim_{L} (\mathbf{f}, \mathbf{g})$, *we have that* $d_{\mathrm{SVD}}(\mathbf{L}^{\top}\mathbf{f}(\mathbf{x}), \mathbf{L}'^{\top}\mathbf{f}'(\mathbf{x})) = d_{\mathrm{SVD}}(\mathbf{L}^{*\top}\mathbf{f}^{*}(\mathbf{x}), \mathbf{L}'^{\top}\mathbf{f}'(\mathbf{x}))$. *A similar argument also holds for* $d_{\mathrm{SVD}}(\mathbf{N}^{\top}\mathbf{g}(y), \mathbf{N}'^{\top}\mathbf{g}'(y))$.

Theorem 4.7 shows that for the measures $d_{\mathrm{LLV}}^{\lambda}$ and $d_{\mathbf{f},\mathbf{g}}$, closeness in distribution (i.e., small values of $d_{\mathrm{LLV}}^{\lambda}$) bounds representational dissimilarity (as measured by $d_{\mathbf{f},\mathbf{g}}$).

Notice that in Theorem 4.7, we can freely choose any set of inputs $\mathcal{X}_{\mathrm{LLV}} \subset \mathcal{X}$ as long as they satisfy (1) and (2). Hence, if there is a choice making $t_2$ in Definition 4.4 as small as possible, it will also make $d_{\mathrm{SVD}}(\mathbf{N}^{\top}\mathbf{g}(y), \mathbf{N}'^{\top}\mathbf{g}'(y))$ small. There is also some flexibility for $\mathcal{Y}_{\mathrm{LLV}} \subset \mathcal{Y}$, where a careful choice of pivot point $y_0$ and the left-out label can make the term $t_1$ in Definition 4.4 small and, as a consequence, $d_{\mathrm{SVD}}(\mathbf{L}^{\top}\mathbf{f}(\mathbf{x}), \mathbf{L}'^{\top}\mathbf{f}'(\mathbf{x}))$ also small.

# 5 Experiments

In this section, we present our key experimental findings: (1) For models trained for classification on CIFAR-10 [32], we observe cases of similarly well-performing models with highly dissimilar representations, arising from a mechanism analogous to the constructive proof of Theorem 3.1 (Section 5.1); (2) On synthetic data, training wider neural networks results in a reduction of both the mean and variance of $d_{\mathrm{LLV}}^{\lambda}$ between models and leads to higher representational similarity (Section 5.2). Additional experiments illustrating the phenomena in Theorem 3.1 and visualizing the bound in Theorem 4.7 are summarized in Section 5.3. For all experiments we use $\lambda = 10^{-5}$. See Appendix F for implementation details. The code can be found on GitHub.[12]

## 5.1 Dissimilar Representations in CIFAR-10 Models with Similar Performance

**Experimental setup.** We trained classification models on CIFAR-10 [32] with two-dimensional embedding and unembedding representations. The embedding network is a ResNet18 [21], and the unembedding network consists of three fully connected layers of width 128, followed by the final representation layer.

**Results.** Embeddings of images with the same labels form clusters, and certain label pairs—such as "truck" and "automobile"—consistently appear as neighbors across retrainings (See Appendix F.5). However, some clusters are permuted: Fig. 3 *(Left)* shows two retrainings where, in the first model, the "truck" cluster lies between "automobile" and "ship", while in the second it lies between "automobile" and "horse". This mirrors the construction in Theorem 3.1, where models close in KL divergence have dissimilar representations. The models in Fig. 3 *(Left)* have $d_{\mathrm{KL}}$ of 1.13 (0→1) and 1.12 (1→0), similar test losses (1.19 and 1.18), but a large $d_{\mathrm{LLV}}^{\lambda}$ ($\sim 1.55$) and small $m_{\mathrm{CCA}}$ (0.55).

---

[12] github.com/bemigini/close-dist-rep-sim, DOI: 10.5281/zenodo.17249361.

## 5.2 Wider Networks Learn Closer Distributions and More Similar Representations

**Experimental setup.** We train models on a 2D classification task with 4, 6, 10 or 18 labels. Data are samples from a 2D Gaussian ($\mu = 0$, $\sigma = 3$), and labels are based on the angle to the first axis, consisting of a slice of the circle and the opposite slice (see Appendix F.2). We train twenty random seeds of models with two-dimensional representations. Each model consists of three fully connected layers with widths 16, 32, 64, 128 or 256. We retain only models achieving over 90% accuracy. For 4 classes, this yields 19 models at width 16 and 20 models for all other widths. For 6 classes, we retain 16, 17, 19, 19, and 20 models across the five widths, respectively.

**Results.** We find that $d_{\mathrm{LLV}}^\lambda$ and $d_{\mathbf{f},\mathbf{g}}$ between the models trained with wider networks have smaller both mean and standard deviation. In Fig. 3 *(Right)* we show the trend for models with 4 and 6 classes. Similar plots for 10 and 18 classes can be seen in Appendix F.7. While the bound in Theorem 4.7 is only non-vacuous when $d_{\mathbf{f},\mathbf{g}}((\mathbf{f}, \mathbf{g}), (\mathbf{f}', \mathbf{g}')) \leq 2M\epsilon < 1$ (i.e., when $\epsilon < 1/2M$) our plots suggest that there is a broader relationship between $d_{\mathbf{f},\mathbf{g}}$ and $d_{\mathrm{LLV}}^\lambda$, since the values of $d_{\mathrm{LLV}}^\lambda$ and $d_{\mathbf{f},\mathbf{g}}$ seem to be related even before the bound becomes non-vacuous.

## 5.3 Visualizing Representation Dissimilarity and the Bound from Theorem 4.7

Following the construction in Theorem 3.1, we simulate two models with fixed embeddings and unembeddings with increasing norm. We measure the KL divergence and $d_{\mathrm{LLV}}^\lambda$ between distributions; between representations, we measure the mean canonical correlation ($m_{\mathrm{CCA}}$) [45, 49, 52] and the maximum $d_{\mathrm{SVD}}$ of the embeddings over the input and label sets for $d_{\mathrm{LLV}}^\lambda$ (since this is what Theorem 4.7 bounds). The results are in Table 1: We find that as the unembeddings' norm, $\rho$, increases, $d_{\mathrm{KL}}$ decreases, while $d_{\mathrm{LLV}}^\lambda$ and $d_{\mathrm{SVD}}$ remain high and $m_{\mathrm{CCA}}$ remains low and almost constant. This aligns with the result in Theorem 3.1, while it displays that $d_{\mathrm{LLV}}^\lambda$ is robust to these model constructions.

Table 1: **Kullback–Leibler divergence rapidly vanishes while other measures stay constant.** Cells in the $d_{\mathrm{KL}}$ column are shaded red in proportion to their magnitude (darker $\Rightarrow$ larger value). As the unembedding norm $\rho$ grows, $d_{\mathrm{KL}}$ drops by almost four orders of magnitude, whereas $d_{\mathrm{LLV}}^\lambda$ and the maximal $d_{\mathrm{SVD}}$ between embeddings remain virtually unchanged.

| $\rho$ | $d_{\mathrm{KL}}$ | $d_{\mathrm{LLV}}^\lambda$ | $m_{\mathrm{CCA}}$ | $\max d_{\mathrm{SVD}}$ |
|---|---|---|---|---|
| 3 | 0.8866 | 1.3176 | 0.0006 | 0.9996 |
| 6 | 0.2230 | 1.3169 | 0.0007 | 0.9998 |
| 9 | 0.0369 | 1.3171 | 0.0004 | 0.9995 |
| 12 | 0.0055 | 1.3178 | 0.0006 | 0.9998 |
| 15 | 0.0008 | 1.3175 | 0.0008 | 0.9999 |
| 18 | 0.0001 | 1.3172 | 0.0010 | 0.9998 |

Also, for these model pairs and models trained on synthetic data, we find that the bound in Theorem 4.7 is always satisfied (refer to Appendix F.6 for details and to Fig. 16 for a visualization). However, for models trained on CIFAR-10, the distributions are not close enough for the bound to be non-vacuous.

## 6 Discussion

**Limitations.** To the best of our knowledge, this work provides the first theoretical result that upper-bounds representational dissimilarity by a distributional distance (Theorem 4.7)—a relationship that Theorem 3.1 shows is far from obvious. The bound is nonetheless not tight: it grows linearly even though the empirical trend of $\max d_{\mathrm{SVD}}$ is clearly sub-linear (Fig. 16), and it presently requires the label set to contain every class but one. We conjecture that a tighter, potentially non-linear bound could be obtained and that $d_{\mathrm{LLV}}^\lambda$ would remain a valid distance when computed on a suitably *diverse* subset of labels. The distributional distance itself (Definition 4.4) inherits the same *label-diversity* assumption (Definition 2.1), which prevents its applicability to the whole model family of Equation (1): diversity is more likely to hold in language models, but may fail in standard image classifiers—especially when the number of classes is smaller than the representation dimension [52]. It was recently proved by Marconato et al. [43] that, when relaxing the diversity assumption, the model family in Equation (1) can be identified up to an *extended-linear* equivalence relation. Extending our theory to that setting would enable comparing models with different representation dimension [43].

**Alternative similarity measures.** The invariances of our dissimilarity measure, $d_{\mathbf{f},\mathbf{g}}$, comprise a subset of all invertible linear transforms and always consider embeddings and unembeddings in combination with each other (see Appendix E.7). By contrast, much prior work employs CCA-based measures that are invariant to *any* linear transformation [45, 49] and only measures (dis)similarity

between embeddings. Kornblith et al. [31] argue that similarity measures should be invariant only to orthogonal transformations, based on the task-relevance of relative activation scales and the invariance of gradient-descent dynamics [35]. What measure best captures representational similarity is still an open question [3], with different approaches reflecting distinct goals and trade-offs [30]. Our goal is not to claim our measure is *the* right one, but to present a pairing of dissimilarity measure and distributional distance that enables theoretical analysis and captures key empirical aspects of representational similarity. Extending such guarantees to other (dis)similarity measures and distributional divergences is an important direction for future work.

**Do similarly retrained models learn similar representations?** This has been extensively explored in empirical studies [31, 38, 46, 51, 52, 56, 65]. Retraining with a different seed can result in different likelihoods, and we show that even models with near-zero KL divergence can have dissimilar internal representations. While the existence of constructions such as the one in Theorem 3.1 does not imply that a model will learn them during training, we find it interesting that a mechanism resembling our construction actually emerged in our CIFAR-10 experiments (Section 5.1), where we observe large representational dissimilarity despite close distributions in terms of KL divergence. Such dissimilarity can arise when most output labels have negligible probability—for example, an image might be confidently classified as a "cat" or "dog", with all other classes (e.g., "spaceship", "chair") having near-zero probabilities; and in language modeling, often only few next-tokens are plausible. Overall, these observations suggest that similar performance alone is insufficient to ensure representational similarity [6], though higher network capacity may help, see Section 5.2 and [52, 56].

**Identifiability and robustness.** In nonlinear ICA [10, 18, 20, 24, 28, 57, 72], disentangled and causal representation learning [2, 7, 11, 37, 39, 40, 44, 63, 64, 69–71] models are typically constructed so that their likelihood-maximising solutions are unique up to a fixed (and 'small') set of ambiguities, as guaranteed by identifiability [24, 68]. Robustness when the likelihood is only approximately maximized has been studied less, though Buchholz and Schölkopf [9] analyzed isometric-mixing ICA; Sliwa et al. [55] empirically probed *independent mechanism analysis* [19]; and Träuble et al. [62] showed that independence-based disentanglement lacks robustness when true factors are correlated. Our results further highlight the importance of understanding robustness in nonlinear representation learning.[13] In particular, our Corollary 3.2 suggests that relying solely on identifiability results—such as those by Reizinger et al. [51]—may be insufficient to guarantee that similar representations are practically attainable when optimizing the cross-entropy loss: additional assumptions may be needed.

**Conclusion and Outlook.** We have studied the link between distributional closeness and representational similarity for the embedding and unembedding layers of our model family, focusing on different instances of the *same* model class. Extending our analysis to earlier layers would first require identifiability results for intermediate network representations, which remains an open problem in representation learning [25, 28]. In practice, learned representations often appear surprisingly robust across datasets, architectures, and training objectives [3, 4, 17, 23, 31, 46], and explaining this broader robustness is an exciting direction for future work. By clarifying when distributional closeness does—and does not—imply representational similarity for embeddings and unembeddings within our model family, we take an initial step toward a principled theory of representational similarity.

## Acknowledgments and Disclosure of Funding

We thank Simon Buchholz for helpful discussions and feedback. Beatrix M. G. Nielsen was supported by the Danish Pioneer Centre for AI, DNRF grant number P1. E.M. acknowledges support from TANGO, Grant Agreement No. 101120763, funded by the European Union. Views and opinions expressed are however those of the author(s) only and do not necessarily reflect those of the European Union or the European Health and Digital Executive Agency (HaDEA). Neither the European Union nor the granting authority can be held responsible for them. A.D. acknowledges funding from the Helmholtz Foundation Model Initiative. L.G. was supported by the Danish Data Science Academy, which is funded by the Novo Nordisk Foundation (NNF21SA0069429).

---

[13]$\varepsilon$-non-identifiability [50] is closely related: our work illustrates it for learned representations and highlights that *it depends on the choice of divergence*—holding in $d_{\mathrm{KL}}$ (Theorem 3.1) but not in $d_{\mathrm{LLV}}^{\lambda}$ (Theorem 4.7).

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

# Appendix

## Table of Contents

# A Notation

We here present a table (Table 2) with the notation used in the paper.

Table 2: **Notation.** We use bold letters for vectors and non-bold letters for scalars.

| Symbol | Description | Reference |
|---|---|---|
| $\mathbf{x}$ | vector and vector-valued variables | Section 2 |
| $y$ | scalar-valued variable | Section 2 |
| $(\mathbf{f}, \mathbf{g})$ | a model with embedding and unembedding functions $\mathbf{f}$ and $\mathbf{g}$, see Equation (1) | Section 2 |
| $p_{\mathbf{f},\mathbf{g}}$ | distributions of a model $(\mathbf{f}, \mathbf{g})$ | Section 2 |
| $\mathbf{f}_0(\mathbf{x})$ | the displaced embedding functions $\mathbf{f}(\mathbf{x}) - \mathbf{f}(\mathbf{x}_0)$ | Section 2 |
| $\mathbf{g}_0(y)$ | the displaced unembedding functions $\mathbf{g}(y) - \mathbf{g}(y_0)$ | Section 2 |
| $\mathbf{L}$ | the matrix with columns $\mathbf{g}_0(y)$, see Theorem 2.2 | Section 2 |
| $d_{\mathrm{KL}}$ | the distance between distributions arising from the KL divergence | Section 3 |
| $\mathrm{Var}_{\mathbf{x}}[\cdot]$ | the variance over the inputs from some input distribution, $p_{\mathbf{x}}$ | Section 4 |
| $\mathrm{Unif}(\mathcal{Y})$ | the uniform distribution over the elements of $\mathcal{Y}$ | Section 4 |
| $d_{\mathrm{LLV}}^{\lambda}$ | the log-likelihood variance distance with weighting parameter $\lambda$ | Section 4 |
| $d_{\mathrm{SVD}}$ | the mean PLS-SVD distance between random vectors | Section 4 |
| $d_{\mathbf{f},\mathbf{g}}$ | the distance between representations for our model class | Section 4 |
| $\mathbf{\Sigma}_{\mathbf{zw}}$ | the cross-covariance matrix of random vectors $\mathbf{z}$, $\mathbf{w}$ | Section 4 |
| $\mathbf{N}$ | the matrix with columns $\mathbf{f}_0(\mathbf{x})$ | Section 4 |
| $m_{\mathrm{CCA}}$ | the mean canonical correlation coefficient | Appendix D.2 |

# B Identifiability Result and Proof

We present a unified and slightly adapted version of the identifiability results from [28, 34, 52], see also [43, Corollary 6].

**Theorem 2.2** (Linear Identifiability [28, 34, 52]). *Let $(\mathbf{f}, \mathbf{g}), (\mathbf{f}', \mathbf{g}') \in \Theta$, and $(\mathbf{f}, \mathbf{g})$ satisfy the diversity condition (Definition 2.1). Let $\mathbf{L}$ (resp. $\mathbf{L}'$) be the matrix with columns $\mathbf{g}_0(y)$ (resp. $\mathbf{g}'_0(y)$), and let $\mathbf{A} := \mathbf{L}^{-\top}\mathbf{L}'^{\top} \in \mathbb{R}^{M \times M}$. Then:*

$$p_{\mathbf{f},\mathbf{g}}(y \mid \mathbf{x}) = p_{\mathbf{f}',\mathbf{g}'}(y \mid \mathbf{x}), \ \forall(\mathbf{x}, y) \in \mathcal{X} \times \mathcal{Y} \implies (\mathbf{f}, \mathbf{g}) \sim_L (\mathbf{f}', \mathbf{g}') \tag{4}$$

*where the equivalence relation $\sim_L$ is defined by*

$$(\mathbf{f}, \mathbf{g}) \sim_L (\mathbf{f}', \mathbf{g}') \iff \begin{cases} \mathbf{f}(\mathbf{x}) = \mathbf{A}\mathbf{f}'(\mathbf{x}) \\ \mathbf{g}_0(y) = \mathbf{A}^{-\top}\mathbf{g}'_0(y) \end{cases}. \tag{5}$$

*Proof.* We first prove that $p_{\mathbf{f},\mathbf{g}}(y \mid \mathbf{x}) = p_{\mathbf{f}',\mathbf{g}'}(y \mid \mathbf{x}), \ \forall(\mathbf{x}, y) \in \mathcal{X} \times \mathcal{Y} \implies \mathbf{f}(\mathbf{x}) = \mathbf{A}\mathbf{f}'(\mathbf{x})$ for $\mathbf{A} := \mathbf{L}^{-\top}\mathbf{L}'^{\top}$ invertible.

By assumption, the models $(\mathbf{f}, \mathbf{g}), (\mathbf{f}', \mathbf{g}')$ have equal likelihoods. Moreover, by construction, the two models have representations in $\mathbb{R}^M$. Let $Z(\mathbf{x}, \mathcal{Y}) = \sum_{y_j \in \mathcal{Y}} \exp(\mathbf{f}(\mathbf{x})^\top \mathbf{g}(y_j))$, and similarly for $Z'(\mathbf{x}, \mathcal{Y})$. Then

$$p_{\mathbf{f},\mathbf{g}}(y \mid \mathbf{x}) = p_{\mathbf{f}',\mathbf{g}'}(y \mid \mathbf{x}) \tag{15}$$

$$\mathbf{f}(\mathbf{x})^\top \mathbf{g}(y) - \log(Z(\mathbf{x}, \mathcal{Y})) = \mathbf{f}'(\mathbf{x})^\top \mathbf{g}'(y) - \log(Z'(\mathbf{x}, \mathcal{Y})) \tag{16}$$

for all $\mathbf{x} \in \mathcal{X}$ and all $y \in \mathcal{Y}$. In particular, this equation holds for any fixed $y$. From the diversity condition (Definition 2.1), we can choose $M + 1$ $y$'s, $y_0, \ldots, y_M$, such that the displaced unembeddings $\{\mathbf{g}_0(y_i)\}_{i=1}^M$ are linearly independent. By subtracting the log-conditional distribution for the pivot $y_0 \in \mathcal{Y}$, we obtain for each $y_i \in \mathcal{Y}$ the following:

$$\mathbf{f}(\mathbf{x})^\top \mathbf{g}(y_i) - \mathbf{f}(\mathbf{x})^\top \mathbf{g}(y_0) + \log(Z(\mathbf{x}, \mathcal{Y})) - \log(Z(\mathbf{x}, \mathcal{Y}))$$
$$= \mathbf{f}'(\mathbf{x})^\top \mathbf{g}'(y_i) - \mathbf{f}'(\mathbf{x})^\top \mathbf{g}'(y_0) + \log(Z'(\mathbf{x}, \mathcal{Y})) - \log(Z'(\mathbf{x}, \mathcal{Y})) \tag{17}$$

$$\mathbf{f}(\mathbf{x})^\top \mathbf{g}(y_i) - \mathbf{f}(\mathbf{x})^\top \mathbf{g}(y_0) = \mathbf{f}'(\mathbf{x})^\top \mathbf{g}'(y_i) - \mathbf{f}'(\mathbf{x})^\top \mathbf{g}'(y_0) \tag{18}$$

$$\mathbf{f}(\mathbf{x})^\top \mathbf{g}_0(y_i) = \mathbf{f}'(\mathbf{x})^\top \mathbf{g}'_0(y_i), \tag{19}$$

where the last passage holds by definition of $\mathbf{g}_0(y), \mathbf{g}'_0(y)$, see Section 2. Let $\mathbf{L}$ be the matrix which has $\mathbf{g}_0(y_i)$ as columns and $\mathbf{L}'$ be the matrix which has $\mathbf{g}'_0(y_i)$ as columns. Notice that the diversity condition (Definition 2.1) implies that $\mathbf{L}$ is an invertible matrix. We can then stack the equations to get

$$\mathbf{L}^\top \mathbf{f}(\mathbf{x}) = \mathbf{L}'^\top \mathbf{f}'(\mathbf{x}) \tag{20}$$

and since $\mathbf{L}$ is invertible,

$$\mathbf{f}(\mathbf{x}) = \mathbf{L}^{-\top}\mathbf{L}'^\top \mathbf{f}'(\mathbf{x}). \tag{21}$$

If we set $\mathbf{A} := (\mathbf{L}'\mathbf{L}^{-1})^\top$, we only need to show that $\mathbf{A}$ is invertible. Using the diversity condition, we pick points $\mathbf{x}_0, \ldots, x_M \in \mathcal{X}$ such that the displaced embedding vectors $\{\mathbf{f}_0(\mathbf{x}_i)\}_{i=1}^M$ are linearly independent. Let $\mathbf{N}$ be the matrix with $\mathbf{f}_0(\mathbf{x}_i)$ as columns and $\mathbf{N}'$ be the matrix with $\mathbf{f}'_0(\mathbf{x}_i)$ as columns. Notice that, from the diversity condition $\mathbf{N}$ is an invertible matrix. Then

$$\mathbf{N} = \mathbf{A}\mathbf{N}'. \tag{22}$$

Since any two matrices, $\mathbf{B}, \mathbf{C} \in \mathbb{R}^{M \times M}$, have the property $\mathrm{rank}(\mathbf{BC}) \leq \min(\mathrm{rank}(\mathbf{B}), \mathrm{rank}(\mathbf{C}))$, and $\mathbf{N}$ has rank equal to $M$, we have that necessarily also $\mathbf{A}$ and $\mathbf{N}'$ have rank $M$. In particular, $\mathbf{A}$ is an invertible matrix.

Next we prove that $p_{\mathbf{f},\mathbf{g}}(y \mid \mathbf{x}) = p_{\mathbf{f}',\mathbf{g}'}(y \mid \mathbf{x}), \ \forall(\mathbf{x}, y) \in \mathcal{X} \times \mathcal{Y} \implies \mathbf{g}_0(y) = \mathbf{A}^{-\top}\mathbf{g}'_0(y)$ for $\mathbf{A} := \mathbf{L}^{-\top}\mathbf{L}'^{\top}$ invertible.

As before, we have that

$$\mathbf{f}(\mathbf{x})^\top \mathbf{g}(y) - \log(Z(\mathbf{x}, \mathcal{Y})) = \mathbf{f}'(\mathbf{x})^\top \mathbf{g}'(y) - \log(Z'(\mathbf{x}, \mathcal{Y})) \tag{23}$$

holds for all $\mathbf{x} \in \mathcal{X}$ and all $y \in \mathcal{Y}$. In particular, this equation holds for any specific $\mathbf{x}$. From the diversity condition (Definition 2.1), we can choose $M + 1$ $\mathbf{x}$'s, $\mathbf{x}_0, \ldots, \mathbf{x}_M$, such that the displaced embeddings $\{\mathbf{f}_0(\mathbf{x}_i)\}_{i=1}^M$ are linearly independent. By subtracting the log-conditional distribution for the pivot $\mathbf{x}_0 \in \mathcal{X}$, we obtain for each $\mathbf{x}_i \in \mathcal{X}$ the following:

$$\mathbf{f}(\mathbf{x}_i)^\top \mathbf{g}(y) - \mathbf{f}(\mathbf{x}_0)^\top \mathbf{g}(y) + \log(Z(\mathbf{x}_0, \mathcal{Y})) - \log(Z(\mathbf{x}_i, \mathcal{Y}))$$
$$= \mathbf{f}'(\mathbf{x}_i)^\top \mathbf{g}'(y) - \mathbf{f}'(\mathbf{x}_0)^\top \mathbf{g}'(y) + \log(Z'(\mathbf{x}_0, \mathcal{Y})) - \log(Z'(\mathbf{x}_i, \mathcal{Y})) \tag{24}$$

$$\mathbf{f}(\mathbf{x}_i)^\top \mathbf{g}(y) - \mathbf{f}(\mathbf{x}_0))^\top \mathbf{g}(y) = \mathbf{f}'(\mathbf{x}_i)^\top \mathbf{g}'(y) - \mathbf{f}'(\mathbf{x}_0)^\top \mathbf{g}'(y) + c_i \tag{25}$$

$$\mathbf{f}_0(\mathbf{x}_i)^\top \mathbf{g}(y) = \mathbf{f}_0'(\mathbf{x}_i)^\top \mathbf{g}'(y) + c_i \,, \tag{26}$$

where

$$c_i = \log\left(\frac{Z'(\mathbf{x}_0, \mathcal{Y})}{Z(\mathbf{x}_0, \mathcal{Y})}\right) + \log\left(\frac{Z(\mathbf{x}_i, \mathcal{Y})}{Z'(\mathbf{x}_i, \mathcal{Y})}\right) \,. \tag{27}$$

Let $\mathbf{N}$ be the matrix with $\mathbf{f}_0(\mathbf{x}_i)$ as columns, let $\mathbf{N}'$ be the matrix with $\mathbf{f}_0'(\mathbf{x}_i)$ as columns and let $\mathbf{c}$ be the vector with $c_i$ as entries. Then, since $\mathbf{N}$ is invertible

$$\mathbf{N}^\top \mathbf{g}(y) = \mathbf{N}'^\top \mathbf{g}'(y) + \mathbf{c} \tag{28}$$

$$\mathbf{g}(y) = \mathbf{N}^{-\top} \mathbf{N}'^\top \mathbf{g}'(y) + \mathbf{N}^{-\top} \mathbf{c} \,. \tag{29}$$

Since we found in Equation (22) that $\mathbf{A}$ is invertible, we have that

$$\mathbf{N}' = \mathbf{A}^{-1} \mathbf{N} \,. \tag{30}$$

Therefore, we get

$$\mathbf{g}(y) = \mathbf{N}^{-\top} \mathbf{N}'^\top \mathbf{g}'(y) + \mathbf{N}^{-\top} \mathbf{c} \tag{31}$$

$$\mathbf{g}(y) = \mathbf{N}^{-\top} (\mathbf{A}^{-1} \mathbf{N})^\top \mathbf{g}'(y) + \mathbf{N}^{-\top} \mathbf{c} \tag{32}$$

$$\mathbf{g}(y) = \mathbf{N}^{-\top} \mathbf{N}^\top \mathbf{A}^{-\top} \mathbf{g}'(y) + \mathbf{N}^{-\top} \mathbf{c} \tag{33}$$

$$\mathbf{g}(y) = \mathbf{A}^{-\top} \mathbf{g}'(y) + \mathbf{b} \,, \tag{34}$$

where $\mathbf{b} = \mathbf{N}^{-\top} \mathbf{c}$. So, considering the displaced unembedding vectors, we get:

$$\mathbf{g}_0(y) = \mathbf{g}(y) - \mathbf{g}(y_0) \tag{35}$$

$$= \mathbf{A}^{-\top} \mathbf{g}'(y) + \mathbf{b} - \mathbf{A}^{-\top} \mathbf{g}'(y_0) - \mathbf{b} \tag{36}$$

$$= \mathbf{A}^{-\top} \mathbf{g}_0'(y) \tag{37}$$

This proves the claim. $\qquad\square$

# C   Models close in KL divergence can have dissimilar representations - details

## C.1   KL goes to Zero but Representations are Dissimilar

Here, we restate Theorem 3.1 in full detail:

**Theorem** (Formal statement of Theorem 3.1). *Let $(\mathbf{f}, \mathbf{g}_\rho) \in \Theta$ be a model with representations in $\mathbb{R}^M$. Assume $M \geq 2$. Let $k := |\mathcal{Y}|$ be the total number of unembeddings and assume $k \geq M + 1$. Assume the model satisfies the following requirements:*

  *i) The unembeddings have fixed norm, that is $\|\mathbf{g}_\rho(y_i)\| = \|\mathbf{g}_\rho(y_j)\| = \rho$, for all $y_i, y_j \in \mathcal{Y}$, where $\rho \in \mathbb{R}^+$.*

  *ii) For every $\mathbf{x}_i \in \mathcal{X}$, let there be one element $y_j \in \mathcal{Y}$ for which the cosine similarity $\cos(\mathbf{f}(\mathbf{x}_i), \mathbf{g}_\rho(y_j)) > 0$ and $\cos(\mathbf{f}(\mathbf{x}_i), \mathbf{g}_\rho(y_j)) > \cos(\mathbf{f}(\mathbf{x}_i), \mathbf{g}(y_\ell))$ for all $y_\ell \neq y_j$.*

*Let $(\mathbf{f}', \mathbf{g}'_\rho)$ be a model which also satisfies i) and ii) and which is related to $(\mathbf{f}, \mathbf{g}_\rho)$ in the following way:*

  *iii) For every $\mathbf{x}_i \in \mathcal{X}$, $\|\mathbf{f}'(\mathbf{x}_i)\| = \|\mathbf{f}(\mathbf{x}_i)\| = c(\mathbf{x}_i)$.*

  *iv) For $y_j = \operatorname{argmax}_{y \in \mathcal{Y}}(p(y|\mathbf{x}_i))$ we have that the angle between $\mathbf{f}'(\mathbf{x}_i)$ and $\mathbf{g}'_\rho(y_j)$ is equal to the angle between $\mathbf{f}(\mathbf{x}_i)$ and $\mathbf{g}_\rho(y_j)$, in particular, $\cos(\mathbf{f}(\mathbf{x}_i), \mathbf{g}_\rho(y_j)) = \cos(\mathbf{f}'(\mathbf{x}_i), \mathbf{g}'_\rho(y_j))$.*

*Then, making pairs of models satisfying assumptions i) to iv) and with increasing values of $\rho$ gives us a sequence of pairs of models for which we have:*

$$d_{\mathrm{KL}}(p_{\mathbf{f},\mathbf{g}_\rho}, p_{\mathbf{f}',\mathbf{g}'_\rho}) \to 0 \quad for \quad \rho \to \infty. \tag{38}$$

*Also, it is possible to construct a model $(\mathbf{f}', \mathbf{g}'_\rho) \in \Theta$ which satisfies the requirements above, but where as $\rho \to \infty$, the embeddings stay fixed and $\mathbf{f}'(\mathbf{x})$ is not an invertible linear transformation of $\mathbf{f}(\mathbf{x})$.*

*Proof.* Let the models $(\mathbf{f}, \mathbf{g}_\rho), (\mathbf{f}', \mathbf{g}'_\rho) \in \Theta$ be as described above. Below, we use the shorthand $p := p_{\mathbf{f},\mathbf{g}_\rho}$ and $p' := p_{\mathbf{f}',\mathbf{g}'_\rho}$.

We prove the results in two parts. (1) We show that, for any two models satisfying the requirements i) to iv) above, we get that $d_{\mathrm{KL}}(p, p') \to 0$ as $\rho \to \infty$. (2) We then specifically show how construct a model $(\mathbf{f}, \mathbf{g}_\rho)$ and a model $(\mathbf{f}', \mathbf{g}'_\rho)$, which satisfies the requirements i) to iv) but where as $\rho \to \infty$, the embeddings stay fixed and $\mathbf{f}'(\mathbf{x})$ is not an invertible linear transformation of $\mathbf{f}(\mathbf{x})$.

(1) Since $D_{\mathrm{KL}}(p(y|\mathbf{x}_i)\|p'(y|\mathbf{x}_i)) \to 0$ for all $\mathbf{x}_i \in \mathcal{X}$ implies that $d_{\mathrm{KL}}(p, p') \to 0$, we will show that $D_{\mathrm{KL}}(p(y|\mathbf{x}_i)\|p'(y|\mathbf{x}_i)) \to 0$ for all $\mathbf{x}_i \in \mathcal{X}$. Fix $\mathbf{x}_i \in \mathcal{X}$. For notational brevity, we denote with $c_i \in \mathbb{R}^+$ the value of the norm of the embeddings on $\mathbf{x}_i$, i.e., $\|\mathbf{f}(\mathbf{x}_i)\| = \|\mathbf{f}'(\mathbf{x}_i)\| = c(\mathbf{x}_i)$ (assumption iii)). We then consider the KL divergence

$$D_{\mathrm{KL}}(p(y|\mathbf{x}_i)\|p'(y|\mathbf{x}_i)) = \sum_{j=1}^k p(y_j|\mathbf{x}_i) \log \frac{p(y_j|\mathbf{x}_i)}{p'(y_j|\mathbf{x}_i)}. \tag{39}$$

Since $D_{\mathrm{KL}}(p(y|\mathbf{x}_i)\|p'(y|\mathbf{x}_i))$ involves a sum over $j$, $D_{\mathrm{KL}}(p(y|\mathbf{x}_i)\|p'(y|\mathbf{x}_i))$ will go to zero if each term in the sum goes to zero. To show that each term of this sum goes to zero, we first consider the term for $y_j = \arg\max_{y \in \mathcal{Y}}(p(y|\mathbf{x}_i))$. In this case, we have that

$$p(y_j|\mathbf{x}_i) = \frac{\exp(\mathbf{f}(\mathbf{x}_i)^\top \mathbf{g}_\rho(y_j))}{\sum_{\ell=1}^k \exp(\mathbf{f}(\mathbf{x}_i)^\top \mathbf{g}_\rho(y_\ell))} \tag{40}$$

$$= \left(1 + \sum_{\ell=1, \ell \neq j}^k \exp(\mathbf{f}(\mathbf{x}_i)^\top \mathbf{g}_\rho(y_\ell) - \mathbf{f}(\mathbf{x}_i)^\top \mathbf{g}_\rho(y_j))\right)^{-1}. \tag{41}$$

From iii) we have $\|\mathbf{f}(\mathbf{x}_i)\| = c_i$ and from i) we have that $\|\mathbf{g}_\rho(y_j)\| = \rho$. We can therefore write

$$\mathbf{f}(\mathbf{x}_i)^\top \mathbf{g}_\rho(y_j) = \cos(\mathbf{f}(\mathbf{x}_i), \mathbf{g}_\rho(y_j)) \cdot c_i \cdot \rho \,, \tag{42}$$

and substituting this into Equation (41), we see that

$$p(y_j|\mathbf{x}_i) = \left( 1 + \sum_{\ell=1, \ell \neq j}^{k} \exp\left( c_i \cdot \rho \cdot (\cos(\mathbf{f}(\mathbf{x}_i), \mathbf{g}_\rho(y_\ell)) - \cos(\mathbf{f}(\mathbf{x}_i), \mathbf{g}_\rho(y_j))) \right) \right)^{-1}. \tag{43}$$

Now since by assumption ii) $\cos(\mathbf{f}(\mathbf{x}_i), \mathbf{g}_\rho(y_j)) > 0$ and for all $y_\ell \neq y_j$

$$\cos(\mathbf{f}(\mathbf{x}_i), \mathbf{g}_\rho(y_j)) > \cos(\mathbf{f}(\mathbf{x}_i), \mathbf{g}_\rho(y_\ell)) \,, \tag{44}$$

we have that

$$\cos(\mathbf{f}(\mathbf{x}_i), \mathbf{g}_\rho(y_\ell)) - \cos(\mathbf{f}(\mathbf{x}_i), \mathbf{g}_\rho(y_j)) < 0 \,, \tag{45}$$

Taking the limit in $\rho$, we get

$$\exp(c \cdot \rho \cdot (\cos(\mathbf{f}(\mathbf{x}_i), \mathbf{g}_\rho(y_\ell)) - \cos(\mathbf{f}(\mathbf{x}_i), \mathbf{g}_\rho(y_j)))) \to 0 \quad \text{for} \quad \rho \to \infty \tag{46}$$

Since every term of the sum in Equation (43) goes to zero for $\rho \to \infty$, the whole sum goes to zero, which means that from Equation (43) we get that

$$p(y_j|\mathbf{x}_i) \to \frac{1}{1+0} = 1 \quad \text{for} \quad \rho \to \infty \,. \tag{47}$$

Similarly, we also have that

$$p'(y_j|\mathbf{x}_i) \to 1 \quad \text{for} \quad \rho \to \infty \,. \tag{48}$$

This means that for $y_j = \operatorname{argmax}_{y \in \mathcal{Y}} p(y|\mathbf{x}_i)$, from combining Equation (47), Equation (48) and Equation (39) we get that

$$p(y_j|\mathbf{x}_i) \log \frac{p(y_j|\mathbf{x}_i)}{p'(y_j|\mathbf{x}_i)} \to 1 \cdot \log \frac{1}{1} = 0 \quad \text{for} \quad \rho \to \infty \,. \tag{49}$$

Next, we consider the case where $y_j \neq \operatorname{argmax}_{y \in \mathcal{Y}} p(y|\mathbf{x}_i)$. In this case, we consider

$$p(y_j|\mathbf{x}_i) \log \frac{p(y_j|\mathbf{x}_i)}{p'(y_j|\mathbf{x}_i)} = p(y_j|\mathbf{x}_i)\left( \log p(y_j|\mathbf{x}_i) - \log p'(y_j|\mathbf{x}_i) \right) \tag{50}$$

$$= p(y_j|\mathbf{x}_i)\left( \mathbf{f}(\mathbf{x}_i)^\top \mathbf{g}_\rho(y_j) - \log Z(\mathbf{x}_i) - \mathbf{f}'(\mathbf{x}_i)^\top \mathbf{g}'_\rho(y_j) + \log Z'(\mathbf{x}_i) \right) \tag{51}$$

$$= p(y_j|\mathbf{x}_i)\left( \mathbf{f}(\mathbf{x}_i)^\top \mathbf{g}_\rho(y_j) - \mathbf{f}'(\mathbf{x}_i)^\top \mathbf{g}'_\rho(y_j) \right) + p(y_j|\mathbf{x}_i) \log \frac{Z'(\mathbf{x}_i)}{Z(\mathbf{x}_i)} \,. \tag{52}$$

We will consider the two terms in Equation (52) separately. First, we consider

$$p(y_j|\mathbf{x}_i)\left( \mathbf{f}(\mathbf{x}_i)^\top \mathbf{g}_\rho(y_j) - \mathbf{f}'(\mathbf{x}_i)^\top \mathbf{g}'_\rho(y_j) \right) \,. \tag{53}$$

Similarly as for Equation (42), we have

$$\mathbf{f}(\mathbf{x}_i)^\top \mathbf{g}_\rho(y_j) = \cos(\mathbf{f}(\mathbf{x}_i), \mathbf{g}_\rho(y_j)) \cdot c_i \cdot \rho \,. \tag{54}$$

We use this to rewrite Equation (53) and see that

$$|\mathbf{f}(\mathbf{x}_i)^\top \mathbf{g}_\rho(y_j) - \mathbf{f}'(\mathbf{x}_i)^\top \mathbf{g}'_\rho(y_j)| = |\cos(\mathbf{f}(\mathbf{x}_i), \mathbf{g}_\rho(y_j)) \cdot c_i \cdot \rho - \cos(\mathbf{f}'(\mathbf{x}_i), \mathbf{g}'_\rho(y_j)) \cdot c_i \cdot \rho| \tag{55}$$

$$= |c_i \cdot \rho \cdot (\cos(\mathbf{f}(\mathbf{x}_i), \mathbf{g}_\rho(y_j)) - \cos(\mathbf{f}'(\mathbf{x}_i), \mathbf{g}'_\rho(y_j)))| \tag{56}$$

$$\leq 2c_i \cdot \rho \,. \tag{57}$$

This gives us that

$$p(y_j|\mathbf{x}_i)|\mathbf{f}(\mathbf{x}_i)^\top \mathbf{g}_\rho(y_j) - \mathbf{f}'(\mathbf{x}_i)^\top \mathbf{g}'_\rho(y_j)| \leq 2c_i\rho \cdot p(y_j|\mathbf{x}_i) \tag{58}$$

Since $|x| \to 0$ implies that $x \to 0$ and we have shown that the absolute values of the term is upper bounded by $2c_i \rho \cdot p(y_j | \mathbf{x}_i)$, we will show that the term goes to zero by showing that $2c_i \rho \cdot p(y_j | \mathbf{x}_i)$ goes to zero. We see that

$$2c_i \rho \cdot p(y_j | \mathbf{x}_i) = \frac{2c_i \cdot \rho \cdot \exp(\mathbf{f}(\mathbf{x}_i)^\top \mathbf{g}_\rho(y_j))}{\sum_{\ell=1}^k \exp(\mathbf{f}(\mathbf{x}_i)^\top \mathbf{g}_\rho(y_\ell))} \tag{59}$$

$$= \frac{2c_i \cdot \rho}{1 + \sum_{\ell=1, \ell \neq j}^k \exp(\mathbf{f}(\mathbf{x}_i)^\top \mathbf{g}_\rho(y_\ell) - \mathbf{f}(\mathbf{x}_i)^\top \mathbf{g}_\rho(y_j))} \tag{60}$$

$$= \frac{2c_i \cdot \rho}{1 + \sum_{\ell=1, \ell \neq j}^k \exp(c_i \cdot \rho \cdot (\cos(\mathbf{f}(\mathbf{x}_i), \mathbf{g}_\rho(y_\ell)) - \cos(\mathbf{f}(\mathbf{x}_i), \mathbf{g}_\rho(y_j))))} \tag{61}$$

$$= 2c_i \left/ \left( 1/\rho + \sum_{\ell=1, \ell \neq j}^k \frac{\exp\left(c_i \cdot \rho \cdot (\cos(\mathbf{f}(\mathbf{x}_i), \mathbf{g}_\rho(y_\ell)) - \cos(\mathbf{f}(\mathbf{x}_i), \mathbf{g}_\rho(y_j)))\right)}{\rho} \right) \right. . \tag{62}$$

Consider now $y_r = \operatorname{argmax}_{y \in \mathcal{Y}} p(y | \mathbf{x}_i)$, recall we have $y_j \neq y_r$. Considering this $y_r$, we have by assumption ii) that $\cos(\mathbf{f}(\mathbf{x}_i), \mathbf{g}_\rho(y_r)) > 0$ and

$$\cos(\mathbf{f}(\mathbf{x}_i), \mathbf{g}_\rho(y_r)) - \cos(\mathbf{f}(\mathbf{x}_i), \mathbf{g}_\rho(y_j)) > 0, \tag{63}$$

Since for $\alpha \in \mathbb{R}^+$, we have $\exp(\alpha x)/x \to \infty$ for $x \to \infty$, for this $y_r$, we have that

$$\frac{\exp(c_i \cdot \rho \cdot (\cos(\mathbf{f}(\mathbf{x}_i), \mathbf{g}_\rho(y_r)) - \cos(\mathbf{f}(\mathbf{x}_i), \mathbf{g}_\rho(y_j))))}{\rho} \to \infty \quad \text{for} \quad \rho \to \infty. \tag{64}$$

We can now define the denominator of Equation (62) as

$$d(\rho) := \frac{1}{\rho} + \frac{\exp\left(c_i \rho \left(\cos(\mathbf{f}(\mathbf{x}_i), \mathbf{g}_\rho(y_r)) - \cos(\mathbf{f}(\mathbf{x}_i), \mathbf{g}_\rho(y_j))\right)\right)}{\rho}$$
$$+ \sum_{\ell=1, \ell \neq j, r}^k \frac{\exp\left(c_i \rho \left(\cos(\mathbf{f}(\mathbf{x}_i), \mathbf{g}_\rho(y_\ell)) - \cos(\mathbf{f}(\mathbf{x}_i), \mathbf{g}_\rho(y_j))\right)\right)}{\rho} \tag{65}$$

and see that the first term of Equation (65) goes to zero, the second term goes to infinity Equation (64) and the third term is always positive since $\exp(x)$ is always positive. Therefore we have that

$$d(\rho) \to \infty \quad \text{for} \quad \rho \to \infty \tag{66}$$

which means that

$$2c_i \rho \cdot p(y_j | \mathbf{x}_i) = \frac{2c_i}{d(\rho)} \to 0 \quad \text{for} \quad \rho \to \infty \tag{67}$$

Thus, we get for the first term of Equation (52) that

$$p(y_j | \mathbf{x}_i)(\mathbf{f}(\mathbf{x}_i)^\top \mathbf{g}_\rho(y_j) - \mathbf{f}'(\mathbf{x}_i)^\top \mathbf{g}'_\rho(y_j)) \to 0 \quad \text{for} \quad \rho \to \infty \tag{68}$$

which was the desired result.

We now consider the second term in Equation (52)

$$p(y_j | \mathbf{x}_i) \log \frac{Z'(\mathbf{x}_i)}{Z(\mathbf{x}_i)}. \tag{69}$$

We see that

$$\frac{Z'(\mathbf{x}_i)}{Z(\mathbf{x}_i)} = \frac{\sum_{\ell=1}^k \exp(\cos(\mathbf{f}'(\mathbf{x}_i), \mathbf{g}'_\rho(y_\ell)) \cdot c_i \cdot \rho)}{\sum_{m=1}^k \exp(\cos(\mathbf{f}'(\mathbf{x}_i), \mathbf{g}'_\rho(y_m)) \cdot c_i \cdot \rho)} \tag{70}$$

$$\leq \frac{\sum_{\ell=1}^k \exp(1 \cdot c_i \cdot \rho)}{\sum_{m=1}^k \exp(-1 \cdot c_i \cdot \rho)} \tag{71}$$

$$= \frac{\exp(c_i \cdot \rho)}{\exp(-c_i \cdot \rho)} \tag{72}$$

$$= \exp(2c_i \cdot \rho). \tag{73}$$

Which means we have that

$$p(y_j|\mathbf{x}_i) \log \frac{Z'(\mathbf{x}_i)}{Z(\mathbf{x}_i)} \leq p(y_j|\mathbf{x}_i) \log \exp(2c_i \cdot \rho) = 2c_i \rho \cdot p(y_j|\mathbf{x}_i). \tag{74}$$

We also have that

$$\frac{Z'(\mathbf{x}_i)}{Z(\mathbf{x}_i)} = \frac{\sum_{\ell=1}^{k} \exp(\cos(\mathbf{f}'(\mathbf{x}_i), \mathbf{g}'_\rho(y_\ell)) \cdot c_i \cdot \rho)}{\sum_{m=1}^{k} \exp(\cos(\mathbf{f}'(\mathbf{x}_i), \mathbf{g}'_\rho(y_m)) \cdot c_i \cdot \rho)} \tag{75}$$

$$\geq \frac{\sum_{\ell=1}^{k} \exp(-1 \cdot c_i \cdot \rho)}{\sum_{m=1}^{k} \exp(1 \cdot c_i \cdot \rho)} \tag{76}$$

$$= \frac{\exp(-c_i \cdot \rho)}{\exp(c_i \cdot \rho)} \tag{77}$$

$$= \exp(-2c_i \cdot \rho). \tag{78}$$

Which means we have that

$$p(y_j|\mathbf{x}_i) \log \frac{Z'(\mathbf{x}_i)}{Z(\mathbf{x}_i)} \geq p(y_j|\mathbf{x}_i) \log \exp(-2c_i \cdot \rho) \tag{79}$$

$$= -2c_i \rho \cdot p(y_j|\mathbf{x}_i). \tag{80}$$

which gives us

$$-p(y_j|\mathbf{x}_i) \log \frac{Z'(\mathbf{x}_i)}{Z(\mathbf{x}_i)} \leq 2c_i \rho \cdot p(y_j|\mathbf{x}_i). \tag{81}$$

Now Equation (74) and Equation (81) together give us that

$$\left| p(y_j|\mathbf{x}_i) \log \frac{Z'(\mathbf{x}_i)}{Z(\mathbf{x}_i)} \right| \leq 2c_i \rho \cdot p(y_j|\mathbf{x}_i) \tag{82}$$

Thus, we have upper bounded the absolute value of the term with $2c_i \rho \cdot p(y_j|\mathbf{x}_i)$ like in Equation (58). Using again that $2c_i \rho \cdot p(y_j|\mathbf{x}_i) \to 0$ for $\rho \to \infty$ (Equation (67)), we get that

$$p(y_j|\mathbf{x}_i) \log \frac{Z'(\mathbf{x}_i)}{Z(\mathbf{x}_i)} \to 0 \quad \text{for} \quad \rho \to \infty. \tag{83}$$

Since both terms in the sum (Equation (52)) go to zero for $\rho \to \infty$, the entire sum goes to zero and for $y_j \neq \text{argmax}_{y \in \mathcal{Y}} p(y|\mathbf{x}_i)$, we have that

$$p(y_j|\mathbf{x}_i) \log \frac{p(y_j|\mathbf{x}_i)}{p'(y_j|\mathbf{x}_i)} \to 0 \quad \text{for} \quad \rho \to \infty. \tag{84}$$

Since we now have this result for both $y_j = \text{argmax}_{y \in \mathcal{Y}} p(y|\mathbf{x}_i)$ and $y_j \neq \text{argmax}_{y \in \mathcal{Y}} p(y|\mathbf{x}_i)$, the sum over all $j$ in $D_{\text{KL}}(p(y|\mathbf{x}_i)\|p'(y|\mathbf{x}_i))$ (Equation (39)) also goes to zero. Thus,

$$D_{\text{KL}}(p(y|\mathbf{x}_i)\|p'(y|\mathbf{x}_i)) \to 0 \qquad \forall \mathbf{x}_i \in \mathcal{X}, \tag{85}$$

which means that

$$d_{\text{KL}}(p, p') = \mathbb{E}_{\mathbf{x} \sim p(\mathbf{x})}[D_{\text{KL}}(p(y|\mathbf{x})\|p'(y|\mathbf{x}))] \to 0 \quad \text{for} \quad \rho \to \infty \tag{86}$$

proving the first claim of the theorem.

(2) We now present two constructions which satisfy the requirements i) to iv) above, but where $\mathbf{f}'(\mathbf{x})$ is not an invertible linear transformation of $\mathbf{f}(\mathbf{x})$. In the following, we use $\mathbf{e}_i$ to denote the $i$-th unit vector in $\mathbb{R}^M$, whose $i$-th component is equal to 1, and all other components are equal to zero.

**Construction for $k \geq M + 2$.** For $i = 1, \dots, M$, let $\mathbf{g}_\rho(y_i) = \rho \cdot \mathbf{e}_i$. Let $\mathbf{g}_\rho(y_{M+1}) = -\rho \cdot \mathbf{e}_1$ and $\mathbf{g}_\rho(y_{M+2}) = -\rho \cdot \mathbf{e}_2$. For any remaining labels with index $j > M + 2$, let them be such that $\mathbf{g}_\rho(y_j) \neq \mathbf{g}_\rho(y_\ell)$ for $j \neq \ell$ and more than $\pi/2$ radian angle away from $e_2, -e_1$ and $-e_2$. Let $\{\mathbf{x}_n\}_{n=1}^\infty \subset \mathcal{X}$ and $\{\mathbf{x}_m\}_{m=1}^\infty \subset \mathcal{X}$ be two sets of inputs. We define

$$\mathbf{f}(\mathbf{x}_n) = \begin{pmatrix} \cos(\pi - \frac{\pi}{4}(1 - \frac{1}{n})) \\ \sin(\pi - \frac{\pi}{4}(1 - \frac{1}{n})) \\ 0 \\ \vdots \end{pmatrix} \tag{87}$$

Now $\{\mathbf{f}(\mathbf{x}_n)\}_{n=1}^{\infty}$ is a sequence with a well-defined finite limit, which we for ease of reference shall call $\mathbf{a}$

$$\mathbf{a} = \lim_{n\to\infty} \mathbf{f}(\mathbf{x}_n) = \begin{pmatrix} \cos(\frac{3\pi}{4}) \\ \sin(\frac{3\pi}{4}) \\ 0 \\ \vdots \end{pmatrix} \tag{88}$$

We now define $\mathbf{f}$ for the other set.

$$\mathbf{f}(\mathbf{x}_m) = \begin{pmatrix} \cos(\frac{3\pi}{4} - \frac{\pi}{4m}) \\ \sin(\frac{3\pi}{4} - \frac{\pi}{4m}) \\ 0 \\ \vdots \end{pmatrix} \tag{89}$$

$\{\mathbf{f}(\mathbf{x}_m)\}_{m=1}^{\infty}$ is now a sequence with the same limit, that is,

$$\lim_{m\to\infty} \mathbf{f}(\mathbf{x}_m) = \begin{pmatrix} \cos(\frac{3\pi}{4}) \\ \sin(\frac{3\pi}{4}) \\ 0 \\ \vdots \end{pmatrix} = \mathbf{a} \tag{90}$$

For every remaining $\mathbf{x} \in \mathcal{X}$, construct $\mathbf{f}(\mathbf{x})$ such that there exists a label $y_j \in \mathcal{Y}$ such that we have the cosine similarities $\cos(\mathbf{f}(\mathbf{x}), \mathbf{g}_\rho(y_j)) > 0$ and $\cos(\mathbf{f}(\mathbf{x}), \mathbf{g}_\rho(y_j)) > \cos(\mathbf{f}(\mathbf{x}), \mathbf{g}_\rho(y_\ell))$ for $y_\ell \neq y_j$. With this $(\mathbf{f}, \mathbf{g})$ satisfies i) and ii).

We now construct the second model $(\mathbf{f}', \mathbf{g}')$. Let $\mathbf{g}'_\rho(y_i) = \mathbf{g}_\rho(y_i)$ for $i = 1, \ldots, M$ and $i > M + 2$. Let $\mathbf{g}'_\rho(y_{M+1}) = -\rho \cdot \mathbf{e}_2$ and $\mathbf{g}'_\rho(y_{M+2}) = -\rho \cdot \mathbf{e}_1$. Note that we swapped the unembeddings for $\mathbf{g}'$ of $y_{M+1}$ and $y_{M+2}$ compared to $\mathbf{g}$. For all $\mathbf{x} \in \mathcal{X}$ and the corresponding $y_j = \operatorname{argmax}_{y\in\mathcal{Y}} p(y|\mathbf{x})$ such that $y_j \neq y_{M+1}, y_{M+2}$, let $\mathbf{f}'(\mathbf{x}) = \mathbf{f}(\mathbf{x})$. However, for all $\mathbf{x} \in \mathcal{X}$ and the corresponding $y_j = \operatorname{argmax}_{y\in\mathcal{Y}} p(y|\mathbf{x})$ such that $y_j = M + 1$, let $\|\mathbf{f}'(\mathbf{x})\| = \|\mathbf{f}(\mathbf{x})\|$ and let $\mathbf{f}'(\mathbf{x})$ be rotated counterclockwise with respect to the first two axes (i.e., the first two components of the representations) such that $\cos(\mathbf{f}'(\mathbf{x}), \mathbf{g}'_\rho(y_j)) = \cos(\mathbf{f}(\mathbf{x}), \mathbf{g}_\rho(y_j))$. Also, for all $\mathbf{x}_i \in \mathcal{X}$ and the corresponding $y_j = \operatorname{argmax}_{y\in\mathcal{Y}} p(y|\mathbf{x}_i)$ such that $y_j = M + 2$, let $\|\mathbf{f}'(\mathbf{x}_i)\| = \|\mathbf{f}(\mathbf{x}_i)\|$ and let $\mathbf{f}'(\mathbf{x}_i)$ be rotated clockwise with respect to the first two axes such that $\cos(\mathbf{f}'(\mathbf{x}_i), \mathbf{g}'_\rho(y_j)) = \cos(\mathbf{f}(\mathbf{x}_i), \mathbf{g}_\rho(y_j))$. This construction satisfies the requirements i) to iv).

We will show that $\mathbf{f}'(\mathbf{x})$ is not an invertible linear transformation of $\mathbf{f}(\mathbf{x})$ by showing that any function $\boldsymbol{\tau} : \mathbb{R}^M \to \mathbb{R}^M$ with $\boldsymbol{\tau}(\mathbf{f}(\mathbf{x})) = \mathbf{f}'(\mathbf{x}), \forall \mathbf{x} \in \mathcal{X}$ cannot be continuous.

Let $\boldsymbol{\tau} : \mathbb{R}^M \to \mathbb{R}^M$ be such that $\boldsymbol{\tau}(\mathbf{f}(\mathbf{x})) = \mathbf{f}'(\mathbf{x}), \forall \mathbf{x} \in \mathcal{X}$. We recall that a function $\mathbf{h} : \mathbb{R}^M \to \mathbb{R}^M$ is by definition continuous at $\mathbf{c} \in \mathbb{R}^M$ if

$$\mathbf{h}(\mathbf{x}) \to \mathbf{h}(\mathbf{c}) \quad \text{for} \quad \mathbf{x} \to \mathbf{c} \tag{91}$$

Therefore, we consider the limits of the sequences $\{\boldsymbol{\tau}(\mathbf{f}(\mathbf{x}_n))\}_{n=1}^{\infty}$ and $\{\boldsymbol{\tau}(\mathbf{f}(\mathbf{x}_m))\}_{m=1}^{\infty}$. We first note that by our construction, we have for the rotation matrix

$$\mathbf{R} = \begin{bmatrix} \cos(\frac{\pi}{2}) & -\sin(\frac{\pi}{2}) & 0 & \cdots \\ \sin(\frac{\pi}{2}) & \cos(\frac{\pi}{2}) & 0 & \cdots \\ 0 & 0 & 1 & \cdots \\ \vdots & \vdots & \vdots & \ddots \end{bmatrix} \tag{92}$$

that

$$\mathbf{f}'(\mathbf{x}_n) = \mathbf{R}\mathbf{f}(\mathbf{x}_n) \tag{93}$$

Which means that since $\boldsymbol{\tau}(\mathbf{f}(\mathbf{x}_n)) = \mathbf{f}'(\mathbf{x}_n)$, we have that

$$\boldsymbol{\tau}(\mathbf{f}(\mathbf{x}_n)) = \mathbf{R}\mathbf{f}(\mathbf{x}_n) = \begin{pmatrix} \cos(\frac{3\pi}{2} - \frac{\pi}{4}(1 - \frac{1}{n})) \\ \sin(\frac{3\pi}{2} - \frac{\pi}{4}(1 - \frac{1}{n})) \\ 0 \\ \vdots \end{pmatrix} \tag{94}$$

and we see that when $n \to \infty$,

$$\tau(\mathbf{f}(\mathbf{x}_n)) \to \mathbf{b} := \begin{pmatrix} \cos(\frac{3\pi}{2}) \\ \sin(\frac{3\pi}{2}) \\ 0 \\ \vdots \end{pmatrix} \quad \text{for} \quad \mathbf{f}(\mathbf{x}_n) \to \mathbf{a} \tag{95}$$

However, we also have that $\tau(\mathbf{f}(\mathbf{x}_m)) = \mathbf{f}'(\mathbf{x}_m) = \mathbf{f}(\mathbf{x}_m)$. This means that for $m \to \infty$, we have

$$\tau(\mathbf{f}(\mathbf{x}_m)) \to \mathbf{a} \quad \text{for} \quad \mathbf{f}(\mathbf{x}_m) \to \mathbf{a} \tag{96}$$

Now since for $\mathbf{b}$ from Equation (95), we have $\mathbf{b} \neq \mathbf{a}$, $\tau$ cannot be continuous. In particular, it cannot be an invertible linear transformation, and thus $\mathbf{f}'(\mathbf{x})$ is not an invertible linear transformation of $\mathbf{f}(\mathbf{x})$.

Note that if either $\mathbf{f}$ or $\mathbf{f}'$ is not injective, then both $\mathbf{f}$ and $\mathbf{f}'$ can be smooth. However, if both $\mathbf{f}$ and $\mathbf{f}'$ are injective, then either $\mathbf{f}$ or $\mathbf{f}'$ has to be non-continuous.

Notice also that we can permute additional labels to bring the embeddings further from a linear transformation. See for example Fig. 2 which has been constructed by permuting multiple labels.

**Construction for $k = M + 1$.** For $i = 1, \dots M$, let $\mathbf{g}_\rho(y_i) = \rho \cdot \mathbf{e}_i$ and $\mathbf{g}_\rho(y_{M+1}) = -\rho \cdot \mathbf{e}_1$. Let $\{\mathbf{x}_n\}_{n=1}^\infty \subset \mathcal{X}$ and $\{\mathbf{x}_m\}_{m=1}^\infty \subset \mathcal{X}$ be two sets of inputs. We define

$$\mathbf{f}(\mathbf{x}_n) = \begin{pmatrix} \cos(\pi - \frac{\pi}{4}(1 - \frac{1}{n})) \\ \sin(\pi - \frac{\pi}{4}(1 - \frac{1}{n})) \\ 0 \\ \vdots \end{pmatrix} \tag{97}$$

Now $\{\mathbf{f}(\mathbf{x}_n)\}_{n=1}^\infty$ is a sequence with a well-defined finite limit, which we for ease of reference shall call $\mathbf{a}$

$$\mathbf{a} = \lim_{n \to \infty} \mathbf{f}(\mathbf{x}_n) = \begin{pmatrix} \cos(\frac{3\pi}{4}) \\ \sin(\frac{3\pi}{4}) \\ 0 \\ \vdots \end{pmatrix} \tag{98}$$

We now define $\mathbf{f}$ for the other set.

$$\mathbf{f}(\mathbf{x}_m) = \begin{pmatrix} \cos(\frac{3\pi}{4} - \frac{\pi}{4m}) \\ \sin(\frac{3\pi}{4} - \frac{\pi}{4m}) \\ 0 \\ \vdots \end{pmatrix} \tag{99}$$

$\{\mathbf{f}(\mathbf{x}_m)\}_{m=1}^\infty$ is now a sequence with the same limit, that is,

$$\lim_{m \to \infty} \mathbf{f}(\mathbf{x}_m) = \begin{pmatrix} \cos(\frac{3\pi}{4}) \\ \sin(\frac{3\pi}{4}) \\ 0 \\ \vdots \end{pmatrix} = \mathbf{a} \tag{100}$$

For every remaining $\mathbf{x}_i \in \mathcal{X}$, construct $\mathbf{f}(\mathbf{x}_i)$ such that there exists a label $y_j \in \mathcal{Y}$ such that we have the cosine similarities $\cos(\mathbf{f}(\mathbf{x}_i), \mathbf{g}_\rho(y_j)) > 0$ and $\cos(\mathbf{f}(\mathbf{x}_i), \mathbf{g}_\rho(y_j)) > \cos(\mathbf{f}(\mathbf{x}_i), \mathbf{g}_\rho(y_\ell))$ for $y_\ell \neq y_j$. With this $(\mathbf{f}, \mathbf{g})$ satisfies i) and ii).

We now construct the second model $(\mathbf{f}', \mathbf{g}')$. Let $\mathbf{g}'_\rho(y_i) = \mathbf{g}_\rho(y_i)$ for $i = 1, \dots, M$ and let $\mathbf{g}'_\rho(y_{M+1}) = -\rho \cdot \mathbf{e}_2$. For all $\mathbf{x}_i \in \mathcal{X}$ and the corresponding $y_j = \operatorname{argmax}_{y \in \mathcal{Y}} p(y|\mathbf{x}_i)$ such that $y_j \neq y_{M+1}$, let $\mathbf{f}'(\mathbf{x}_i) = \mathbf{f}(\mathbf{x}_i)$. For all $\mathbf{x}_i \in \mathcal{X}$ and the corresponding $y_j = \operatorname{argmax}_{y \in \mathcal{Y}} p(y|\mathbf{x}_i)$ such that $y_j = y_{M+1}$, construct $\mathbf{f}'(\mathbf{x}_i)$ from $\mathbf{f}(\mathbf{x}_i)$ by rotating it counterclockwise with respect to the first two axes such that $\cos(\mathbf{f}'(\mathbf{x}_i), \mathbf{g}'_\rho(y_j)) = \cos(\mathbf{f}(\mathbf{x}_i), \mathbf{g}_\rho(y_j))$. This construction satisfies the requirements i) to iv) and by the same argument as used for the case with $k \geq M + 2$, $\mathbf{f}'(\mathbf{x})$ is not an invertible linear transformation of $\mathbf{f}(\mathbf{x})$.

$\square$

## C.2 Loss goes to Zero but Representations are Dissimilar

We denote the negative log-likelihood for a model $p$ with respect to a data distribution $p_\mathcal{D}$ as:

$$\text{NLL}_{p_\mathcal{D}}(p) := \mathbb{E}_{(\mathbf{x},y)\sim p_\mathcal{D}}[-\log p(y \mid \mathbf{x})]. \tag{101}$$

In the following, it is useful to establish the link between the negative log-likelihood and the $D_{\text{KL}}$ divergence with the empirical distribution. To this end, we introduce the empirical distribution over inputs and outputs $p_\mathcal{D}(\mathbf{x}, y)$, whose marginal on the input is given by $p_\mathcal{D}(\mathbf{x})$. This can be rewritten as:

$$p_\mathcal{D}(\mathbf{x}, y) = p_\mathcal{D}(y \mid \mathbf{x})p_\mathcal{D}(\mathbf{x}). \tag{102}$$

Notice that the conditional distribution underlies how labels are associated with the input. We can connect the negative log-likelihood to the KL divergence with the empirical distribution, a well-known fact in the literature, as shown in the next Lemma:

**Lemma C.1.** *Let $p_\mathcal{D}(\mathbf{x}, y) = p_\mathcal{D}(y|\mathbf{x})p_\mathcal{D}(\mathbf{x})$ be the ground-truth data distribution. Let $p(y|\mathbf{x})$ be a model such that, for all $\mathbf{x}$ in the support of $p_\mathcal{D}(\mathbf{x})$, we have $p_\mathcal{D}(y \mid \mathbf{x}) \ll p(y \mid \mathbf{x})$.[15] Define the negative log-likelihood as $\text{NLL}_{p_\mathcal{D}}(p) := \mathbb{E}_{(\mathbf{x},y)\sim p_\mathcal{D}}[-\log p(y \mid \mathbf{x})]$. Then:*

$$\text{NLL}_{p_\mathcal{D}}(p) = d_{\text{KL}}(p_\mathcal{D}, p) + \mathbb{E}_{\mathbf{x}\sim p_\mathcal{D}}[H(p_\mathcal{D}(y \mid \mathbf{x}))], \tag{103}$$

*where $H(q) := -\mathbb{E}_{x\sim q}[\log q(x)]$ denotes the entropy of a distribution $q$.*

*Proof.* From the definition of KL divergence and entropy:

$$D_{\text{KL}}(p_\mathcal{D}(y \mid \mathbf{x})\|p(y \mid \mathbf{x})) = \mathbb{E}_{y\sim p_\mathcal{D}(y|\mathbf{x})}[\log p_\mathcal{D}(y \mid \mathbf{x}) - \log p(y \mid \mathbf{x})] \tag{104}$$

$$= -H(p_\mathcal{D}(y \mid \mathbf{x})) + \mathbb{E}_{y\sim p_\mathcal{D}(y|\mathbf{x})}[-\log p(y \mid \mathbf{x})]. \tag{105}$$

Moving the entropy to the other side and taking expectations w.r.t. $p_\mathcal{D}(\mathbf{x})$, we get:

$$\mathbb{E}_{(\mathbf{x},y)\sim p_\mathcal{D}}[-\log p(y \mid \mathbf{x})] = \mathbb{E}_{\mathbf{x}\sim p_\mathcal{D}}[D_{\text{KL}}(p_\mathcal{D}(y \mid \mathbf{x})\|p(y \mid \mathbf{x}))] + \mathbb{E}_{\mathbf{x}\sim p_\mathcal{D}}[H(p_\mathcal{D}(y \mid \mathbf{x}))] \tag{106}$$

The statement follows by the definition of $d_{\text{KL}}$ (Equation (6)) and NLL. $\qquad\square$

Notice that, when the conditional distribution $p_\mathcal{D}(y \mid \mathbf{x})$ is degenerate, i.e., to each $\mathbf{x} \in \mathcal{X}$ a single $y \in \mathcal{Y}$ is associated, the expectation of the conditional Shannon entropy vanishes.

Next, we restate Corollary 3.2 and prove in full details:

**Corollary** (Formal version of Corollary 3.2). *Assume we have a data distribution where only one label is associated with each unique input. Let $(\mathbf{f}, \mathbf{g}) \in \Theta$ be a model with representations in $\mathbb{R}^M$. Assume $M \geq 2$. Let $k := |\mathcal{Y}|$ be the total number of unembeddings and assume $k \geq M + 1$. Assume the model satisfies the following requirements:*

- *i) The unembeddings have fixed norm, that is $\|\mathbf{g}(y_i)\| = \|\mathbf{g}(y_j)\| = \rho$, for all $y_i, y_j \in \mathcal{Y}$, where $\rho \in \mathbb{R}^+$.*

- *ii) For every $\mathbf{x}_i \in \mathcal{X}$, let there be one element $y_j \in \mathcal{Y}$ for which the cosine similarity $\cos(\mathbf{f}(\mathbf{x}_i), \mathbf{g}(y_j)) > 0$ and $\cos(\mathbf{f}(\mathbf{x}_i), \mathbf{g}(y_j)) > \cos(\mathbf{f}(\mathbf{x}_i), \mathbf{g}(y_\ell))$ for all $y_\ell \neq y_j$. Also, assume $y_j$ is the only label associated with $\mathbf{x}_i$.*

*Let $(\mathbf{f}', \mathbf{g}')$ be a model which also satisfies i) and ii) and which is related to $(\mathbf{f}, \mathbf{g})$ in the following way:*

- *iii) For every $\mathbf{x}_i \in \mathcal{X}$, $\|\mathbf{f}'(\mathbf{x}_i)\| = \|\mathbf{f}(\mathbf{x}_i)\|$.*

- *iv) For $\hat{y} = \arg\max_{y\in\mathcal{Y}}(p(y|\mathbf{x}_i))$ we have that the angle between $\mathbf{f}'(\mathbf{x}_i)$ and $\mathbf{g}'(\hat{y})$ is equal to the angle between $\mathbf{f}(\mathbf{x}_i)$ and $\mathbf{g}(\hat{y})$, in particular, $\cos(\mathbf{f}(\mathbf{x}_i), \mathbf{g}(\hat{y})) = \cos(\mathbf{f}'(\mathbf{x}_i), \mathbf{g}'(\hat{y}))$.*

*Then, making a sequence of pairs of models indexed by $\rho$, we have*

$$\text{NLL}(p_{\mathbf{f},\mathbf{g}}) \to 0 \quad \text{for} \quad \rho \to \infty \tag{107}$$

$$\text{and}$$

$$\text{NLL}(p_{\mathbf{f}',\mathbf{g}'}) \to 0 \quad \text{for} \quad \rho \to \infty. \tag{108}$$

---

[15] Here $\ll$ denotes absolute continuity.

*Proof.* Below, we use the shorthand $p := p_{\mathbf{f},\mathbf{g}}$ and $p' := p_{\mathbf{f}',\mathbf{g}'}$. We make use of the same calculations as for the proof of Theorem 3.1. We consider $p(y_j|\mathbf{x}_i)$ for $y_j$ the label assigned to $\mathbf{x}_i$ according to the data. We denote with $c_i \in \mathbb{R}^+$ the value of the norm of the embeddings on $\mathbf{x}_i$, i.e., $\|\mathbf{f}(\mathbf{x}_i)\| = \|\mathbf{f}'(\mathbf{x}_i)\| = c_i$. In this case we have that

$$p(y_j|\mathbf{x}_i) = \frac{\exp(\mathbf{f}(\mathbf{x}_i)^\top \mathbf{g}(y_j))}{\sum_{\ell=1}^k \exp(\mathbf{f}(\mathbf{x}_i)^\top \mathbf{g}(y_\ell))} \tag{109}$$

$$= \frac{1}{1 + \sum_{\ell=1,\ell\neq j}^k \exp(\mathbf{f}(\mathbf{x}_i)^\top \mathbf{g}(y_\ell) - \mathbf{f}(\mathbf{x}_i)^\top \mathbf{g}(y_j))} \tag{110}$$

By construction, $\|\mathbf{f}(\mathbf{x}_i)\| = c_i$. We use that

$$\mathbf{f}(\mathbf{x}_i)^\top \mathbf{g}(y_j) = \cos(\mathbf{f}(\mathbf{x}_i), \mathbf{g}(y_j)) \cdot c_i \cdot \rho \tag{111}$$

and see that

$$p(y_j|\mathbf{x}_i) = \left(1 + \sum_{\ell=1,\ell\neq j}^k \exp(c_i \cdot \rho \cdot (\cos(\mathbf{f}(\mathbf{x}_i), \mathbf{g}(y_\ell)) - \cos(\mathbf{f}(\mathbf{x}_i), \mathbf{g}(y_j))))\right)^{-1} \tag{112}$$

Now since by construction, $\cos(\mathbf{f}(\mathbf{x}_i), \mathbf{g}(y_j)) > 0$, and for all $\ell \neq j$

$$\cos(\mathbf{f}(\mathbf{x}_i), \mathbf{g}(y_j)) > \cos(\mathbf{f}(\mathbf{x}_i), \mathbf{g}(y_\ell)) \tag{113}$$

we have for all $\ell$ that

$$\cos(\mathbf{f}(\mathbf{x}_i), \mathbf{g}(y_\ell)) - \cos(\mathbf{f}(\mathbf{x}_i), \mathbf{g}(y_j)) < 0 \tag{114}$$

and

$$\exp(c_i \cdot \rho \cdot (\cos(\mathbf{f}(\mathbf{x}_i), \mathbf{g}(y_\ell)) - \cos(\mathbf{f}(\mathbf{x}_i), \mathbf{g}(y_j)))) \to 0 \quad \text{for} \quad \rho \to \infty \tag{115}$$

which means that combined with Equation (112) we get

$$p(y_j|\mathbf{x}_i) \to \frac{1}{1+0} = 1 \quad \text{for} \quad \rho \to \infty \tag{116}$$

Similarly, we have that

$$p'(y_j|\mathbf{x}_i) \to 1 \quad \text{for} \quad \rho \to \infty. \tag{117}$$

Therefore, we have for the NLL:

$$\text{NLL}(p) = -\sum_{i=1}^n \log p(y_j|\mathbf{x}_i) \to 0 \quad \text{for} \quad \rho \to \infty \tag{118}$$

and

$$\text{NLL}(p') = -\sum_{i=1}^n \log p'(y_j|\mathbf{x}_i) \to 0 \quad \text{for} \quad \rho \to \infty. \tag{119}$$

$\square$

# D  Partial Least Squares (PLS) and Canonical Correlation Analysis (CCA)

Partial least squares (PLS) algorithms are techniques that, for two random variables $\mathbf{z}$ and $\mathbf{w}$, derive the cross-covariance matrix $\Sigma_{\mathbf{zw}}$ using latent variables. We give here a short description of the two variants that are relevant to our work. A more detailed description can be found in [66].

In our case, we will work with random variables of the same dimension. So in the following, we consider $\mathbf{z}, \mathbf{w} \in \mathbb{R}^M$. We will mostly work with the centered and normalized versions of the random variables. So if $z_i$ is the $i$'th dimension of the original random variable, we will consider

$$z_i' = \frac{z_i - \mathbb{E}[z_i]}{\sqrt{\mathrm{Var}[z_i]}} . \tag{120}$$

## D.1  PLS-SVD

PLS-SVD [54, 60] is a variant of PLS where the latent vectors are simply the left and right singular vectors in the singular value decomposition of the cross-covariance matrix, $\Sigma_{\mathbf{zw}}$.

In Algorithm 1, we report the algorithm in the case where we have $n$ samples of our $M$-dimensional random variables. In this case, we work with $(n \times M)$ matrices $\mathbf{Z}, \mathbf{W}$, where each row is a sample. The aim of the algorithm is to get projection vectors $\mathbf{u}^{(r)}, \mathbf{v}^{(r)}$ such that $\mathrm{Cov}_{\mathbf{ZW}}[\mathbf{Zu}^{(r)}, \mathbf{Wv}^{(r)}]$ is as large as possible under the constraint that $\|\mathbf{u}\| = \|\mathbf{v}\| = 1$. The original algorithm includes a choice of signs for the covariances. In this version of the algorithm, we have chosen all signs to be positive. $R > 0$ denotes the maximal rank of the algorithm.

---
**Algorithm 1** Iterative SVD Projection Extraction from Cross-Covariance Matrix
---
1: Set $r \leftarrow 1$
2: Center and scale $\mathbf{Z}$ and $\mathbf{W}$
3: Compute cross-covariance matrix: $\mathbf{C} \leftarrow \mathbf{Z}^\top \mathbf{W}$
4: Set $\mathbf{C}^{(1)} \leftarrow \mathbf{C}$
5: **while** $\mathbf{C}^{(r)} \neq 0$ and $r \leq R$ **do**
6:     Perform SVD: $\mathbf{C}^{(r)} = \mathbf{UDV}^\top$
7:     Extract leading singular vectors: $\mathbf{u}_1^{(r)} \leftarrow$ first column of $\mathbf{U}$, $\mathbf{v}_1^{(r)} \leftarrow$ first column of $\mathbf{V}$
8:     Save $\mathbf{u}_1^{(r)}$ and $\mathbf{v}_1^{(r)}$ as the $r$-th projection vectors
9:     Let $\sigma_r \leftarrow$ leading singular value of $\mathbf{C}^{(r)}$
10:     Update matrix: $\mathbf{C}^{(r+1)} \leftarrow \mathbf{C}^{(r)} - \sigma_r \mathbf{u}_1^{(r)} \mathbf{v}_1^{(r)\top}$
11:     $r \leftarrow r + 1$
---

Considering this algorithm, we see that the projection vectors we get, $\mathbf{u}^{(r)}, \mathbf{v}^{(r)}$, are exactly the first $m$ singular vectors of $\mathbf{Z}^\top \mathbf{W}$ corresponding to the largest $m$ singular values, where $m$ is the value of $r$ at the exit point. Note that this relates to principal component analysis (PCA) in the sense that when doing PCA, one uses the singular value decomposition of the covariance matrix to obtain components that capture the variance in a dataset (a random variable). When doing PLS-SVD, one uses singular value decomposition of the cross-covariance matrix to obtain components that capture the covariance between two random variables.

## D.2  Canonical Correlation Analysis

Canonical Correlation Analysis (CCA) [22] seeks pairs of unit-length vectors $\mathbf{u}_\ell \in \mathbb{R}^{d_Z}$ and $\mathbf{v}_\ell \in \mathbb{R}^{d_W}$ ($\ell = 1, \dots, r$) that maximize the correlation between the one-dimensional projections $\mathbf{u}_\ell^\top \mathbf{z}$ and $\mathbf{v}_\ell^\top \mathbf{w}$, subject to mutual orthogonality of successive directions. In other words, CCA finds the singular value decomposition of the matrix $\mathbf{Q} = \Sigma_{\mathbf{zz}}^{-\frac{1}{2}} \Sigma_{\mathbf{zw}} \Sigma_{\mathbf{ww}}^{-\frac{1}{2}}$, where $\Sigma_{\mathbf{zw}}$ is a cross-covariance matrix and $\Sigma_{\mathbf{zz}}, \Sigma_{\mathbf{ww}}$ are covariance matrices. Equivalently, $\mathbf{Q}$ is the cross-covariance of the whitened variables $\Sigma_{\mathbf{zz}}^{-\frac{1}{2}} \mathbf{z}$ and $\Sigma_{\mathbf{ww}}^{-\frac{1}{2}} \mathbf{w}$, which have identity covariance. Whitening removes the influence of marginal variances, so the eigenvalues found by SVD are the canonical correlations. The results of CCA and PLS-SVD will differ if there are high covariances between the $z_i$'s or the $w_i$'s.

We can use CCA to define a similarity measure between random variables $\mathbf{z}$ and $\mathbf{w}$, known as the mean canonical correlation $m_{\mathrm{CCA}}(\mathbf{z}, \mathbf{w})$, which is the mean of the maxima correlations for the

centered and rescaled random variables. Because of these centering and rescaling operations, $m_{\mathrm{CCA}}$ will be invariant to any invertible linear transformation.

# E   When Closeness in Distribution Implies Representational Similarity - Details

We here include all the technical details of Section 4.

## E.1   The Connection Between Representations and Log-Likelihoods

Let $(\mathbf{f}, \mathbf{g}) \in \Theta$ be a model satisfying the diversity condition (Definition 2.1) and consider $\mathbf{x}_1, \ldots, \mathbf{x}_M \in \mathcal{X}$ and $y_1, ..., y_M \in \mathcal{Y}$ such that both the vectors $\{\mathbf{f}_0(\mathbf{x})\}_{i=1}^{M}$ and $\{\mathbf{g}_0(y_i)\}_{i=1}^{M}$ are linearly independent. We recall some notation for use in the Lemma. We denote with $\mathbf{L}$ and $\mathbf{N}$ the following matrices:

$$\mathbf{L} := \big(\mathbf{g}_0(y_1), \ldots, \mathbf{g}_0(y_M)\big), \quad \mathbf{N} := \big(\mathbf{f}_0(\mathbf{x}_1), \ldots, \mathbf{f}_0(\mathbf{x}_M)\big). \tag{121}$$

We recall the definition of the following functions:

$$\psi_{\mathbf{x}}(y_i; p) := \sqrt{\mathrm{Var}_{\mathbf{x}}[\log p(y_i|\mathbf{x}) - \log p(y_0|\mathbf{x})]} \ \text{ and } \ \psi_y(\mathbf{x}_j; p) := \sqrt{\mathrm{Var}_y[\log p(y|\mathbf{x}_j) - \log p(y|\mathbf{x}_0)]} \ . \tag{122}$$

We also denote with $\mathbf{S}, \mathbf{S}', \mathbf{D}, \mathbf{D}' \in \mathbb{R}^{M \times M}$ the diagonal matrices with entries $S_{ii} := \frac{1}{\psi_{\mathbf{x}}(y_i; p)}, D_{ii} := \frac{1}{\psi_y(\mathbf{x}_j; p)}, S'_{ii} := \frac{1}{\psi_{\mathbf{x}}(y_i; p')}, D'_{ii} := \frac{1}{\psi_y(\mathbf{x}_j; p')}$. We now have everything we need to state a Lemma relating any two model representations.

Here, we provide the full statement for Lemma 4.1 and prove it:

**Lemma** (Complete version of Lemma 4.1). *For any $(\mathbf{f}, \mathbf{g}), (\mathbf{f}', \mathbf{g}') \in \Theta$ satisfying the diversity condition (Definition 2.1). Let $\tilde{\mathbf{A}} := \mathbf{L}^{-\top}\mathbf{S}^{-1}\mathbf{S}'\mathbf{L}'^{\top}$ and $\mathbf{h_f}(\mathbf{x}) := \mathbf{L}^{-\top}\mathbf{S}^{-1}\boldsymbol{\epsilon}_y(\mathbf{x})$, where the $i$-th entry of $\boldsymbol{\epsilon}_y(\mathbf{x})$ is given by*

$$\epsilon_{y_i}(\mathbf{x}) = \frac{\log p_{\mathbf{f}, \mathbf{g}}(y_i|\mathbf{x}) - \log p_{\mathbf{f}, \mathbf{g}}(y_0|\mathbf{x})}{\psi_{\mathbf{x}}(y_i; p_{\mathbf{f}, \mathbf{g}})} - \frac{\log p_{\mathbf{f}', \mathbf{g}'}(y_i|\mathbf{x}) - \log p_{\mathbf{f}', \mathbf{g}'}(y_0|\mathbf{x})}{\psi_{\mathbf{x}}(y_i; p_{\mathbf{f}', \mathbf{g}'})} \ .$$

*Let $\mathbf{B} := \mathbf{N}^{-\top}\mathbf{D}^{-1}\mathbf{D}'\mathbf{N}'^{\top}$ and $\mathbf{h_g}(y) := \mathbf{N}^{-\top}\mathbf{D}^{-1}\boldsymbol{\epsilon}_{\mathbf{x}}(y)$, where the $j$-th entry of $\boldsymbol{\epsilon}_{\mathbf{x}}(\mathbf{y})$ is given by*

$$\epsilon_{\mathbf{x}_j}(y) = \frac{\log p_{\mathbf{f}, \mathbf{g}}(y|\mathbf{x}_j) - \log p_{\mathbf{f}, \mathbf{g}}(y|\mathbf{x}_0)}{\psi_y(\mathbf{x}_j; p_{\mathbf{f}, \mathbf{g}})} - \frac{\log p_{\mathbf{f}', \mathbf{g}'}(y|\mathbf{x}_j) - \log p_{\mathbf{f}', \mathbf{g}'}(y|\mathbf{x}_0)}{\psi_y(\mathbf{x}_j; p_{\mathbf{f}', \mathbf{g}'})} \ .$$

*Then we have:*

$$\mathbf{f}(\mathbf{x}) = \tilde{\mathbf{A}}\mathbf{f}'(\mathbf{x}) + \mathbf{h_f}(\mathbf{x}), \quad \mathbf{g}(y) = \mathbf{B}\mathbf{g}'(y) + \mathbf{h_g}(y). \tag{123}$$

*Proof.* Using the definition of $\boldsymbol{\epsilon}(\mathbf{x})$, we can relate the embeddings of the two models as follows:

$$\mathbf{S}\mathbf{L}\mathbf{f}(\mathbf{x}) = \mathbf{S}'\mathbf{L}'\mathbf{f}'(\mathbf{x}) + \mathbf{S}\mathbf{L}\mathbf{f}(\mathbf{x}) - \mathbf{S}'\mathbf{L}'\mathbf{f}'(\mathbf{x}) \tag{124}$$

$$= \mathbf{S}'\mathbf{L}'\mathbf{f}'(\mathbf{x}) + \boldsymbol{\epsilon}_y(\mathbf{x}). \tag{125}$$

Therefore, multiplying by the inverse of $\mathbf{S}$ and of $\mathbf{L}$, we get

$$\mathbf{f}(\mathbf{x}) = \mathbf{L}^{-\top}\mathbf{S}^{-1}\mathbf{S}'\mathbf{L}'^{\top}\mathbf{f}'(\mathbf{x}) + \mathbf{L}^{-\top}\mathbf{S}^{-1}\boldsymbol{\epsilon}_y(\mathbf{x}) \tag{126}$$

$$= \tilde{\mathbf{A}}\mathbf{f}'(\mathbf{x}) + \mathbf{h_f}(\mathbf{x}). \tag{127}$$

With similar steps, we obtain the relation between the unembeddings of the two models, proving Equation (123). $\qquad\square$

Notice that the error term $\boldsymbol{\epsilon}_y(\mathbf{x})$ is a function of $\mathbf{x}$ and comprises the $y_i$'s from the diversity condition (Definition 2.1). Also, $\mathbf{B}$ results from the product of the scaling matrices and those constructed from the diversity condition applied to $\mathbf{f}$. $\mathbf{h_g}(y)$ depends on $\boldsymbol{\epsilon}_{\mathbf{x}}(y)$, which is a function of $y$ using the $\mathbf{x}_i$'s from the diversity condition.

The following corollary connects Lemma 4.1 to the identifiability results of Theorem 2.2:

**Corollary E.1.** *Under the same assumptions of [Lemma 4.1](#), if we assume that the distributions of the models $(\mathbf{f}, \mathbf{g}), (\mathbf{f}', \mathbf{g}') \in \Theta$ are equal, then we get (as in [Theorem 2.2](#)):*

$$\mathbf{f}(\mathbf{x}) = \mathbf{A}\mathbf{f}'(\mathbf{x}) \tag{128}$$

$$\mathbf{g}_0(y) = \mathbf{A}^{-\top}\mathbf{g}_0'(y), \tag{129}$$

*where $\mathbf{A} = \mathbf{L}^{-\top}\mathbf{L}'^{\top}$.*

*Proof.* Below, we use the shorthand $p := p_{\mathbf{f},\mathbf{g}}$ and $p' := p_{\mathbf{f}',\mathbf{g}'}$.

When the two models entail the same distribution, we get that

$$\psi_{\mathbf{x}}(y_i; p) = \sqrt{\operatorname{Var}_{\mathbf{x}}[\log p(y_i|\mathbf{x}) - \log p(y_0|\mathbf{x})]} = \psi_{\mathbf{x}}(y_i; p'). \tag{130}$$

which means that

$$
\begin{aligned}
\epsilon_{y_i}(\mathbf{x}) &= \frac{\mathbf{f}(\mathbf{x})^{\top}\mathbf{g}_0(y_i)}{\sqrt{\operatorname{Var}_{\mathbf{x}}[\mathbf{L}_i^{\top}\mathbf{f}(\mathbf{x})]}} - \frac{\mathbf{f}'(\mathbf{x})^{\top}\mathbf{g}_0'(y_i)}{\sqrt{\operatorname{Var}_{\mathbf{x}}[\mathbf{L}_i'^{\top}\mathbf{f}'(\mathbf{x})]}} \\
&= \frac{\log p_{\mathbf{f},\mathbf{g}}(y_i|\mathbf{x}) - \log p_{\mathbf{f},\mathbf{g}}(y_0|\mathbf{x})}{\psi_{\mathbf{x}}(y_i; p_{\mathbf{f},\mathbf{g}})} - \frac{\log p_{\mathbf{f}',\mathbf{g}'}(y_i|\mathbf{x}) - \log p_{\mathbf{f}',\mathbf{g}'}(y_0|\mathbf{x})}{\psi_{\mathbf{x}}(y_i; p_{\mathbf{f}',\mathbf{g}'})} \\
&= \frac{1}{\psi_{\mathbf{x}}(y_i; p_{\mathbf{f},\mathbf{g}})}(\log p_{\mathbf{f},\mathbf{g}}(y_i|\mathbf{x}) - \log p_{\mathbf{f}',\mathbf{g}'}(y_i|\mathbf{x}) + \log p_{\mathbf{f}',\mathbf{g}'}(y_0|\mathbf{x}) - \log p_{\mathbf{f},\mathbf{g}}(y_0|\mathbf{x})) \\
&= 0.
\end{aligned}
$$

Also, since

$$\sqrt{\operatorname{Var}_{\mathbf{x}}[\mathbf{L}_i^{\top}\mathbf{f}(\mathbf{x})]} = \sqrt{\operatorname{Var}_{\mathbf{x}}[\log p(y_i|\mathbf{x}) - \log p(y_0|\mathbf{x})]} \tag{131}$$

$$= \sqrt{\operatorname{Var}_{\mathbf{x}}[\log p'(y_i|\mathbf{x}) - \log p'(y_0|\mathbf{x})]} \tag{132}$$

$$= \sqrt{\operatorname{Var}_{\mathbf{x}}[\mathbf{L}_i'^{\top}\mathbf{f}'(\mathbf{x})]} \tag{133}$$

and $\mathbf{S}$ is a diagonal matrix, we have that

$$\mathbf{S}^{-1}\mathbf{S}' = \mathbf{S}^{-1}\mathbf{S} = \mathbf{I}. \tag{134}$$

Finally, we get

$$\mathbf{A} = \mathbf{L}^{-\top}\mathbf{S}^{-1}\mathbf{S}'\mathbf{L}'^{\top} = \mathbf{L}^{-\top}\mathbf{L}'^{\top}. \tag{135}$$

and since the error term vanishes, we obtain:

$$\mathbf{f}(\mathbf{x}) = \mathbf{A}\mathbf{f}'(\mathbf{x}), \tag{136}$$

showing the identifiability of the embeddings. The result for the unembeddings proceeds in a similar way, by noticing that also $\epsilon_{\mathbf{x}}$ is zero, therefore getting the same result of [Theorem 2.2](#). $\square$

The result of [Lemma 4.1](#) also gives us the following corollary, showing a connection between the distributions and the representations:

**Corollary E.2.** *Under the same assumptions as in [Lemma 4.1](#), we have:*

$$\operatorname{Var}_{\mathbf{x}}[\epsilon_{y_i}(\mathbf{x})] = 2(1 - \operatorname{Corr}[\mathbf{L}_i\mathbf{f}(\mathbf{x}), \mathbf{L}_i'\mathbf{f}'(\mathbf{x})]). \tag{137}$$

*Proof.* We can write

$$\mathbf{S}\mathbf{L}^{\top}\mathbf{f}(\mathbf{x}) = \mathbf{S}'\mathbf{L}'^{\top}\mathbf{f}'(\mathbf{x}) + \mathbf{S}\mathbf{L}^{\top}\mathbf{f}(\mathbf{x}) - \mathbf{S}'\mathbf{L}'^{\top}\mathbf{f}'(\mathbf{x}). \tag{138}$$

So setting

$$\epsilon_y(\mathbf{x}) = \mathbf{S}\mathbf{L}^{\top}\mathbf{f}(\mathbf{x}) - \mathbf{S}'\mathbf{L}'^{\top}\mathbf{f}'(\mathbf{x}) \tag{139}$$

gives $\epsilon_{y_i}(\mathbf{x})$ as in [Equation (8)](#). Since $\mathbf{L}^{\top}$ and $\mathbf{S}$ are invertible, the inverses exist, and we can left multiply with these inverses. We now consider the variance of $\epsilon_{yi}(\mathbf{x})$. Recall that

$$\operatorname{Var}_{zw}[z + w] = \operatorname{Var}_z[z] + \operatorname{Var}_w[w] + 2\operatorname{Cov}_{zw}[z, w] \tag{140}$$

This results in

$$\text{Var}_{\mathbf{x}}[\epsilon_{yi}(\mathbf{x})] = \text{Var}_{\mathbf{x}}\left[\frac{\mathbf{L}_i\mathbf{f}(\mathbf{x})}{\sqrt{\text{Var}_{\mathbf{x}}[\mathbf{L}_i\mathbf{f}(\mathbf{x})]}} - \frac{\mathbf{L}_i'\mathbf{f}'(\mathbf{x})}{\sqrt{\text{Var}[\mathbf{L}_i'\mathbf{f}'(\mathbf{x})]}}\right] \tag{141}$$

$$= \text{Var}_{\mathbf{x}}\left[\frac{\mathbf{L}_i\mathbf{f}(\mathbf{x})}{\sqrt{\text{Var}_{\mathbf{x}}[\mathbf{L}_i\mathbf{f}(\mathbf{x})]}}\right] + \text{Var}_{\mathbf{x}}\left[\frac{\mathbf{L}_i'\mathbf{f}'(\mathbf{x})}{\sqrt{\text{Var}[\mathbf{L}_i'\mathbf{f}'(\mathbf{x})]}}\right] \tag{142}$$

$$- 2\,\text{Cov}_{\mathbf{xx}}\left[\frac{\mathbf{L}_i\mathbf{f}(\mathbf{x})}{\sqrt{\text{Var}[\mathbf{L}_i\mathbf{f}(\mathbf{x})]}}, \frac{\mathbf{L}_i'\mathbf{f}'(\mathbf{x})}{\sqrt{\text{Var}[\mathbf{L}_i'\mathbf{f}'(\mathbf{x})]}}\right] \tag{143}$$

$$= 2 - 2\,\text{Cov}_{\mathbf{xx}}\left[\frac{\mathbf{L}_i\mathbf{f}(\mathbf{x})}{\sqrt{\text{Var}[\mathbf{L}_i\mathbf{f}(\mathbf{x})]}}, \frac{\mathbf{L}_i'\mathbf{f}'(\mathbf{x})}{\sqrt{\text{Var}[\mathbf{L}_i'\mathbf{f}'(\mathbf{x})]}}\right] \tag{144}$$

$$= 2(1 - \text{Corr}[\mathbf{L}_i\mathbf{f}(\mathbf{x}), \mathbf{L}_i'\mathbf{f}'(\mathbf{x})]), \tag{145}$$

where we use Equation (140) for the second equality and that $\text{Var}_z[z/\text{std}(z)] = 1$ in the third equality. We thus have the result. $\qquad\square$

There are three things that are important from this result.

Firstly, as seen in Corollary E.1, when the distributions are equal, the error term becomes zero and we are back in the case of Theorem 2.2.

Secondly, the shape of the error term suggests what a distance should measure if we want it to give a guarantee for how far representations are from being invertible linear transformations of each other. We notice that Equation (8) can be rewritten as

$$\epsilon_{y_i}(\mathbf{x}) = \frac{\log p(y_i|\mathbf{x})}{\sqrt{\text{Var}_{\mathbf{x}}[\log p(y_i|\mathbf{x}) - \log p(y_0|\mathbf{x})]}} - \frac{\log p'(y_i|\mathbf{x})}{\sqrt{\text{Var}_{\mathbf{x}}[\log p'(y_i|\mathbf{x}) - \log p'(y_0|\mathbf{x})]}} \tag{146}$$

$$+ \frac{\log p'(y_0|\mathbf{x})}{\sqrt{\text{Var}_{\mathbf{x}}[\log p'(y_i|\mathbf{x}) - \log p'(y_0|\mathbf{x})]}} - \frac{\log p(y_0|\mathbf{x})}{\sqrt{\text{Var}_{\mathbf{x}}[\log p(y_i|\mathbf{x}) - \log p(y_0|\mathbf{x})]}} \tag{147}$$

that is, entirely in terms of the logarithm of distributions. We also see that if $\epsilon_{y_i}(\mathbf{x})$ is a constant (or equivalently, $\text{Var}[\epsilon_{y_i}(\mathbf{x})] = 0$), the embeddings will be invertible linear transformations of each other.

Thirdly, Lemma 4.1 fits with the observation made in Section 3. The KL divergence depends on the difference of the logarithms of distributions, but it also multiplies that difference by the likelihood of the label. Which means that we can have a relatively large difference in logarithms of distributions, even though the KL divergence would result in a small value.

A similar result for the unembeddings can be found below.

**Corollary E.3.** *Under the same assumptions as in Lemma 4.1, we have:*

$$\text{Var}_y[\epsilon_{\mathbf{x}i}(y)] = 2(1 - Corr[\mathbf{N}_i\mathbf{g}(y), \mathbf{N}_i'\mathbf{g}'(y)]). \tag{148}$$

*Proof.* We can write

$$\mathbf{SN}^\top\mathbf{g}(y) = \mathbf{S}'\mathbf{N}'^\top\mathbf{g}'(y) + \mathbf{SN}^\top\mathbf{g}(y) - \mathbf{S}'\mathbf{N}'^\top\mathbf{g}'(y). \tag{149}$$

So setting

$$\epsilon_{\mathbf{x}}(y) = \mathbf{SN}^\top\mathbf{g}(y) - \mathbf{S}'\mathbf{N}'^\top\mathbf{g}'(y) \tag{150}$$

gives $\epsilon_{\mathbf{x}i}(y)$ as the error term for the unembeddings.

We now consider the variance of $\epsilon_{\mathbf{x}_i}(y)$.

$$\mathrm{Var}_y[\epsilon_{\mathbf{x}i}(y)] = \mathrm{Var}_y \left[ \frac{\mathbf{N}_i\mathbf{g}(y)}{\sqrt{\mathrm{Var}_y[\mathbf{N}_i\mathbf{g}(y)]}} - \frac{\mathbf{N}_i'\mathbf{g}'(y)}{\sqrt{\mathrm{Var}_y[\mathbf{N}_i'\mathbf{g}'(y)]}} \right] \tag{151}$$

$$= \mathrm{Var}_y[\frac{\mathbf{N}_i\mathbf{g}(y)}{\sqrt{\mathrm{Var}_y[\mathbf{N}_i\mathbf{g}(y)]}}] + \mathrm{Var}_y[\frac{\mathbf{N}_i'\mathbf{g}'(y)}{\sqrt{\mathrm{Var}_y[\mathbf{N}_i'\mathbf{g}'(y)]}}] \tag{152}$$

$$- 2\,\mathrm{Cov}_{yy}[\frac{\mathbf{N}_i\mathbf{g}(y)}{\sqrt{\mathrm{Var}_y[\mathbf{N}_i\mathbf{g}(y)]}}, \frac{\mathbf{N}_i'\mathbf{g}'(y)}{\sqrt{\mathrm{Var}_y[\mathbf{N}_i'\mathbf{g}'(y)]}}] \tag{153}$$

$$= 2 - 2\,\mathrm{Cov}_{yy}[\frac{\mathbf{N}_i\mathbf{g}(y)}{\sqrt{\mathrm{Var}_y[\mathbf{N}_i\mathbf{g}(y)]}}, \frac{\mathbf{N}_i'\mathbf{g}'(y)}{\sqrt{\mathrm{Var}_y[\mathbf{N}_i'\mathbf{g}'(y)]}}] \tag{154}$$

$$= 2(1 - \mathrm{Corr}[\mathbf{N}_i\mathbf{g}(y), \mathbf{N}_i'\mathbf{g}'(y)])\,, \tag{155}$$

which gives us the result. □

## E.2 Distance Between Distributions

Recall that we use the following notation:

$$\psi_{\mathbf{x}}(y_i; p) := \sqrt{\mathrm{Var}_{\mathbf{x}}[\log p(y_i|\mathbf{x}) - \log p(y_0|\mathbf{x})]} \text{ and } \psi_y(\mathbf{x}_j; p) := \sqrt{\mathrm{Var}_y[\log p(y|\mathbf{x}_j) - \log p(y|\mathbf{x}_0)]}\,.$$

We restate Definition 4.4 and then prove its a distance metric.

**Definition 4.4** (Log-likelihood variance distance between distributions). *For any two probability distributions $p, p'$ for which there exists a common choice of $\mathcal{X}_{\mathrm{LLV}} \subset \mathcal{X}$ and $\mathcal{Y}_{\mathrm{LLV}} \subset \mathcal{Y}$ such that they both satisfy Assumption 4.3, for a fixed $\lambda \in \mathbb{R}^+$, and by considering the following terms:*

$$t_1 := \max_{y \in \mathcal{Y}_{\mathrm{LLV}} \backslash \{y_0\}} \left\{ \sqrt{\mathrm{Var}_{\mathbf{x}} \left[ \frac{\log p(y|\mathbf{x})}{\psi_{\mathbf{x}}(y;p)} - \frac{\log p'(y|\mathbf{x})}{\psi_{\mathbf{x}}(y;p')} \right]}, \sqrt{\mathrm{Var}_{\mathbf{x}} \left[ \frac{\log p(y_0|\mathbf{x})}{\psi_{\mathbf{x}}(y;p)} - \frac{\log p'(y_0|\mathbf{x})}{\psi_{\mathbf{x}}(y;p')} \right]} \right\}$$

$$t_2 := \max_{\mathbf{x} \in \mathcal{X}_{\mathrm{LLV}} \backslash \{\mathbf{x}_0\}} \left\{ \sqrt{\mathrm{Var}_y \left[ \frac{\log p(y|\mathbf{x})}{\psi_y(\mathbf{x};p)} - \frac{\log p'(y|\mathbf{x})}{\psi_y(\mathbf{x};p')} \right]}, \sqrt{\mathrm{Var}_y \left[ \frac{\log p(y|\mathbf{x}_0)}{\psi_y(\mathbf{x};p)} - \frac{\log p'(y|\mathbf{x}_0)}{\psi_y(\mathbf{x};p')} \right]} \right\}$$

$$t_3 := \max_{y \in \mathcal{Y}_{\mathrm{LLV}} \backslash \{y_0\}} |\psi_{\mathbf{x}}(y;p) - \psi_{\mathbf{x}}(y;p')|\,, \quad t_4 := \max_{\mathbf{x} \in \mathcal{X}_{\mathrm{LLV}} \backslash \{\mathbf{x}_0\}} |\psi_y(\mathbf{x};p) - \psi_y(\mathbf{x};p')|\,,$$

*the log-likelihood variance (LLV) distance between $p$ and $p'$ is given by*

$$d_{\mathrm{LLV}}^{\lambda}(p,p') := \max\{t_1, t_2, \lambda t_3, \lambda t_4\}\,. \tag{10}$$

To show that $d_{\mathrm{LLV}}^{\lambda}$ is a distance metric between sets of conditional probability distributions, we have to prove that: i) it is non-negative, ii) it is zero if and only if the models are equal (that is, the distributions are equal for all labels and all inputs) iii) it is symmetric and iv) it satisfies the triangle inequality.

*Proof.* i) Since variance is non-negative, the first two terms are non-negative. The second two terms are absolute values, so also non-negative. Thus, the maximum of these four terms is also non-negative.

ii) From the expression of $d_{\mathrm{LLV}}^{\lambda}$, if the distributions are equal for all $\mathbf{x} \in \mathcal{X}$ and all $y \in \mathcal{Y}$, then the distance becomes zero.

Now, assume the distance is zero. Then, in particular, $t_4 = 0$. So we have

$$\sqrt{\mathrm{Var}_y[\log p(y|\mathbf{x}_j) - \log p(y|\mathbf{x}_0)]} = \sqrt{\mathrm{Var}_y[\log p'(y|\mathbf{x}_j) - \log p'(y|\mathbf{x}_0)]} \tag{156}$$

for all $\mathbf{x}_j \in \mathcal{X}_{\mathrm{LLV}} \backslash \{\mathbf{x}_0\}$. So since $t_2 = 0$, we have

$$\mathrm{Var}_y [\log p(y|\mathbf{x}_j) - \log p'(y|\mathbf{x}_j)] = 0\,, \tag{157}$$

for all $\mathbf{x}_j \in \mathcal{X}_{\mathrm{LLV}}$. This means that for each $\mathbf{x}_j$, we have that the log difference of the distributions is a constant for all $y_i \in \mathcal{Y}_{\mathrm{LLV}}$, and so

$$\log p(y_i|\mathbf{x}_j) - \log p'(y_i|\mathbf{x}_j) = c_j\,, \tag{158}$$

for all $y_i$. This means that

$$\log p(y_i|\mathbf{x}_j) = \log p'(y_i|\mathbf{x}_j) + c_j \tag{159}$$

$$p(y_i|\mathbf{x}_j) = p'(y_i|\mathbf{x}_j) \cdot \exp(c_j). \tag{160}$$

Now since $p$ and $p'$ are probability distributions over the $y_i$'s, we have

$$1 = \sum_{i=1}^{c} p(y_i|\mathbf{x}_j) = \exp(c_j) \sum_{i=1}^{c} p'(y_i|\mathbf{x}_j) = \exp(c_j) \tag{161}$$

This means that $p(y|\mathbf{x}_j) = p'(y|\mathbf{x}_j)$ for all $y \in \mathcal{Y}$ and for the $\mathbf{x}_j \in \mathcal{X}_{\mathrm{LLV}}$. Since we have $t_1 = t_3 = 0$, we get that

$$\mathrm{Var}_{\mathbf{x}} \left[ \log p(y_i|\mathbf{x}) - \log p'(y_i|\mathbf{x}) \right] = 0, \tag{162}$$

for all $y_i \in \mathcal{Y}_{\mathrm{LLV}}$ and all inputs $\mathbf{x} \in \mathcal{X}$. Thus, we have

$$\log p(y_i|\mathbf{x}) - \log p'(y_i|\mathbf{x}) = c_i \tag{163}$$

for every choice of $y_i$ except one left out of the label set and all $\mathbf{x}$. This results in

$$p(y_i|\mathbf{x}) = p'(y_i|\mathbf{x}) \cdot \exp(c_i) \tag{164}$$

for all $\mathbf{x} \in \mathcal{X}$. In particular, this is true for $\mathbf{x}_j \in \mathcal{X}_{\mathrm{LLV}}$. So we have

$$p(y_i|\mathbf{x}_j) = p'(y_i|\mathbf{x}_j) \cdot \exp(c_i) = p'(y_i|\mathbf{x}_j) \tag{165}$$

and thus, $\exp(c_i) = 1$ for all $y_i \in \mathcal{Y}_{\mathrm{LLV}}$. Since the probability of the last label, $y_k$ is fixed by the other labels, i.e.

$$p(y_k|\mathbf{x}) = 1 - \sum_{i \neq k} p(y_i|\mathbf{x}) \tag{166}$$

this gives us that

$$p(y_i|\mathbf{x}) = p'(y_i|\mathbf{x}) \tag{167}$$

for all $y_i \in \mathcal{Y}$ and all $\mathbf{x} \in \mathcal{X}$.

iii) $d_{\mathrm{LLV}}^{\lambda}$ is symmetric because it depends on variances and on absolute values, which are symmetric.

iv) We prove that $d_{\mathrm{LLV}}^{\lambda}$ satisfies the triangle inequality by showing that all terms in it satisfy the triangle inequality. Assume we have three random variables $X, Y, Z$, then

$$\sqrt{\mathrm{Var}[X - Z]} = \sqrt{\mathrm{Var}[X - Y + Y - Z]} \tag{168}$$

$$= \sqrt{\mathrm{Var}[X - Y] + \mathrm{Var}[Y - Z] + 2\,\mathrm{Cov}[X - Y, Y - Z]} \tag{169}$$

$$\leq \sqrt{\mathrm{Var}[X - Y] + \mathrm{Var}[Y - Z] + 2\sqrt{\mathrm{Var}[X - Y]}\sqrt{\mathrm{Var}[Y - Z]}} \tag{170}$$

$$= \sqrt{\left(\sqrt{\mathrm{Var}[X - Y]} + \sqrt{\mathrm{Var}[Y - Z]}\right)^2} \tag{171}$$

$$= \sqrt{\mathrm{Var}[X - Y]} + \sqrt{\mathrm{Var}[Y - Z]}. \tag{172}$$

So the square root of the variance of differences of random variables satisfies the triangle inequality. Taking the maximum also preserves the triangle inequality. Therefore, $t_1$ and $t_2$ satisfy the triangle inequality. Concerning the last two terms $t_3$ and $t_4$, we see that for models $p, p', p^*$ we have

$$\left| \sqrt{\mathrm{Var}_y[\log p(y|\mathbf{x}_j) - \log p(y|\mathbf{x}_k)]} - \sqrt{\mathrm{Var}_y[\log p'(y|\mathbf{x}_j) - \log p'(y|\mathbf{x}_k)]} \right|$$

$$= \left| \sqrt{\mathrm{Var}_y[\log p(y|\mathbf{x}_j) - \log p(y|\mathbf{x}_k)]} - \sqrt{\mathrm{Var}_y[\log p^*(y|\mathbf{x}_j) - \log p^*(y|\mathbf{x}_k)]} \right.$$

$$\left. + \sqrt{\mathrm{Var}_y[\log p^*(y|\mathbf{x}_j) - \log p^*(y|\mathbf{x}_k)]} - \sqrt{\mathrm{Var}_y[\log p'(y|\mathbf{x}_j) - \log p'(y|\mathbf{x}_k)]} \right|$$

$$\leq \left| \sqrt{\mathrm{Var}_y[\log p(y|\mathbf{x}_j) - \log p(y|\mathbf{x}_k)]} - \sqrt{\mathrm{Var}_y[\log p^*(y|\mathbf{x}_j) - \log p^*(y|\mathbf{x}_k)]} \right|$$

$$+ \left| \sqrt{\mathrm{Var}_y[\log p^*(y|\mathbf{x}_j) - \log p^*(y|\mathbf{x}_k)]} - \sqrt{\mathrm{Var}_y[\log p'(y|\mathbf{x}_j) - \log p'(y|\mathbf{x}_k)]} \right|.$$

So the second two terms also satisfy the triangle inequality. Thus, the sum of these four terms satisfies the triangle inequality. $\qquad \square$

### E.3 Implementation of the Log-Likelihood Variance Distance

Here, we collect the implementation details of $d_{\mathrm{LLV}}^{\lambda}$. We set the weighting constant $\lambda$ to $10^{-5}$. To choose the best pivot $y_0 \in \mathcal{Y}$ and the left-out label $y_\varnothing \in \mathcal{Y}$, we evaluate $t_1$ for all possible choices of $(y_0, y_\varnothing)$, then select those giving the smallest value of $d_{\mathrm{LLV}}^{\lambda}$. For the input set $\mathcal{X}_{\mathrm{LLV}}$, we use $M + 1$ inputs. In our experiments, the representational dimension is 2, so we collect 3 inputs. We randomly draw 200 samples of sets from $\mathcal{X}$ containing 3 inputs, and choose the set giving the smallest $t_2$.

In our experiments, we only make pairwise comparisons of models. Notice that, in the case where one would like to compare three or more models, the same pivot $y_0$, left-out label $y_\varnothing$, and input set $\mathcal{X}_{\mathrm{LLV}}$ must be chosen for all models.

### E.4 Using PLS SVD to Define a Dissimilarity Measure

We restate Definition 4.5 and then prove some important properties.

**Definition 4.5** (PLS SVD distance between vectors). *Let $\mathbf{z}, \mathbf{w}$ be two $M$-dimensional random vectors, and define $\mathbf{z}', \mathbf{w}'$ by standardizing their components: $z_i' = (z_i - \mathbb{E}[z_i])/\operatorname{std}(z_i)$, $w_i' = (w_i - \mathbb{E}[w_i])/\operatorname{std}(w_i)$. Let $\{\mathbf{u}_i\}_{i=1}^{M}$ and $\{\mathbf{v}_i\}_{i=1}^{M}$ be the left and right singular vectors of the cross-covariance matrix $\mathbf{\Sigma}_{\mathbf{z}'\mathbf{w}'}$. We define*

$$m_{\mathrm{SVD}}(\mathbf{z}, \mathbf{w}) := \frac{1}{M} \sum_{i=1}^{M} \operatorname{Cov}_{\mathbf{z}'\mathbf{w}'}[\mathbf{u}_i^\top \mathbf{z}', \mathbf{v}_i^\top \mathbf{w}'], \quad d_{\mathrm{SVD}}(\mathbf{z}, \mathbf{w}) := 1 - m_{\mathrm{SVD}}(\mathbf{z}, \mathbf{w}) . \tag{12}$$

We show that this measure is zero from a vector to itself and is invariant to multiplications of the scaled variables with orthonormal matrices. Before proceeding, we recall the following property of the trace of two matrices:

$$tr(\mathbf{AB}) = tr(\mathbf{BA}) . \tag{173}$$

In the next Proposition, we prove that $d_{\mathrm{SVD}}(\mathbf{z}, \mathbf{z}) = 0$.

**Proposition E.4.** *Let $\mathbf{z}$ be an $M$-dimensional random variable and assume that the covariance matrix $\mathbf{\Sigma}_{\mathbf{z}'\mathbf{z}'}$ of the centered and scaled variable $\mathbf{z}'$ is non-singular. Then*

$$m_{\mathrm{SVD}}(\mathbf{z}, \mathbf{z}) = \frac{1}{M} \sum_{i=1}^{M} \operatorname{Cov}_{\mathbf{z}'\mathbf{z}'}[\mathbf{u}_i^\top \mathbf{z}', \mathbf{u}_i^\top \mathbf{z}'] = 1 , \tag{174}$$

*where $\mathbf{u}_i$ is the $i$'th singular vector of $\mathbf{\Sigma}_{\mathbf{z}'\mathbf{z}'}$.*

*Proof.* Let

$$\mathbf{\Sigma}_{\mathbf{z}'\mathbf{z}'} = \mathbf{U}\mathbf{D}\mathbf{U}^\top \tag{175}$$

be the singular value decomposition of the covariance matrix of $\mathbf{z}'$ with itself. Then, using Equation (173), we have that

$$tr(\mathbf{\Sigma}_{\mathbf{z}'\mathbf{z}'}) = tr(\mathbf{U}\mathbf{D}\mathbf{U}^\top) = tr(\mathbf{U}^\top \mathbf{U}\mathbf{D}) = tr(\mathbf{D}) = \sum_{i=1}^{M} \sigma_i , \tag{176}$$

where $\sigma_i$ are the singular values of $\mathbf{\Sigma}_{\mathbf{z}'\mathbf{z}'}$. Since $\operatorname{Var}[z_i'] = 1$ for all $i \in \{1, \ldots, M\}$, we also have that

$$tr(\mathbf{\Sigma}_{\mathbf{z}'\mathbf{z}'}) = \sum_{i=1}^{M} \operatorname{Var}_{\mathbf{z}'}[z_i'] = M . \tag{177}$$

Combining the two results, we get

$$\sum_{i=1}^{M} \sigma_i = M . \tag{178}$$

Therefore, the mean of the covariances becomes

$$m_{\text{SVD}}(\mathbf{z}, \mathbf{z}) = \frac{1}{M} \sum_{i=1}^{M} \text{Cov}_{\mathbf{z'z'}}[\mathbf{u}_i^\top \mathbf{z}', \mathbf{u}_i^\top \mathbf{z}'] \tag{179}$$

$$= \frac{1}{M} tr(\mathbf{U}^\top \text{Cov}_{\mathbf{z'z'}}[\mathbf{z}', \mathbf{z}']\mathbf{U}) \tag{180}$$

$$= \frac{1}{M} \sum_{i=1}^{M} \sigma_i = \frac{M}{M} = 1 \tag{181}$$

which proves the claim. $\qquad\square$

Next, we prove the invariance to translation and orthogonal transformations of the members in $m_{\text{SVD}}$:

**Proposition E.5.** *Let $\mathbf{z}$ and $\mathbf{w}$ be $M$-dimensional random variables. Then, the measure*

$$m_{\text{SVD}}(\mathbf{z}, \mathbf{w}) = \frac{1}{M} \sum_{i=1}^{M} \text{Cov}_{\mathbf{z'w'}}[\mathbf{u}_i^\top \mathbf{z}', \mathbf{v}_i^\top \mathbf{w}'] \tag{182}$$

*is invariant to translations and multiplications with an orthonormal matrix after the scaling.*

*Proof.* **Invariance to translations**. Since $m_{\text{SVD}}(\mathbf{z}, \mathbf{w})$ is based on covariances, and covariances are invariant to translations, $m_{\text{SVD}}(\mathbf{z}, \mathbf{w})$ is invariant to translations.

**Invariance to multiplication with orthonormal matrix**. Assume $\mathbf{T}$ and $\mathbf{R}$ are orthonormal matrices. Let $\mathbf{h} = \mathbf{Tz}'$ and $\mathbf{k} = \mathbf{Rw}'$. Let the singular value decomposition of $\mathbf{\Sigma}_{\mathbf{z'w'}}$ be:

$$\mathbf{\Sigma}_{\mathbf{z'w'}} = \mathbf{UDV}^\top, \tag{183}$$

then

$$\mathbf{\Sigma}_{\mathbf{hk}} = \mathbf{TUDV}^\top \mathbf{R}^\top \tag{184}$$

$$= (\mathbf{TU})\mathbf{D}(\mathbf{RV})^\top. \tag{185}$$

Since $\mathbf{T}$ and $\mathbf{R}$ are orthonormal, $\mathbf{TU}$ and $\mathbf{RV}$ are also orthonormal, so $\text{Cov}_{\mathbf{hk}}[\mathbf{h}, \mathbf{k}]$ has the same singular values as $\mathbf{\Sigma}_{\mathbf{z'w'}}$. Thus, the measure is invariant to multiplications with orthonormal matrices (rotations) after the scaling. $\qquad\square$

### E.5 Lower Bound on Mean PLS-SVD

To show how the probability distributions are connected to the representations, we need an intermediate result, where we bound $m_{\text{SVD}}$. To prove this bound, we will make use of Weyl's Inequality and we therefore restate it from [59].

**Theorem E.6** (Weyl's Inequality)**.** *Let $\mathbf{A}$ be a $n \times m$ matrix. Let $\mathbf{B} = \mathbf{A} + \mathbf{E}$. Let $\sigma_i$ be the $i$'th singular value of $\mathbf{A}$ and let $\tilde{\sigma}_i$ be the $i$'th singular value of $\mathbf{B}$ (ordered from largest to smallest). Then*

$$|\sigma_i - \tilde{\sigma}_i| \leq \|\mathbf{E}\|_s, \tag{186}$$

*where $\|.\|_s$ is the spectral norm defined as*

$$\|\mathbf{E}\|_s = \max_{\|\mathbf{v}\|=1} \|\mathbf{Ev}\|_2 \tag{187}$$

*and $\|.\|_2$ is the Euclidean vector norm. Or equivalently:*

$$\|\mathbf{E}\|_s = \sigma_{\max}(\mathbf{E}), \tag{188}$$

*where $\sigma_{\max}(\mathbf{E})$ is the largest singular value of $\mathbf{E}$.*

We now state and prove a bound on $m_{\text{SVD}}$.

**Lemma E.7.** *Let $\mathbf{z}, \mathbf{w}$ be two $M$-dimensional random vectors, and define $\mathbf{z}', \mathbf{w}'$ by standardizing their components: $z_i' = (z_i - \mathbb{E}[z_i])/\operatorname{std}(z_i)$, $w_i' = (w_i - \mathbb{E}[w_i])/\operatorname{std}(w_i)$. Assume the $M \times M$ matrices $\mathbf{\Sigma}_{\mathbf{z}'\mathbf{z}'}$ and $\mathbf{\Sigma}_{\mathbf{z}'\mathbf{w}'} \in \mathbb{R}^{M \times M}$ are full-rank. Let $\{\mathbf{u}_i\}_{i=1}^M$ and $\{\mathbf{v}_i\}_{i=1}^M$ be the left and right singular vectors of $\mathbf{\Sigma}_{\mathbf{z}'\mathbf{w}'}$. Then, the following bound holds:*

$$m_{\mathrm{SVD}}(\mathbf{z}, \mathbf{w}) = \frac{1}{M} \sum_{i=1}^M \operatorname{Cov}_{\mathbf{z}'\mathbf{w}'}\left[\mathbf{u}_i^\top \mathbf{z}', \mathbf{v}_i^\top \mathbf{w}'\right] \geq 1 - \sqrt{M \sum_{l=1}^M \operatorname{Var}_{\mathbf{z}',\mathbf{w}'}[z_l' - w_l']} \quad (189)$$

where $\operatorname{Var}_{\mathbf{z},\mathbf{w}}[z_l' - w_l']$ denotes the variance over the joint distribution over $\mathbf{z}'$ and $\mathbf{w}'$.

*Proof.* For ease of notation, assume $\mathbf{z}$ and $\mathbf{w}$ are $M$-dimensional random variables which are already centered and scaled. So $\operatorname{Var}[z_i] = 1$ for all components $i = 1, \ldots, M$. Below we will make use of the covariance inequality, which says that for scalar variables $z, w$,

$$-\sqrt{\operatorname{Var}_z[z]}\sqrt{\operatorname{Var}_w[w]} \leq \operatorname{Cov}_{zw}[z, w] \leq \sqrt{\operatorname{Var}_z[z]}\sqrt{\operatorname{Var}_w[w]}. \quad (190)$$

Let $\mathbf{A} := \mathbf{\Sigma}_{\mathbf{zw}}$ be the cross-covariance matrix of $\mathbf{z}$ and $\mathbf{w}$, and let $\mathbf{B} := \mathbf{\Sigma}_{\mathbf{zz}}$ be the covariance matrix of $\mathbf{z}$. Then

$$\mathbf{B} = \mathbf{A} + (\mathbf{B} - \mathbf{A}), \quad (191)$$

and we define $\mathbf{E} := \mathbf{B} - \mathbf{A}$. We can write the $i, j$'th entry of $\mathbf{E}$ as

$$(\mathbf{E})_{i,j} = \operatorname{Cov}_{\mathbf{zz}}[z_i, z_j] - \operatorname{Cov}_{\mathbf{zw}}[z_i, w_j] = \operatorname{Cov}_{\mathbf{zw}}[z_i, z_j - w_j], \quad (192)$$

because of the bilinearity of the covariance, see section 13.2.7 of Adhikari and Pitman [1], "The Main Property: Bilinearity".

The singular value decompositions of $\mathbf{A}$ and $\mathbf{B}$ always exist and can be written as follows:

$$\mathbf{A} = \mathbf{U}\mathbf{D}\mathbf{V}^\top, \quad \mathbf{B} = \mathbf{W}\mathbf{\Sigma}\mathbf{W}^\top, \quad (193)$$

where $\mathbf{U}, \mathbf{V}$ and $\mathbf{W}$ are orthonormal matrices and $\mathbf{D}$ and $\mathbf{\Sigma}$ are diagonal matrices containing the singular values. Note that since $\mathbf{B}$ is symmetric and positive definite, the left and right singular vectors coincide. Let $\tilde{\sigma}_i$ be the $i$'th singular value of $\mathbf{A}$ (sorted from largest to smallest) and let $\sigma_i$ be the $i$'th singular value of $\mathbf{B}$. Now Weyl's inequality E.6 gives us that

$$|\sigma_i - \tilde{\sigma}_i| \leq \|\mathbf{E}\|_s. \quad (194)$$

Since the spectral norm of $\mathbf{E}$ is the largest singular value of $\mathbf{E}$

$$\|\mathbf{E}\|_s = \sigma_{\max}(\mathbf{E}) \quad (195)$$

and the frobenius norm is the square root of the sum of the squared singular values

$$\|\mathbf{E}\|_F = \sqrt{\sum_{i=1}^M \sigma_i^2(\mathbf{E})} = \sqrt{\operatorname{trace}(\mathbf{E}^\top \mathbf{E})}, \quad (196)$$

we have that

$$\|\mathbf{E}\|_s \leq \|\mathbf{E}\|_F = \sqrt{\operatorname{trace}(\mathbf{E}^\top \mathbf{E})}. \quad (197)$$

Next, we bound the trace of $\mathbf{E}^\top \mathbf{E}$:

$$\operatorname{trace}(\mathbf{E}^\top \mathbf{E}) = \sum_{l=1}^M \sum_{k=1}^M \operatorname{Cov}_{\mathbf{zw}}[z_k, z_l - w_l]^2 \quad (198)$$

$$\leq \sum_{l=1}^M \sum_{k=1}^M (\sqrt{\operatorname{Var}_{\mathbf{z}}[z_k]}\sqrt{\operatorname{Var}_{\mathbf{z},\mathbf{w}}[z_l - w_l]})^2 \quad (199)$$

$$= \sum_{l=1}^M M\operatorname{Var}_{\mathbf{z},\mathbf{w}}[z_l - w_l], \quad (200)$$

where in the first step we use the covariance inequality (Equation (190)) and in the second we use that $\mathrm{Var}[z_k] = 1$ by construction. Combining the inequalities from Equations (194), (197) and (200) we obtain

$$|\sigma_i - \tilde{\sigma}_i| \leq \sqrt{\sum_{l=1}^{M} M \mathrm{Var}_{\mathbf{z},\mathbf{w}}[z_l - w_l]}, \tag{201}$$

which means that the difference between $\sigma_i$ and $\tilde{\sigma}_i$ is at most $\sqrt{\sum_{l=1}^{M} M \mathrm{Var}_{\mathbf{z},\mathbf{w}}[z_l - w_l]}$. This means that if $\tilde{\sigma}_i < \sigma_i$, we still have :

$$\tilde{\sigma}_i \geq \sigma_i - \sqrt{\sum_{l=1}^{M} M \mathrm{Var}_{\mathbf{z},\mathbf{w}}[z_l - w_l]}. \tag{202}$$

and if $\tilde{\sigma}_i \geq \sigma_i$, we also have

$$\tilde{\sigma}_i \geq \sigma_i - \sqrt{\sum_{l=1}^{M} M \mathrm{Var}_{\mathbf{z},\mathbf{w}}[z_l - w_l]}. \tag{203}$$

since $\sqrt{\sum_{l=1}^{M} M \mathrm{Var}[z_l - w_l]} \geq 0$. Let $\mathbf{v}_i$ be the $i$'th column of $\mathbf{V}$ and recall that from the SVD of we have

$$\tilde{\sigma}_i = \mathbf{u}_i^\top \mathbf{A} \mathbf{v}_i = \mathrm{Cov}_{\mathbf{zw}} \left[ \sum_{k=1}^{M} \mathbf{u}_{ik} z_k, \sum_{l=1}^{M} \mathbf{v}_{il} w_l \right] \tag{204}$$

$$= \mathrm{Cov}_{\mathbf{zw}} \left[ \mathbf{u}_i^\top \mathbf{z}, \mathbf{v}_i^\top \mathbf{w} \right], \tag{205}$$

because of the bilinearity of the covariance. Thus, Equation (202) gives us the bound

$$\mathrm{Cov}_{\mathbf{zw}} \left[ \mathbf{u}_i^\top \mathbf{z}, \mathbf{v}_i^\top \mathbf{w} \right] \geq \sigma_i - \sqrt{\sum_{l=1}^{M} M \mathrm{Var}_{\mathbf{z},\mathbf{w}}[z_l - w_l]} \tag{206}$$

and therefore

$$m_{\mathrm{SVD}}(\mathbf{z}, \mathbf{w}) = \frac{1}{m} \sum_{i=1}^{M} [\mathrm{Cov}_{\mathbf{zw}}[\mathbf{u}_i^\top \mathbf{z}, \mathbf{v}_i^\top \mathbf{w}]] \geq 1 - \sqrt{\sum_{l=1}^{M} M \mathrm{Var}_{\mathbf{z},\mathbf{w}}[z_l - w_l]}, \tag{207}$$

where we used that $\frac{1}{m} \sum_{i=1}^{M} \sigma_i = 1$ from Proposition E.4. $\qquad \square$

### E.6 Bounding Representational Similarity with Distribution Distance

We can now use Definitions 4.4 and 4.5 and Lemma E.7 to show how a bound on the distance between probability distributions can give us a bound on the distance between representations. For the result below, let $\mathbf{L}, \mathbf{L}'$ be defined as in Theorem 2.2. Let $\mathbf{N}$ (resp. $\mathbf{N}'$) be the matrix with columns $\mathbf{f}_0(\mathbf{x})$ (resp. $\mathbf{f}_0'(\mathbf{x})$). We denote with $\mathbf{z}_1 := \mathbf{L}^\top \mathbf{f}(\mathbf{x})$, $\mathbf{z}_2 := \mathbf{L}'^\top \mathbf{f}'(\mathbf{x})$, $\mathbf{w}_1 := \mathbf{N}^\top \mathbf{g}(y)$ and $\mathbf{w}_2 := \mathbf{N}'^\top \mathbf{g}'(y)$.

**Theorem 4.7.** *Let* $(\mathbf{f}, \mathbf{g}), (\mathbf{f}', \mathbf{g}') \in \Theta$ *be two models such that: (1) There exist* $\mathcal{X}_{\mathrm{LLV}} \subset \mathcal{X}$ *and* $\mathcal{Y}_{\mathrm{LLV}} \subset \mathcal{Y}$, *consisting of a pivot point and all labels but one, such that all* $\mathbf{L}, \mathbf{L}'$ *and* $\mathbf{N}, \mathbf{N}'$ *matrices constructed from these sets are invertible;[16] (2) Both* $p_{\mathbf{f},\mathbf{g}}$ *and* $p_{\mathbf{f}',\mathbf{g}'}$ *satisfy Assumption 4.3 for* $\mathcal{X}_{\mathrm{LLV}}$ *and* $\mathcal{Y}_{\mathrm{LLV}}$; *(3) The covariance matrices* $\mathbf{\Sigma}_{\mathbf{z}_1 \mathbf{z}_1}, \mathbf{\Sigma}_{\mathbf{w}_1 \mathbf{w}_1}$ *and the cross-covariance matrices* $\mathbf{\Sigma}_{\mathbf{z}_1 \mathbf{z}_2}, \mathbf{\Sigma}_{\mathbf{w}_1 \mathbf{w}_2}$ *are non-singular. Then, for any weighting constant* $\lambda \in \mathbb{R}^+$, *we have*

$$d_{\mathrm{LLV}}^\lambda(p_{\mathbf{f},\mathbf{g}}, p_{\mathbf{f}',\mathbf{g}'}) \leq \epsilon \implies d_{\mathbf{f},\mathbf{g}}((\mathbf{f}, \mathbf{g}), (\mathbf{f}', \mathbf{g}')) \leq 2M\epsilon. \tag{14}$$

---

[16]This is slightly stronger than the diversity condition (Definition 2.1) in the sense that we need diversity to hold for all sets using labels from $\mathcal{Y}_{\mathrm{LLV}}$ and the chosen pivot point.

*Proof.* We start from

$$d_{\text{SVD}}(\mathbf{L}^\top \mathbf{f}(\mathbf{x}), \mathbf{L}^{'\top} \mathbf{f}'(\mathbf{x})) = 1 - m_{\text{SVD}}(\mathbf{L}^\top \mathbf{f}(\mathbf{x}), \mathbf{L}^{'\top} \mathbf{f}'(\mathbf{x})) \tag{208}$$

and consider the components $\mathbf{L}_l, \mathbf{L}'_l$, which are $l$'th row of $\mathbf{L}^\top, \mathbf{L}^{'\top}$, respectively. In the following, notice that

$$\psi_{\mathbf{x}}(y_l; p) := \sqrt{\text{Var}_{\mathbf{x}}[\log p(y_l|\mathbf{x}) - \log p(y_0|\mathbf{x})]} = \sqrt{\text{Var}[\mathbf{L}_l\mathbf{f}(\mathbf{x})]} \tag{209}$$

Using Lemma E.7, we have that

$$m_{\text{SVD}}(\mathbf{L}^\top \mathbf{f}(\mathbf{x}), \mathbf{L}^{'\top} \mathbf{f}'(\mathbf{x})) \geq 1 - \sqrt{\sum_{l=1}^{M} M\text{Var}\left[\frac{\mathbf{L}_l\mathbf{f}(\mathbf{x})}{\sqrt{\text{Var}[\mathbf{L}_l\mathbf{f}(\mathbf{x})]}} - \frac{\mathbf{L}'_l\mathbf{f}'(\mathbf{x})}{\sqrt{\text{Var}[\mathbf{L}'_l\mathbf{f}'(\mathbf{x})]}}\right]}. \tag{210}$$

Therefore, we have

$$d_{\text{SVD}}(\mathbf{L}^\top \mathbf{f}(\mathbf{x}), \mathbf{L}^{'\top} \mathbf{f}'(\mathbf{x})) \leq \sqrt{\sum_{l=1}^{M} M\text{Var}\left[\frac{\mathbf{L}_l\mathbf{f}(\mathbf{x})}{\sqrt{\text{Var}[\mathbf{L}_l\mathbf{f}(\mathbf{x})]}} - \frac{\mathbf{L}'_l\mathbf{f}'(\mathbf{x})}{\sqrt{\text{Var}[\mathbf{L}'_l\mathbf{f}'(\mathbf{x})]}}\right]}. \tag{211}$$

Considering the variance term, we can rewrite it as follows:

$$\text{Var}\left[\frac{\mathbf{L}_l\mathbf{f}(\mathbf{x})}{\sqrt{\text{Var}[\mathbf{L}_l\mathbf{f}(\mathbf{x})]}} - \frac{\mathbf{L}'_l\mathbf{f}'(\mathbf{x})}{\sqrt{\text{Var}[\mathbf{L}'_l\mathbf{f}'(\mathbf{x})]}}\right] = \text{Var}\left[\frac{\log p(y_l|x) - \log p(y_0|x)}{\sqrt{\text{Var}[\mathbf{L}_l\mathbf{f}(\mathbf{x})]}} - \frac{\log p'(y_l|x) - \log p'(y_0|x)}{\sqrt{\text{Var}[\mathbf{L}'_l\mathbf{f}'(\mathbf{x})]}}\right]$$

$$= \text{Var}\left[\frac{\log p(y_l|x)}{\sqrt{\text{Var}[\mathbf{L}_l\mathbf{f}(\mathbf{x})]}} - \frac{\log p'(y_l|x)}{\sqrt{\text{Var}[\mathbf{L}'_l\mathbf{f}'(\mathbf{x})]}}\right]$$

$$+ \text{Var}\left[\frac{\log p'(y_0|x)}{\sqrt{\text{Var}[\mathbf{L}'_l\mathbf{f}'(\mathbf{x})]}} - \frac{\log p(y_0|x)}{\sqrt{\text{Var}[\mathbf{L}_l\mathbf{f}(\mathbf{x})]}}\right]$$

$$+ 2\,\text{Cov}\left[\frac{\log p(y_l|x)}{\sqrt{\text{Var}[\mathbf{L}_l\mathbf{f}(\mathbf{x})]}} - \frac{\log p'(y_l|x)}{\sqrt{\text{Var}[\mathbf{L}'_l\mathbf{f}'(\mathbf{x})]}}, \frac{\log p'(y_0|x)}{\sqrt{\text{Var}[\mathbf{L}'_l\mathbf{f}'(\mathbf{x})]}} - \frac{\log p(y_0|x)}{\sqrt{\text{Var}[\mathbf{L}_l\mathbf{f}(\mathbf{x})]}}\right]$$

$$\leq \text{Var}\left[\frac{\log p(y_l|x)}{\sqrt{\text{Var}[\mathbf{L}_l\mathbf{f}(\mathbf{x})]}} - \frac{\log p'(y_l|x)}{\sqrt{\text{Var}[\mathbf{L}'_l\mathbf{f}'(\mathbf{x})]}}\right]$$

$$+ \text{Var}\left[\frac{\log p'(y_0|x)}{\sqrt{\text{Var}[\mathbf{L}'_l\mathbf{f}'(\mathbf{x})]}} - \frac{\log p(y_0|x)}{\sqrt{\text{Var}[\mathbf{L}_l\mathbf{f}(\mathbf{x})]}}\right]$$

$$+ 2\sqrt{\text{Var}\left[\frac{\log p(y_l|x)}{\sqrt{\text{Var}[\mathbf{L}_l\mathbf{f}(\mathbf{x})]}} - \frac{\log p'(y_l|x)}{\sqrt{\text{Var}[\mathbf{L}'_l\mathbf{f}'(\mathbf{x})]}}\right]}\sqrt{\text{Var}\left[\frac{\log p'(y_0|x)}{\sqrt{\text{Var}[\mathbf{L}'_l\mathbf{f}'(\mathbf{x})]}} - \frac{\log p(y_0|x)}{\sqrt{\text{Var}[\mathbf{L}_l\mathbf{f}(\mathbf{x})]}}\right]}.$$

By assumption, $d_{\text{LLV}}^\lambda(p, p') \leq \epsilon$, which means that for all $y_l \in \mathcal{Y}_{\text{LLV}}$:

$$\sqrt{\text{Var}\left[\frac{\log p(y_l|x)}{\sqrt{\text{Var}[\mathbf{L}_l\mathbf{f}(\mathbf{x})]}} - \frac{\log p'(y_l|x)}{\sqrt{\text{Var}[\mathbf{L}'_l\mathbf{f}'(\mathbf{x})]}}\right]} \leq \epsilon. \tag{212}$$

Hence, we obtain

$$d_{\text{SVD}}(\mathbf{L}^\top \mathbf{f}(\mathbf{x}), \mathbf{L}^{'\top} \mathbf{f}'(\mathbf{x})) \leq \sqrt{\sum_{l=1}^{M} M\text{Var}\left[\frac{\mathbf{L}_l\mathbf{f}(\mathbf{x})}{\sqrt{\text{Var}[\mathbf{L}_l\mathbf{f}(\mathbf{x})]}} - \frac{\mathbf{L}'_l\mathbf{f}'(\mathbf{x})}{\sqrt{\text{Var}[\mathbf{L}'_l\mathbf{f}'(\mathbf{x})]}}\right]} \tag{213}$$

$$\leq \sqrt{\sum_{l=1}^{M} M(\epsilon^2 + \epsilon^2 + 2\epsilon^2)} \tag{214}$$

$$= \sqrt{M^2 4\epsilon^2} = 2M\epsilon, \tag{215}$$

giving the first part of the result. Next, we consider

$$d_{\text{SVD}}(\mathbf{N}^\top \mathbf{g}(y), \mathbf{N}'^\top \mathbf{g}'(y)) = 1 - m_{\text{SVD}}(\mathbf{N}^\top \mathbf{g}(y), \mathbf{N}'^\top \mathbf{g}'(y)). \tag{216}$$

Let $\mathbf{N}_j, \mathbf{N}'_j$ be the $j$'th row of $\mathbf{N}^\top, \mathbf{N}'^\top$. Using Lemma E.7, we obtain:

$$m_{\text{SVD}}(\mathbf{N}^\top \mathbf{g}(y), \mathbf{N}'^\top \mathbf{g}'(y)) \geq 1 - \sqrt{\sum_{l=1}^{M} M \text{Var} \left[ \frac{\mathbf{N}_j \mathbf{g}(y)}{\sqrt{\text{Var}[\mathbf{N}_j \mathbf{g}(y)]}} - \frac{\mathbf{N}'_j \mathbf{g}'(y)}{\sqrt{\text{Var}[\mathbf{N}'_j \mathbf{g}'(y)]}} \right]}. \tag{217}$$

This implies the following:

$$d_{\text{SVD}}(\mathbf{N}^\top \mathbf{g}(y), \mathbf{N}'^\top \mathbf{g}'(y)) \leq \sqrt{\sum_{l=1}^{M} M \text{Var} \left[ \frac{\mathbf{N}_j \mathbf{g}(y)}{\sqrt{\text{Var}[\mathbf{N}_j \mathbf{g}(y)]}} - \frac{\mathbf{N}'_j \mathbf{g}'(y)}{\sqrt{\text{Var}[\mathbf{N}'_j \mathbf{g}'(y)]}} \right]}. \tag{218}$$

Considering the variance term, we can rework it as follows:

$$\text{Var} \left[ \frac{\mathbf{N}_j \mathbf{g}(y)}{\sqrt{\text{Var}[\mathbf{N}_j \mathbf{g}(y)]}} - \frac{\mathbf{N}'_j \mathbf{g}'(y)}{\sqrt{\text{Var}[\mathbf{N}'_j \mathbf{g}'(y)]}} \right] = \text{Var} \left[ \frac{\log p(y|\mathbf{x}_j) - \log p(y|\mathbf{x}_0)}{\sqrt{\text{Var}[\mathbf{N}_j \mathbf{g}(y)]}} - \frac{\log p'(y|\mathbf{x}_j) - \log p'(y|\mathbf{x}_0)}{\sqrt{\text{Var}[\mathbf{N}'_j \mathbf{g}'(y)]}} \right]$$

$$= \text{Var} \left[ \frac{\log p(y|\mathbf{x}_j)}{\sqrt{\text{Var}[\mathbf{N}_j \mathbf{g}(y)]}} - \frac{\log p'(y|\mathbf{x}_j)}{\sqrt{\text{Var}[\mathbf{N}'_j \mathbf{g}'(y)]}} \right]$$

$$+ \text{Var} \left[ \frac{\log p'(y|\mathbf{x}_0)}{\sqrt{\text{Var}[\mathbf{N}'_j \mathbf{g}'(y)]}} - \frac{\log p(y|\mathbf{x}_0)}{\sqrt{\text{Var}[\mathbf{N}_j \mathbf{g}(y)]}} \right]$$

$$+ 2 \text{Cov} \left[ \frac{\log p(y|\mathbf{x}_j)}{\sqrt{\text{Var}[\mathbf{N}_j \mathbf{g}(y)]}} - \frac{\log p'(y|\mathbf{x}_j)}{\sqrt{\text{Var}[\mathbf{L}'_j \mathbf{f}'(\mathbf{x})]}}, \frac{\log p'(y|\mathbf{x}_0)}{\sqrt{\text{Var}[\mathbf{L}'_j \mathbf{f}'(\mathbf{x})]}} - \frac{\log p(y|\mathbf{x}_0)}{\sqrt{\text{Var}[\mathbf{N}_j \mathbf{g}(y)]}} \right]$$

$$\leq \text{Var} \left[ \frac{\log p(y|\mathbf{x}_j)}{\sqrt{\text{Var}[\mathbf{N}_j \mathbf{g}(y)]}} - \frac{\log p'(y|\mathbf{x}_j)}{\sqrt{\text{Var}[\mathbf{N}'_j \mathbf{g}'(y)]}} \right]$$

$$+ \text{Var} \left[ \frac{\log p'(y|\mathbf{x}_0)}{\sqrt{\text{Var}[\mathbf{N}'_j \mathbf{g}'(y)]}} - \frac{\log p(y|\mathbf{x}_0)}{\sqrt{\text{Var}[\mathbf{N}_j \mathbf{g}(y)]}} \right]$$

$$+ 2 \sqrt{\text{Var} \left[ \frac{\log p(y|\mathbf{x}_j)}{\sqrt{\text{Var}[\mathbf{N}_j \mathbf{g}(y)]}} - \frac{\log p'(y|\mathbf{x}_j)}{\sqrt{\text{Var}[\mathbf{N}'_j \mathbf{g}'(y)]}} \right]} \sqrt{\text{Var} \left[ \frac{\log p'(y|\mathbf{x}_0)}{\sqrt{\text{Var}[\mathbf{N}'_j \mathbf{g}'(y)]}} - \frac{\log p(y|\mathbf{x}_0)}{\sqrt{\text{Var}[\mathbf{N}_j \mathbf{g}(y)]}} \right]}.$$

Since by assumption $d_{\text{LLV}}^\lambda(p, p') \leq \epsilon$, this implies that $t_2 \leq \epsilon$ and that for $\mathbf{x}_j \in \mathcal{X}_{\text{LLV}}$:

$$\sqrt{\text{Var} \left[ \frac{\log p(y|\mathbf{x}_j)}{\sqrt{\text{Var}[\mathbf{N}_j \mathbf{g}(y)]}} - \frac{\log p'(y|\mathbf{x}_j)}{\sqrt{\text{Var}[\mathbf{N}'_j \mathbf{g}'(y)]}} \right]} \leq \epsilon \tag{219}$$

and

$$\sqrt{\text{Var} \left[ \frac{\log p(y|\mathbf{x}_0)}{\sqrt{\text{Var}[\mathbf{N}_j \mathbf{g}(y)]}} - \frac{\log p'(y|\mathbf{x}_0)}{\sqrt{\text{Var}[\mathbf{N}'_j \mathbf{g}'(y)]}} \right]} \leq \epsilon. \tag{220}$$

Therefore, we get:

$$d_{\text{SVD}}(\mathbf{N}^\top \mathbf{g}(y), \mathbf{N}'^\top \mathbf{g}'(y)) \leq \sqrt{\sum_{l=1}^M M \text{Var}\left[\frac{\mathbf{N}_j \mathbf{g}(y)}{\sqrt{\text{Var}[\mathbf{N}_j \mathbf{g}(y)]}} - \frac{\mathbf{N}_j' \mathbf{g}'(y)}{\sqrt{\text{Var}[\mathbf{N}_j' \mathbf{g}'(y)]}}\right]} \quad (221)$$

$$\leq \sqrt{\sum_{l=1}^M M(\epsilon^2 + \epsilon^2 + 2\epsilon^2)} \quad (222)$$

$$= \sqrt{M^2 4\epsilon^2} = 2M\epsilon \quad (223)$$

showing the second part of the result. Taking the maximum between Equation (215) and Equation (223), we get the result of the claim. □

To see how this result connects to how far the embedding functions $\mathbf{f}(\mathbf{x}), \mathbf{f}'(\mathbf{x})$ are from being invertible linear transformations of each other, notice that if $\mathbf{L}^\top$ and $\mathbf{L}'^\top$ are both invertible, then if there exists an invertible linear transformation, $\mathbf{B}$, such that $\mathbf{L}^\top \mathbf{f}(\mathbf{x}) = \mathbf{B} \mathbf{L}'^\top \mathbf{f}'(\mathbf{x})$, then we also have an invertible linear transformation $\mathbf{A} = \mathbf{L}^{-\top} \mathbf{B} \mathbf{L}'^\top$ such that $\mathbf{f}(\mathbf{x}) = \mathbf{A} \mathbf{f}'(\mathbf{x})$.

Since $m_{\text{SVD}}(\mathbf{x}, y)$ is always non-negative, for this bound in Theorem 4.7 to be non-vacuous, we need

$$d_{\text{SVD}}(\mathbf{L}^\top \mathbf{f}(\mathbf{x}), \mathbf{L}'^\top \mathbf{f}'(\mathbf{x})) \leq 2M\epsilon < 1 \quad \implies \quad \epsilon < \frac{1}{2M} \,. \quad (224)$$

This means that for higher dimensional representations, we need the variance of the differences of log-likelihoods to be smaller, if we want a guarantee from this result.

Next, we prove that the dissimilarity between representations induced by Theorem 4.7 is invariant to substituting the members with models that are $\sim_L$-equivalent.

**Lemma E.8.** *For any two models* $(\mathbf{f}, \mathbf{g}), (\mathbf{f}', \mathbf{g}') \in \Theta$*, and for any other model* $(\mathbf{f}^*, \mathbf{g}^*) \in \Theta$ *such that* $(\mathbf{f}^*, \mathbf{g}^*) \sim_L (\mathbf{f}, \mathbf{g})$*, we have that:*

$$d_{\text{SVD}}(\mathbf{L}^\top \mathbf{f}(\mathbf{x}), \mathbf{L}'^\top \mathbf{f}'(\mathbf{x})) = d_{\text{SVD}}(\mathbf{L}^{*\top} \mathbf{f}^*(\mathbf{x}), \mathbf{L}'^\top \mathbf{f}'(\mathbf{x})) \,, \quad (225)$$

$$d_{\text{SVD}}(\mathbf{N}^\top \mathbf{g}(y), \mathbf{N}'^\top \mathbf{g}'(y)) = d_{\text{SVD}}(\mathbf{N}^{*\top} \mathbf{g}^*(y), \mathbf{N}'^\top \mathbf{g}'(y)) \,. \quad (226)$$

*Proof.* The proof follows using the linear equivalence relation Theorem 2.2. For any two models $(\mathbf{f}, \mathbf{g}) \sim (\mathbf{f}^*, \mathbf{g}^*)$ we have that:

$$\mathbf{g}_0(y)^\top \mathbf{f}(\mathbf{x}) = \mathbf{g}_0'(y)^\top \mathbf{f}'(\mathbf{x}) \,, \quad (227)$$

for all $\mathbf{x} \in \mathcal{X}$ and $y \in \mathcal{Y}$. By considering $M$ elements in $\mathcal{Y}$, we get

$$\mathbf{L}^\top \mathbf{f}(\mathbf{x}) = \mathbf{L}'^\top \mathbf{f}'(\mathbf{x}) \,, \quad (228)$$

where $\mathbf{L}, \mathbf{L}' \in \mathbb{R}^{M \times M}$ are the matrices constructed with columns the vectors $\mathbf{g}_0(y)$ and $\mathbf{g}_0'(y)$, respectively. Therefore, we get Equation (225):

$$d_{\text{SVD}}(\mathbf{L}^\top \mathbf{f}(\mathbf{x}), \mathbf{L}'^\top \mathbf{f}'(\mathbf{x})) = d_{\text{SVD}}(\mathbf{L}^{*\top} \mathbf{f}^*(\mathbf{x}), \mathbf{L}'^\top \mathbf{f}'(\mathbf{x})) \,. \quad (229)$$

To obtain Equation (226), notice that with similar steps we can write

$$\mathbf{N}^\top \mathbf{g}(y) = \mathbf{N}'^\top \mathbf{g}'(y) + \mathbf{b} \,. \quad (230)$$

where $\mathbf{b} \in \mathbb{R}$ is a displacement vector. We have

$$d_{\text{SVD}}(\mathbf{N}^\top \mathbf{g}(y), \mathbf{N}'^\top \mathbf{g}'(y)) = d_{\text{SVD}}(\mathbf{N}^{*\top} \mathbf{g}^*(y) + \mathbf{b}, \mathbf{N}'^\top \mathbf{g}'(y)) \quad (231)$$

and given that $d_{SVD}$ is invariant to translations, we arrive at the final result

$$d_{\text{SVD}}(\mathbf{N}^\top \mathbf{g}(y), \mathbf{N}'^\top \mathbf{g}'(y)) = d_{\text{SVD}}(\mathbf{N}^{*\top} \mathbf{g}^*(y), \mathbf{N}'^\top \mathbf{g}'(y)) \,. \quad (232)$$

This shows the claim. □

### E.7 Invariances of our Representational Distance and CCA

As noted in Appendix D.2, $m_{\mathrm{CCA}}(\mathbf{f}, \mathbf{f}')$ is invariant to any invertible linear transformation of $\mathbf{f}$ and $\mathbf{f}'$. In contrast, when considering our dissimilarity measure, $d_{\mathbf{f},\mathbf{g}}$, and the transformations to which it is invariant, it is important to note that it relies on both $d_{\mathrm{SVD}}(\mathbf{L}^\top \mathbf{f}(\mathbf{x}), \mathbf{L}'^\top \mathbf{f}'(\mathbf{x}))$ and $d_{\mathrm{SVD}}(\mathbf{N}^\top \mathbf{g}(y), \mathbf{N}'^\top \mathbf{g}'(y))$. In both these expressions, embeddings and unembeddings are coupled, since the matrices $\mathbf{L}, \mathbf{L}'$ (resp. $\mathbf{N}, \mathbf{N}'$) depend on the unembeddings $\mathbf{g}, \mathbf{g}'$ (resp. the embeddings $\mathbf{f}, \mathbf{f}'$). By contrast, $m_{\mathrm{CCA}}(\mathbf{f}, \mathbf{f}')$ can be computed independently of the unembeddings $\mathbf{g}, \mathbf{g}'$.

Consequently, the two similarity measures (dissimilarity in the case of $d_{\mathbf{f},\mathbf{g}}$) differ in their invariance properties, as demonstrated below. Consider two models $(\mathbf{f}, \mathbf{g}), (\mathbf{f}', \mathbf{g}') \in \Theta$ such that $\mathbf{f}(\mathbf{x}) = \mathbf{A}\mathbf{f}'(\mathbf{x})$ and $\mathbf{g}_0(y) = \mathbf{B}\mathbf{g}_0'(y)$, with $\mathbf{A}$ and $\mathbf{B}$ invertible matrices but such that $\mathbf{B} \neq \mathbf{A}^{-\top}$, i.e., $(\mathbf{f}, \mathbf{g})$ and $(\mathbf{f}', \mathbf{g}')$ are not in the same identifiability class. Since $\mathbf{A}$ and $\mathbf{B}$ are invertible matrices, we have $m_{\mathrm{CCA}}(\mathbf{f}, \mathbf{f}') = m_{\mathrm{CCA}}(\mathbf{g}, \mathbf{g}') = 1$. In contrast, we get:

$$d_{\mathbf{f},\mathbf{g}}((\mathbf{f}, \mathbf{g}), (\mathbf{f}', \mathbf{g}')) = \max \left\{ d_{\mathrm{SVD}}(\mathbf{L}^\top \mathbf{f}(\mathbf{x}), \mathbf{L}'^\top \mathbf{f}'(\mathbf{x})), d_{\mathrm{SVD}}(\mathbf{N}^\top \mathbf{g}(y), \mathbf{N}'^\top \mathbf{g}'(y)) \right\} \geq 0 \,,$$

(233)

where the equality holds if and only if there exist orthogonal matrices $\mathbf{O}, \mathbf{O}'$ and diagonal matrices $\mathbf{S}, \mathbf{S}', \mathbf{D}, \mathbf{D}' \in \mathbb{R}^{M \times M}$ with entries $S_{ii} \coloneqq (\psi_{\mathbf{x}}(y_i; p))^{-1}$, $D_{ii} \coloneqq (\psi_y(\mathbf{x}_j; p))^{-1}$, $S'_{ii} \coloneqq (\psi_{\mathbf{x}}(y_i; p'))^{-1}$, $D'_{ii} \coloneqq (\psi_y(\mathbf{x}_j; p'))^{-1}$, and displacement vectors $\mathbf{a}, \mathbf{b}$ such that:

$$\mathbf{L}^\top \mathbf{f}(\mathbf{x}) = \mathbf{S}^{-1} \mathbf{O} \mathbf{S}' \mathbf{L}'^\top \mathbf{f}'(\mathbf{x}) + \mathbf{a}, \quad \mathbf{N}^\top \mathbf{g}(y) = \mathbf{D}^{-1} \mathbf{O}' \mathbf{D}' \mathbf{N}'^\top \mathbf{g}'(y) + \mathbf{b} \,. \qquad (234)$$

Because the set of matrices described above is a subset of all linear invertible transformations, it is then possible to find cases where, in Equation (233), the equality does not hold for a careful choice of $(\mathbf{f}, \mathbf{g}), (\mathbf{f}', \mathbf{g}') \in \Theta$.

# F  Experimental Details

This section contains details of the implementation of all our experiments. A repository containing the code to reproduce our experiments is available on GitHub.[17]

## F.1  Constructed Models

In the following, we will detail how to construct the models which we will use to illustrate Theorem 3.1 and generate Table 1; and to illustrate the bound in Theorem 4.7 (see Appendix F.6).

We choose classification models with a representation space equal to $\mathbb{R}^2$. To construct the reference model $(\mathbf{f}, \mathbf{g}) \in \Theta$, we distribute its unembeddings uniformly on the unit circle, ensuring that the angle between any two adjacent unembeddings is equal. We then sample its embeddings such that each embedding corresponding to a given label lies closer—measured by angular distance—to its associated unembedding than to any other.

We construct another model $(\mathbf{f}'\mathbf{g}') \in \Theta$ by permuting the unembeddings and their associated embeddings. For both models, we then vary the norm of the unembeddings and measure $d_{\mathrm{KL}}$ and $d_{\mathrm{LLV}}^\lambda$ between models and the maximum of $d_{\mathrm{SVD}}$ between embeddings.

To illustrate how $d_{\mathrm{LLV}}^\lambda$ and the bound derived in Theorem 4.7 increase with increasing differences in representations, we compare a reference model with a perturbed version constructed by adding a small amount of Gaussian noise to the reference model's embeddings. By increasing the amount of noise, $d_{\mathrm{LLV}}^\lambda$ grows, as well as the maximum of $d_{\mathrm{SVD}}$.

## F.2  Models Trained on Synthetic Data

**Synthetic data generation**. We consider data with two-dimensional input and with $c$ classes, where each class consists of a slice of the circle together with the opposite slice. See Fig. 4 for an example with $c = 6$. We construct the data by drawing $20,000$ samples from a two-dimensional Gaussian ($\mu = \mathbf{0}, \sigma = 3$) and assigning labels based on angle to the first axis.

**Model training**. We trained classification models with $c \in \{4, 6, 10, 18\}$ classes, using a representation space equal to $\mathbb{R}^2$. Each model consists of three fully-connected layers, with layer sizes chosen from $\{16, 32, 64, 128, 256\}$. We use LeakyReLU activation functions. We train four types of models: one where the norms of both the embeddings and unembeddings are constrained to be equal to 20, one where the norm of the embeddings are equal to 20 and with unconstrained unembeddings, one where the norm of the embeddings is unconstrained and the norm of the unembeddings is 20, and one with no constraints on the norms. To obtain the results in Section 5.2, we only consider models with unconstrained norms. For each combination of the number of classes, the layer size, and whether the restriction is applied or not, we train with 20 random seeds. All models are trained with a batch size of 128 for $15,000$ steps using the ADAM optimizer [29].

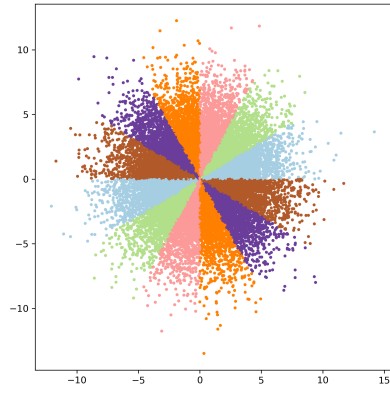

Figure 4: Illustration of training data for 6 classes. Each color represents a different class label.

## F.3  Models Trained on CIFAR-10

We trained classification models on CIFAR-10 [32], where the embedding network consisted of a ResNet18 [21][18] but choosing the representation space to be 2, 3 or 5-dimensional. For each dimension we trained 10 seeds. The unembedding network consisted of three fully connected layers of width 128, followed by an output layer of size 2, 3 or 5, thus giving us representations in the desired number of dimensions. The models were trained for $20,000$ steps with a batch size of 32

---

[17]github.com/bemigini/close-dist-rep-sim
[18]For this, we used code based on https://github.com/kuangliu/pytorch-cifar/blob/master/models/resnet.py

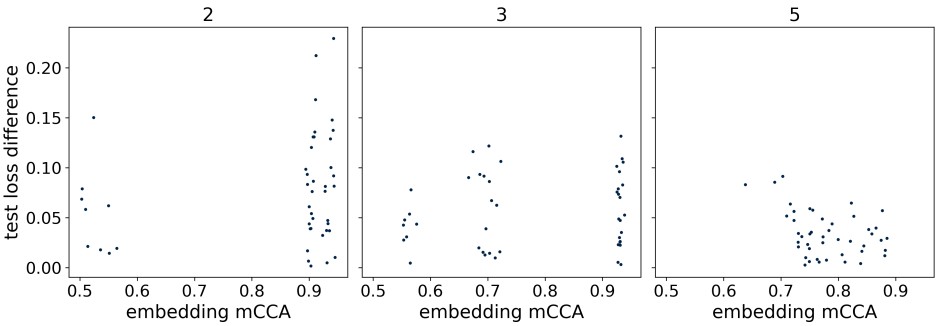

Figure 5: For models trained on CIFAR-10 with representational dimensions of 2, 3 and 5, difference in test loss vs $m_{\mathrm{CCA}}$ of the embeddings of the models. We see that there can both be a small difference in loss and a larger difference in representations or a larger difference in loss and a smaller difference in representations.

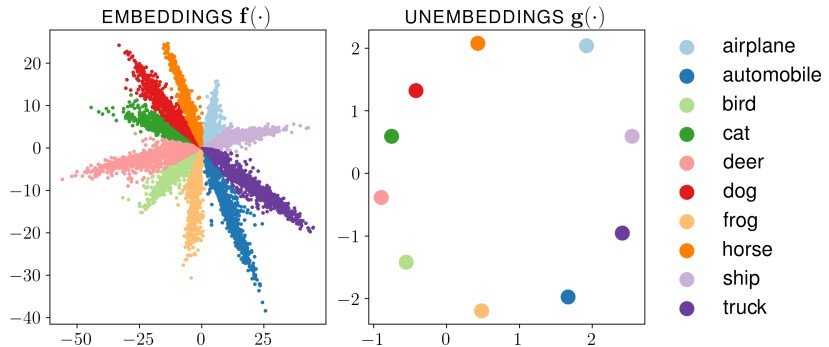

Figure 6: Illustration of representations of model trained on CIFAR-10, seed 0.

using the ADAM optimizer [29]. The ResNet model used ReLU activation functions, while the networks for the unembeddings used LeakyReLU activation functions.

## F.4 Loss Difference vs Embedding $m_{\mathrm{CCA}}$

To illustrate Corollary 3.2 and that a small difference in test loss does not guarantee similar representations for our models trained on CIFAR-10, we present Fig. 5. Here we compare difference in test loss with $m_{\mathrm{CCA}}$ of embeddings of models, where the dimension of the representations are 2, 3 and 5. We see that there can both be a small difference in loss and a larger difference in representations or a larger difference in loss and a smaller difference in representations.

## F.5 All Two-dimensional Representations of CIFAR-10 Models

We here present all the embedding and unembedding representations of our models trained on CIFAR-10 with 2-dimensional representations: seed 0 in Fig. 6, seed 1 in Fig. 7, seed 2 in Fig. 8, seed 3 in Fig. 9, seed 4 in Fig. 10, seed 5 in Fig. 11, seed 6 in Fig. 12, seed 7 in Fig. 13, seed 8 in Fig. 14, seed 9 in Fig. 15.

We see that some labels are neighbours for all ten models. For example, "automobile" and "truck" are neighbours for all seeds and "cat" and "dog" are neighbours for all seeds. However, other labels sometimes have varying neighbours. For example "airplane" and "bird" are neighbours for seeds 2, 3, 5, 6, 7, but not for the remaining seeds. In seeds 1, 4, 8 and 9, the "frog" label is put between them, and in seed 0 the labels are permuted even further. This behaviour might arise because inputs for some labels are so similar that it would aversely affect the performance of the model to place them far apart, while the embeddings of inputs for other labels can be placed in several equally good ways.

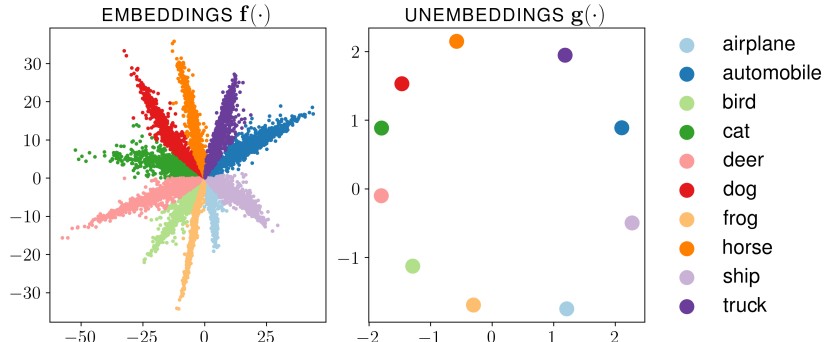

Figure 7: Illustration of representations of model trained on CIFAR-10, seed 1.

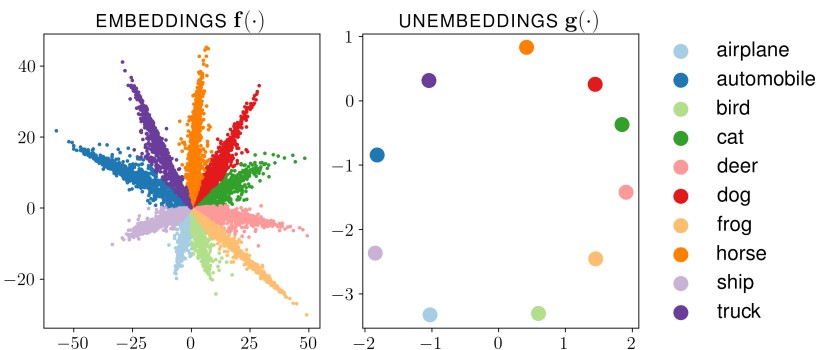

Figure 8: Illustration of representations of model trained on CIFAR-10, seed 2.

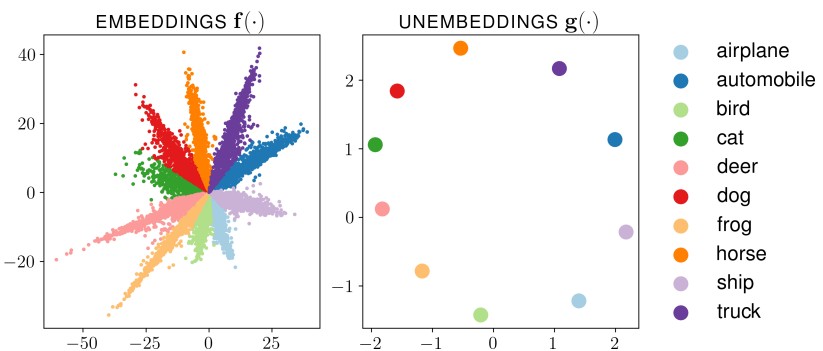

Figure 9: Illustration of representations of model trained on CIFAR-10, seed 3.

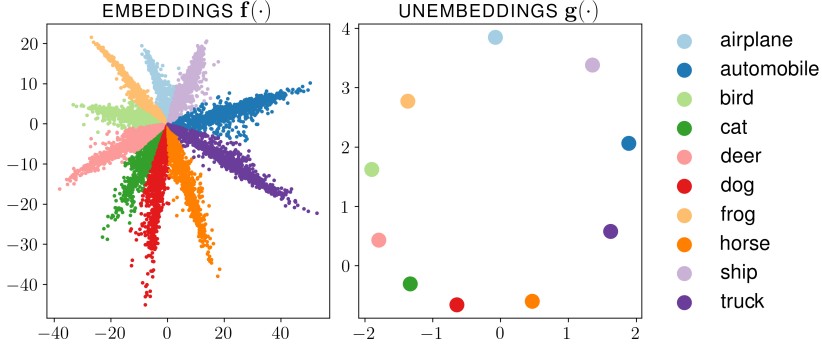

Figure 10: Illustration of representations of model trained on CIFAR-10, seed 4.

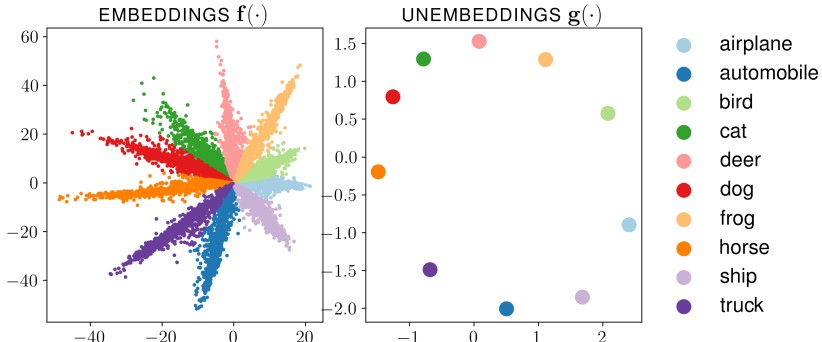

Figure 11: Illustration of representations of model trained on CIFAR-10, seed 5.

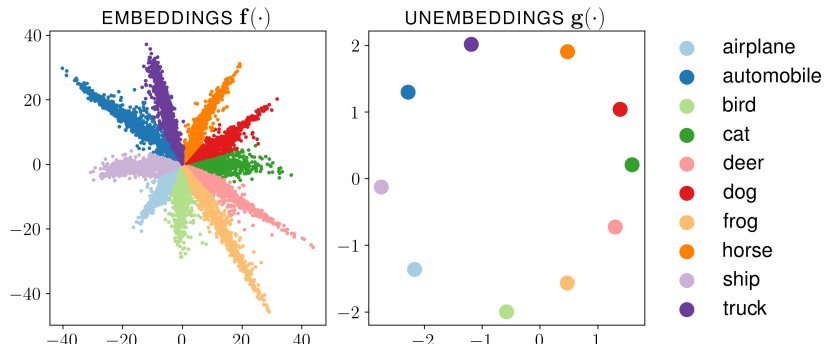

Figure 12: Illustration of representations of model trained on CIFAR-10, seed 6.

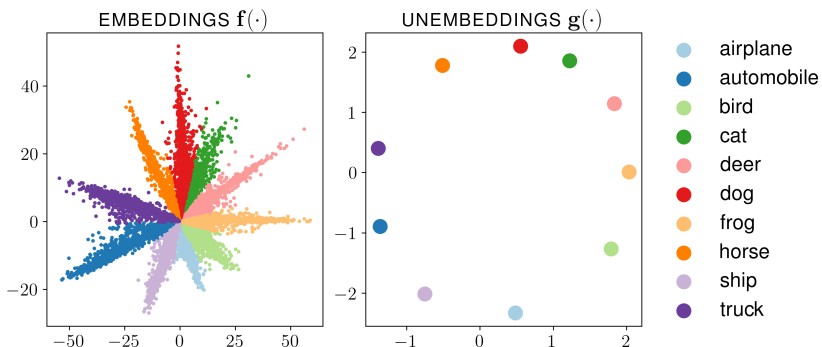

Figure 13: Illustration of representations of model trained on CIFAR-10, seed 7.

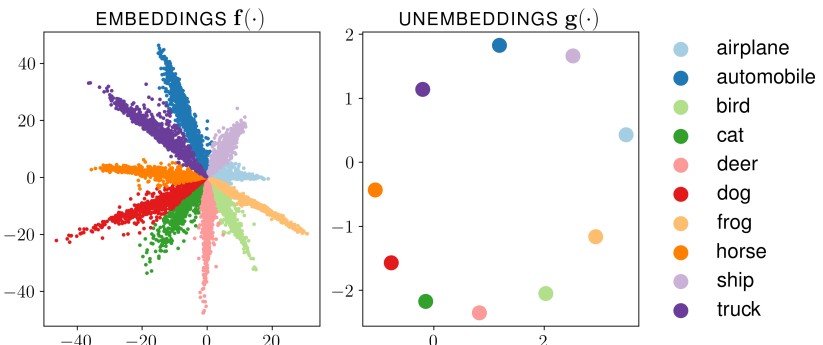

Figure 14: Illustration of representations of model trained on CIFAR-10, seed 8.

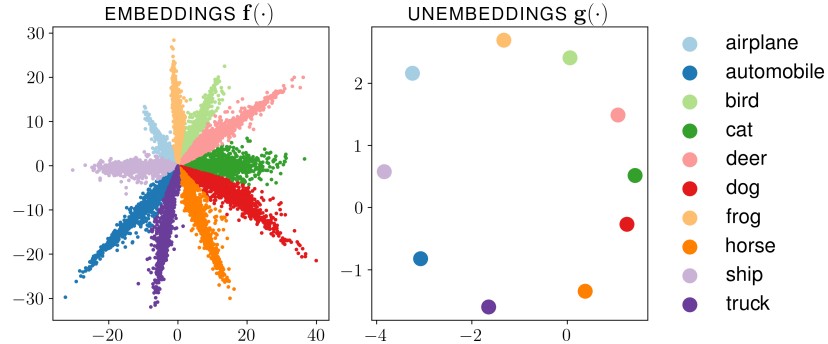

Figure 15: Illustration of representations of model trained on CIFAR-10, seed 9.

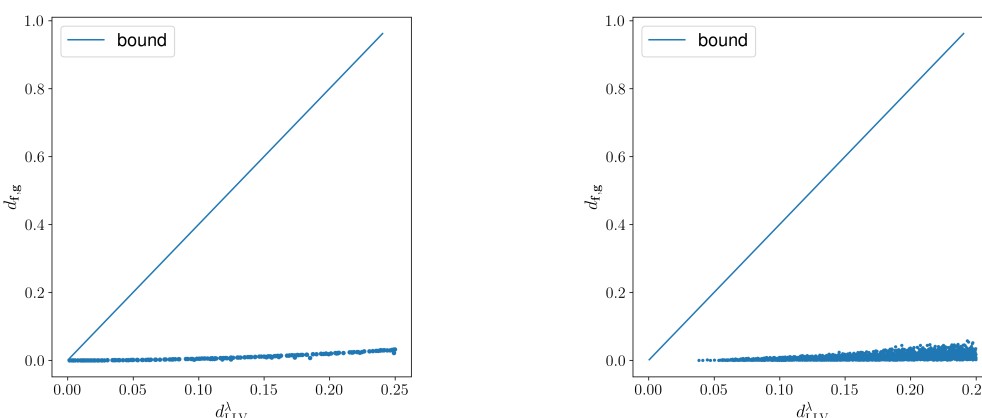

Figure 16: (Left) We evaluate $d_{\mathrm{LLV}}^{\lambda}$ and $d_{\mathbf{f},\mathbf{g}}$ for a reference model and a constructed one where we progressively increase the noise added to the embeddings. As more noise is injected, the two models display higher $d_{\mathrm{LLV}}^{\lambda}$ (up to 0.25), while the representation dissimilarity $d_{\mathbf{f},\mathbf{g}}$ does not exceed 0.1. (Right) We observe a similar trend for trained models, with a slight increase in $d_{\mathbf{f},\mathbf{g}}$ when $d_{\mathrm{LLV}}^{\lambda}$ approaches 0.25. We note that the bound given by Theorem 4.7 remains valid for both cases.

## F.6 Illustration of Bound on Constructed and Trained Models

**Experimental setup**. We compare both constructed models and models trained on synthetic data. For the constructed models, we compare a reference model to a perturbed version constructed by adding Gaussian noise to the reference model's embedding representations (as described in Appendix F.1). For models trained on synthetic data, the setup is the same as described in Appendix F.2

**Results**. In Fig. 16 (left), we observe that the maximum distance between representations gradually increases with the distance between probability distributions while staying below the bound of the Theorem 4.7. We find that not all the trained models have distance $d_{\mathrm{LLV}}^{\lambda}$ small enough for the bound to be non-vacuous. In this case, we report only those distances that are small enough for the bound to be non-vacuous between trained models in Fig. 16 (right) and see that the bound remains valid.

## F.7 Wider Models Have more Similar Distributions - Extra Plots

We here report the empirical trend that wider networks induce more similar distributions and more similar representations for models with 10 classes (Fig. 17 $d_{\mathrm{LLV}}$(left) and $\max d_{\mathrm{SVD}}$(right)) and 18 classes (Fig. 18 $d_{\mathrm{LLV}}$(left) and $\max d_{\mathrm{SVD}}$(right)). Note that with more classes, we have less models among the most narrow networks which achieve more than 90% accuracy. We have therefore left out results in the plots where we had fewer than 5 models for the comparison. For 10 classes, this means that starting at a width of 32, we have 9, 16, 19, and 19 models for the comparisons in the plot. For 18 classes, starting at a width of 64, we have 10, 12, and 19 models for the comparisons in the plot.

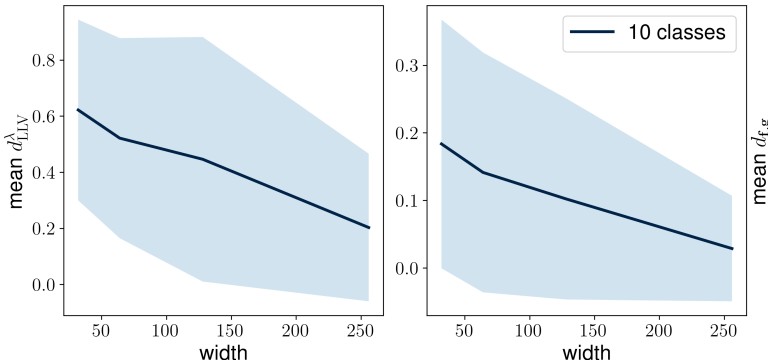

Figure 17: We display how mean $d_{\text{LLV}}$ and max $d_{\text{LLV}}$ varies with increasing the neural network width for models trained to classify among 10 classes. (Left) We observe decreasing mean $d_{\text{LLV}}$ as the network width grows. The shaded area represents the standard deviations evaluated from different random seeds retraining. (Right) A similar trend is also observed for max $d_{\text{LLV}}$ when increasing the network width.

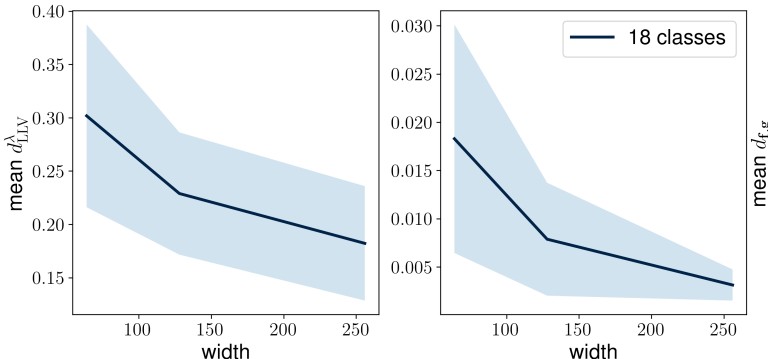

Figure 18: We display how mean $d_{\text{LLV}}$ and max $d_{\text{LLV}}$ varies with increasing the neural network width for models trained to classify among 18 classes. (Left) We observe decreasing mean $d_{\text{LLV}}$ as the network width grows. The shaded area represents the standard deviations evaluated from different random seeds retraining. (Right) A similar trend is also observed for max $d_{\text{LLV}}$ when increasing the network width.

# G Computing Resources

Each model was trained using a single NVIDIA RTX A5000. For each number of classes (4, 6, 10, 18), training 20 seeds on the synthetic data took about 34 hours, summing to a total of 136 hours to train the models on synthetic data. For the models on CIFAR-10, training 10 seeds took ∼27 hours.

The distances are calculated on a CPU-only machine: computing the distances for the models on synthetic data required less than 2 hours, whereas the evaluation on models in CIFAR-10 required around 4 hours. Evaluating the accuracy of models on synthetic data was also done on the CPU, taking less than 20 minutes in total.

# H Assets

**CIFAR-10**. We used the CIFAR-10 dataset as loaded with `torchvision` package[19]. The dataset from[32] and contains $50,000$ images for train and validation, and $10,000$ images for testing.

**Python Packages**. All experiments are conducted with `Python` 3.11 and used `pytorch` 2.5.1. Other packages are reported in the repository at `github.com/bemigini/close-dist-rep-sim`.

---

[19] `https://docs.pytorch.org/vision/main/generated/torchvision.datasets.CIFAR10.html`

