# OpenReview forum: "When Does Closeness in Distribution Imply Representational Similarity? An Identifiability Perspective"
_NeurIPS.cc/2025/Conference — NeurIPS 2025 poster_

### Official Review · Reviewer_r8r8 · 2025-07-01

**Clarity:** 3
**Significance:** 3
**Originality:** 4
**Rating:** 5
**Confidence:** 4

**Summary:**

This paper studies when two independently trained neural networks learn similar internal representations if their output distributions are close. It argues that a good similarity measure should be invariant to transformations that leave the model's output distribution unchanged. The paper argues that a small KL divergence between two models output distributions does not guarantee that their learned representations are similar, specifically that for any arbitrarily small $\varepsilon$, one can find two models with $d_{KL} \le \varepsilon$ whose embeddings are far from linearly equivalent. Even models with arbitrarily low classification loss can learn entirely dissimilar features. It introduces a new log-likelihood variance (LLV) distance between distributions and a PLS-SVD-based dissimilarity between representations. LLV is tailored to identifiability and is a proper metric. Using these, they prove that if two models distributions are close under the proposed LLV distance, then their representations are guaranteed to be similar. Concisely, closeness in the usual KL sense is insufficient to ensure representational similarity, but closeness in the new metric does imply it.

Experimentally, the authors train two ResNet18 models with a 2D bottleneck on CIFAR-10. Although both runs reach virtually identical test loss, the embeddings of the data are distinct, making CCA moderate and KL low while LLV distance remains high, validating their arguments. On a synthetic 2D classification toy dataset they show that as width grows, the mean and variance of LLV across seeds fall steadily, and the corresponding dSVD gap shrinks, suggesting higher-capacity models converge toward a common representation when their distributions align under LLV.

**Questions:**

- Have you looked at LLV for larger models / higher dimensional representations? Do the same conclusions hold? Curious about practicality of the metric.
- It seems like this might be intractable for something like language models, do you see anyway of accelerating / approximating this?
- I assume other f-divergences would have a similar issue as KL, was this looked at?
- Any thoughts on whether LLV could be used as an explicit penalty (loss) in something like distillation?

**Ethical Concerns:**

["NO or VERY MINOR ethics concerns only"]

**Final Justification:**

Authors addressed my questions appropriately, and I had little issue with the paper in the first place.  Showing that representational dissimilarity can be bounded using a distance between distributions is a nice contribution. Despite the loose bounds and computational complexity I think this is a promising direction of work. Maintaining my score of accept.

**Limitations:**

yes

**Quality:**

4

**Strengths And Weaknesses:**

# Strengths
- Theoretical: The paper proposes a rigorous perspective on representational similarity which is novel. Theorem 3.1 demonstrates a counter intuitive fact that even extremely small differences in output distributions can correspond to large differences in internal representations. This a relevant result w.r.t. efforts such as model distillation. Theorem 4.6 is likewise a meaningful contribution: a new distributional distance specifically designed to measure representation similarity under some assumptions.
- Experiments: While not large scale, they are well-designed to illustrate the argument. Section 5.1 shows a concrete example of two standard models with indistinguishable performance metrics but significantly different internal representations. This example makes the theoretical phenomenon tangible. The synthetic experiments in Section 5.2 show a clear trend that increasing model capacity yields closer distributions and more aligned representations.
- Clarity: The paper is generally well-written, albeit a bit dense.
- Novelty: LLV is a novel, theoretically validated distance that shows promise in experiments.
- Reproducibility: code included in supplementary materials, public release promised.

# Weaknesses:
- Assumptions: Theorem 3.1 requires certain conditions that somewhat limit its scope ($k \geq M+1$). It's not entirely clear how severe this limitation is, and perhaps is a reasonable assumption to make. Language models frequently have far more tokens than the embedding dimension, but it may be much closer (or reversed) with a vision model on ImageNet. Similarly, Theorem 4.6 comes with some assumptions, but again not clear on the severity.
- Bounds: The bound on dSVD ($2M\varepsilon$, will worsen with dimension) is not very tight, though the authors acknowledge this. The CIFAR experiments didn't enter the setting where the bound is non vacuous.
- Experimental: While the experiments do support the claims, they could be considered limited in scope. All experiments use relatively low-dimensional representations and simple datasets (e.g., CIFAR). Validation in higher dimensional cases or on a larger dataset (i.e., ImageNet) would be a valuable addition. Combined with the loose bounds, assessing the overall practical utility of LLV is a bit harder.

---

> ### Author Rebuttal · Authors · 2025-07-29
>
> We thank the reviewer for their encouraging feedback, and we are pleased that they found our results relevant and meaningful.
>
> > Clarification on the assumptions of our theorems:
>
> A central assumption of our work is the _diversity condition_ (Definition 2.1). As the reviewer rightfully noted, this condition is more likely to hold for language models, whereas for regular image classification models, it might not be satisfied (especially when the number of classes is smaller than the representation dimensionality). We will clarify this point in the paper. As we noted in Section 6, recent work [1] may serve as a starting point for exploring how to proceed when the diversity condition does not hold, potentially extending the results to typical image classifiers. However, we leave this direction for future work.
> We remark that Theorem 4.6 has an assumption slightly stronger than the usual diversity condition (all $\mathbf{L}, \mathbf{L}’, \mathbf{N}, \mathbf{N}’$ constructed from the input and label sets are invertible). However, we expect that if the diversity condition can be relaxed, then this slightly stronger assumption may also be relaxed while retaining a qualitatively similar result.
>
>
> > “The bound on dSVD [...] is not very tight”
>
> We agree with the reviewer’s observation and remark that this point is also acknowledged in Section 6 (“Limitations” paragraph). Nonetheless, we believe this result provides an important proof of principle: It demonstrates that **representational dissimilarity can be bounded using a distance between distributions**---a fact that is far from trivial, as Theorem 3.1 illustrates. Our experiments on synthetic data indicate that this bound could potentially be tightened; however, determining how to achieve this is left for future work.
>
> > “Validation in higher dimensional cases or on a larger dataset (i.e., ImageNet) would be a valuable addition”
>
> We thank the reviewer for the positive feedback on our experiments: “While not large scale, they are well-designed to illustrate the argument.” We agree that expanding to larger-scale experiments is an important step for future work, especially to bridge the gap to practical applications. Furthermore, we are particularly interested in experimenting with language models, as they appear to align most naturally with our assumptions.
>
> > “accelerating / approximating LLV computation”
>
> We hope that in future work we can reduce the number of labels required so that the metric scales with representational dimension rather than the number of labels.
> This would significantly reduce computation time for language models where the number of tokens far exceeds the representational dimension (e.g., GPT-2 has ~50k tokens vs. ~1000 dimensions).
> Approximating the computation of $d_\mathrm{LLV}$ is another promising direction for future work and could make it more suitable as a loss function.
>
>
> > Considering other f-divergences:
>
> One of the reasons why the KL divergence cannot be used to bound representational similarity stems from the fact that unlikely labels (with $p(y|\mathbf{x})$ very small) give a small contribution to the KL, which enables constructions as the one in Theorem 3.1. We expect that other divergences with similar behavior would also not allow us to derive bounds on representational similarity. Conversely, divergences that do not put particularly low weight on unlikely labels (like $d_\mathrm{LLV}$) might be better suited for deriving bounds on representational dissimilarity, as in our Theorem 4.6. We plan to test other divergences in future work.
>
>
> > “Any thoughts on whether LLV could be used as an explicit penalty (loss) in something like distillation?”
>
> We thank the reviewer for sharing this thought. We agree that this is an interesting direction to pursue, and we explicitly mentioned distillation in the introduction. However, we believe further research is needed before recommending log-likelihood variance distance as a loss function. We hope it will be possible to either find a simplified version of the distance or identify an upper bound or approximation that can serve as an effective and efficient distillation loss.
>
> [1] Marconato, E., Lachapelle, S., Weichwald, S., & Gresele, L. All or none: Identifiable linear properties of next-token predictors in language modeling. AISTATS 2025

---

> > ### Comment · Reviewer_r8r8 · 2025-08-02
> >
> > Thanks for the thorough response to my questions and the reminder of the importance of showing that representational dissimilarity can be bounded using a distance between distributions. I'm largely had no concerns to begin with, so I will maintain my positive score.

---

### Official Review · Reviewer_Hbxo · 2025-07-03

**Clarity:** 4
**Significance:** 3
**Originality:** 3
**Rating:** 5
**Confidence:** 4

**Summary:**

A typical identifiability treatment tells us that equal likelihoods imply similar representations, up to an identifiability class. What about unequal but similar likelihoods? The authors in this work study model and representational similarity in a large class of models, and show that similar likelihoods do not necessarily imply similar representations. The authors define the identifiability class of their model to establish their results, and use it to define a new distributional distance. They then use this distributional distance to bound representational similarity. Their results are supported by numerical results on the CIFAR dataset and synthetic experiments.

**Questions:**

See weaknesses above for some questions, with just one more here:
- Shape metrics from Williams et al (https://arxiv.org/abs/2110.14739) use **groups** to define invariance in shape classes, going beyond CCA -- can this be leveraged for your identifiability class, to be more specific than the CCA-based measures that consider any linear transformations?

**Ethical Concerns:**

["NO or VERY MINOR ethics concerns only"]

**Final Justification:**

Satisfied by comments, keeping acceptance score.

**Limitations:**

Limitations are discussed in the last section. Perhaps including an example of the failure modes mentioned could strengthen the text?

**Paper Formatting Concerns:**

Very well formatted! Only typo: L172, first sentence seems off.

**Quality:**

4

**Strengths And Weaknesses:**

Strengths:
- Well-motivated and well-framed question on identifiability and its relaxations.
- Impressive theoretical results and new definitions.
- A simple synthetic example corroborates a proof.
- Numerical experiment on CIFAR is non-trivial.

Weaknesses:
- Lacks comparison to the conclusions that could be drawn from existing approaches. For instance, CCA is the most raised alternative -- could the relationship to this model be further discussed? What would we miss by just using CCA?
- Some assumptions are not discussed to their fullest extent. For instance, does the pivot $x_0$ (especially its probability $p_{\mathcal{D}}(x_0)$) influence the distance or log-likelihood variance distance terms?

---

> ### Author Rebuttal · Authors · 2025-07-29
>
> We thank the reviewer for their encouraging feedback and for acknowledging the strength of our theoretical results.
>
> > “What would we miss by just using CCA?"
>
> This is an important technical point. To summarize:
>
> (1) Similarity measures based on CCA take only pairs of embeddings, or unembeddings, as input, whereas our **$d_{\mathrm{SVD}}\left(\mathbf{L}^{\top} \mathbf{f}(\mathbf{x}), \mathbf{L}^{\prime \top} \mathbf{f}^{\prime}(\mathbf{x})\right)$ takes both the embeddings and the unembeddings of both models as inputs**, since the matrices $\mathbf{L}$ and $\mathbf{L}^{\prime}$ depend on the unembeddings $\mathbf{g}$. Note in fact that the distribution of the model (Equation (1) in the paper) depends on both the embeddings and the unembeddings.
>
> (2) CCA-measures have thus different invariances and can lead to different conclusions than our dissimilarity measure $d_{\mathrm{SVD}}\left(\mathbf{L}^{\top} \mathbf{f}(\mathbf{x}), \mathbf{L}^{\prime \top} \mathbf{f}^{\prime}(\mathbf{x})\right)$. For example, mCCA between model embeddings is invariant to any linear transformation of the embeddings.
> In principle, one could find two models giving rise to different distributions where the embeddings of a model are a linear transformation of the embeddings of a second model, whereas their unembeddings are not equal up to the same linear transformation. These two models would thus have maximum mCCA (equal to 1) between embeddings, but they would have a positive $d_{\mathrm{SVD}}\left(\mathbf{L}^{\top} \mathbf{f}(\mathbf{x}), \mathbf{L}^{\prime \top} \mathbf{f}^{\prime}(\mathbf{x})\right)$ (not the minimum value of zero).
>
> Thanks for prompting this explanation: We will clarify this in the revised version of the paper, also based on a discussion of (2) we currently have in Appendix E.8.
>
> > “...does the pivot… influence the distance or log-likelihood variance distance terms?”
>
> We thank the reviewer for pointing out that this requires clarification. When models have different distributions, the choice of pivot (as well as the input and label sets) can indeed affect the value of the distance $d_\mathrm{LLV}$. As long as the diversity condition is satisfied, any choice of pivot is valid. If, on the other hand, the distributions of two models are equal, then the distance $d_\mathrm{LLV}$ will be zero no matter the choice of pivot and sets. We will make this clearer in the paper. In the experiments, we draw a sample of possible inputs with pivot sets and choose the set that gives the smallest value for the term $t_2$ in Definition 4.4 (we specified this in Appendix E.3: Implementation of the Log-Likelihood Variance Distance). We have not yet analyzed the effect of the pivot point probability on the value of the $t_2$ term in $d_\mathrm{LLV}$. It might be interesting to explore this in future work.
>
> > "Shape metrics from Williams et al…”
>
> We thank the reviewer for this interesting reference and will include it as related work. We will also consider whether its ideas can inform the design of other similarity measures with the desired invariance properties.

---

> > ### Comment · Reviewer_Hbxo · 2025-08-05
> >
> > Thank you to the authors for their response and clarifications. They answered my questions! I support adding these clarifications on CCA and pivot choice in the revised manuscript. I maintain my acceptance score.

---

### Official Review · Reviewer_66wY · 2025-07-03

**Clarity:** 3
**Significance:** 3
**Originality:** 3
**Rating:** 5
**Confidence:** 3

**Summary:**

The paper looks at co-embedding-based classification models, where inputs and outputs are mapped to a shared space and then inner products are used in a softmax to obtain conditional distributions.  The central question is whether, given access to all the conditional distributions for two models trained on the same data, one can claim anything about the similarity of the representations of the two models in the shared space (pre-softmax).

As a first guess, one might try using one of the popular statistical distances between corresponding conditional distributions, e.g. KL divergence, to assess classification similarity and infer that the models’ representations are closely related if the distance vanishes.  The authors show that this is not the case in general -- there are ways to obtain arbitrarily close output distributions with representation spaces that are not relatable through linear transformations.

The paper then proposes a statistical distance -- which additionally satisfies the four properties of a metric -- that essentially inverts the softmax operation by operating on log probability ratios and can bound how similar the representation spaces are up to linear transformation.  Some experiments on CIFAR10 and synthetic examples are included to bolster the theory, with an interesting result suggesting larger intermediate dimensions increase the final representation space similarity.

**Questions:**

- It would be helpful to understand whether the use of 2D embedding spaces in the experiments was driven by practical considerations (e.g., visualization) or if similar behaviors are expected in higher-dimensional regimes as well.

- The CIFAR point clouds (Fig 3 and Supp) look like radial information of the image embeddings might not be playing a role (due to the high density near the origin).  A quick look at the code suggests the norm might be fixed for these experiments.  While fixing the norm for the label embeddings serves a purpose for the different synthetic experiments, applying norm constraints to the *image* embeddings for CIFAR10 seems less directly motivated by the theory.  This may weaken the empirical alignment with the theoretical results and limit the already constrained experimental scope.  Could the authors clarify whether this design choice is discussed in the text?  And more generally, can they speak to the rationale for norm-fixing in this setting?

**Ethical Concerns:**

["NO or VERY MINOR ethics concerns only"]

**Final Justification:**

The authors have addressed the majority of my concerns.  While the experiments' low dimensionality remains a weakness, it is outweighed by the rest of the paper.  I raise my score to a solid accept.

**Limitations:**

Yes

**Quality:**

3

**Strengths And Weaknesses:**

## Strengths

- The paper is clearly written and well-structured.  The flow from proposing KL divergence, to providing an intuitive counterexample, to proposing the LLV distance is effective.  Prior work and limitations are addressed with transparency.

- Linking the set of classification probability distributions to the underlying representation space, for a format that is broadly used in machine learning (the co-embedding space matching x to y), offers a foothold into novel assessments of representation similarity.  The identifiability angle is powerful.  The metric properties of d_LLV, though not utilized in this work, are an added bonus.

## Weaknesses

The experimental scope is limited, especially in that all embedding spaces are restricted to 2D, raising concerns about whether the observed behaviors extend to higher-dimensional settings typical of practical applications.

---

> ### Author Rebuttal · Authors · 2025-07-29
>
> We thank the reviewer for their constructive feedback and for recognizing the potential of our analysis.
>
> > “all embedding spaces are restricted to 2D”
>
> Regarding the CIFAR-10 experiments in Section 5.1: to guarantee the diversity condition, the representation dimension needs to be smaller than the number of labels, and we specifically chose a dimension of 2 for ease of visualization. This makes it easier to compare our findings in Figure 3 (Left) with Figure 2, which illustrates the construction used in Theorem 3.1.
> To address the reviewer’s concern, we will add results from models we have trained on CIFAR-10 with 3- and 5-dimensional representation spaces. For 3- and 5-dimensional representations, we cannot provide a visualization like in Figure 3 (Left). Instead, we will include scatterplots in the appendix showing the difference in test loss versus the mean canonical correlation (mCCA) of embeddings from models trained on CIFAR-10. In these scatterplots, we see that it is possible to get both smaller loss differences and less similar representations, and models with higher loss difference and more similar representations. For 2 and 3 dimensions, we see that the minimum representational similarity is lower than for 5 dimensions, and the maximum similarity achieved is slightly higher.
> All in all, these scatter plots suggest that a small loss difference does not imply that the underlying representations are similar, in agreement with our Corollary 3.2.
>
> > “The experimental scope is limited”
>
> The primary goal of our work is to deepen the mathematical understanding of representational similarity through the lens of identifiability. We appreciate reviewer `r8r8`’s comment that “while not large scale, [our experiments] are well-designed to illustrate [our] argument,” and we believe that the additional results described above and prompted by this review further strengthen our experimental validation. We agree that an empirical study with larger-scale experiments and models is an important direction for future research.
>
>
> > “CIFAR point clouds… the [embedding] norms might be fixed for these experiments”
>
> We thank the reviewer for pointing out that this aspect of the code was unclear. **We did not constrain the embedding or unembedding norms of the models presented in the main paper**:
> The embeddings and unembeddings are plotted in Figure 5 to Figure 14 in Appendix F, and it can be seen that their norms are not fixed. However, we have some additional experiments in Appendix F where either embedding or unembedding norms, or both, were fixed. These ablations were not included in the main paper, but only in Appendix F.5, "Illustration of Bound on Constructed and Trained Models". We will clarify this point once we publish the code.

---

> ### Comment · Reviewer_66wY · 2025-08-01
>
> I might be misunderstanding: it seems that the authors are suggesting the visualizations in Figs 3 and 5-14 (where the image embeddings clearly have different radii) are evidence that the vector norms were not fixed when calculating likelihoods.  The likelihood calculation in the code is not simply the inner product between the embeddings and the coembeddings: it has optional normalization in `model_class.py` that was turned on for some of the experiments specified in `ARTICLE_model_variations_config_resnetcifar10_128_fd2.json`.
>
> To the contrary, the density of points near the origin and the crisp azimuthal decision boundaries suggests the opposite, that cosine similarity was effectively used instead of the inner product described in the paper.

---

> > ### Author Response · Authors · 2025-08-02
> >
> > We did not fix the norms of the CIFAR-10 models when calculating likelihoods. We clarify this by outlining the relevant parts of our code.
> >
> > The code for calculating the likelihood can be found in `src/models/model_class.py`. It uses the dot function:
> >
> > ```
> > def dot(self, features, target):
> > ```
> >
> > Which fixes the norm if the `fix_length_gs` and `fix_length_fs` options are set to a number larger than zero, as can be seen in lines 45 and 50, e.g.
> > ```
> > if self.fix_length_gs > 0:
> > ```
> > This is the "optional normalization" mentioned by reviewer 66wY, but this option was not used to generate the plots (fig 3 and 5-14) in our paper. The code for generating the plots is in `plots/cifar10_embeddings_can_be_permuted.py`.
> > Lines 106–107 set the relevant config options:
> > ```
> > current_fix_f_option = model_var_config.fix_length_gs[0]
> > current_fix_g_option = model_var_config.fix_length_fs[0]
> > ```
> > In the config, the first element of each list is 0, corresponding to the setting where the norms of f and g are not fixed.
> >
> >
> > Note also, that the normalization is included in the `get_g_reps` and `get_f_reps` functions in `src/models/model_class.py` (lines 118 and 130),
> > and these functions are used in the plotting code (`plots/cifar10_embeddings_can_be_permuted.py`, lines 128 and 137).
> > Therefore, if the norms were fixed for these models, it would be reflected in the plots. That is, the embeddings and unembeddings would lie exactly on a circle.

---

> > > ### Comment · Reviewer_66wY · 2025-08-03
> > >
> > > Ok, my mental model for what the embeddings "should" look like was flawed.  Thank you for the detailed clarification.

---

### Official Review · Reviewer_TFNc · 2025-07-03

**Clarity:** 3
**Significance:** 2
**Originality:** 2
**Rating:** 4
**Confidence:** 4

**Summary:**

In this work, the authors aim to analyze whether similarity between two models at the level of behavior (predictions made by a model) implies similarity at the level of representations. To this end, the authors contextualize results from identifiability theory (which show equal likelihood implies representational similarity up to invertible transformations under certain diversity conditions) and argue that developing a meaningful analysis requires first formalizing what similarity metrics even make sense. The authors thus propose a new "distributional distance" that bounds representational dissimilarity, and enables statements on similarity of representations based on closeness of predictions.

**Questions:**

See weaknesses.

**Ethical Concerns:**

["NO or VERY MINOR ethics concerns only"]

**Final Justification:**

The comments were very helpful in helping clarify my questions! I'll update my score accordingly.

**Limitations:**

Yes

**Quality:**

3

**Strengths And Weaknesses:**

**Strengths.** The question of how and when similarity in the predictions space implies similarity in the representation space is extremely useful: if true, training different models to merely learn the distribution will yield the same representations and be of similar utility. The paper is very well-written, though partially lacking in experiments that could have been helpful for emphasizing the theory.

**Weaknesses.** The weaknesses below are based on my interpretation of results. Please correct me if I'm wrong in how I understood the results and I'll happily change my scores in accordance with that.

- Construction-centric nature of Theorem 3.1 and related arguments: The primary argument in the formal analysis is based on construction of a model class that is equivalent in predictions, but not in representations. While fair, the argument ignores learning dynamics: if a model does not ever learn solutions beyond a narrow subset of the elements of this prediction-equivalence class, should we worry about the *possibility* of representational dissimilarity?

- Permutation based representation dissimilarity in Theorem 3.1: My understanding of the arguments up to theorem 3.1 was that authors deem models with representations equivalent up to linear transformations as representationally similar. Wouldn't permutations fall under the scope of this definition? If so, I don't follow why the intuitive figure (Fig 2) relies on permutation of representations to produce representationally dissimilar models. More broadly, isn't a permutation transformation too trivial to consider models representationally dissimilar? I would in fact argue that representational similarity analyses should by default ignore all trivial symmetries of neural network architectures (e.g., permutations of neurons yielding a behaviorally equivalent model).

- Empirical characterization: I regard the paper mostly as a theory work and hence this is a minor weakness. Specifically, I found the empirical analysis somewhat underwhelming. For example, the authors argue based upon their defined measure that representational similarity increases with width, an arguably known result from past work on CCA / CKA based similarity analyses [1]. However, offering a tool should ideally be grounded in some interesting conclusion or application.

[1] https://arxiv.org/abs/1905.00414

---

> ### Author Rebuttal · Authors · 2025-07-29
>
> We thank the reviewer for their helpful questions and for emphasizing the significance of our research question.
>
> > “Construction-centric nature of Theorem 3.1…”
>
> It is true that just because these constructions exist doesn't mean a model will learn them during training. We will make this clearer in the article.    However, identifying these examples is essential for mathematical understanding of the problem: identifiability research has similarly made progress by starting from constructed counterexamples (e.g. [1]) to understand non-identifiability issues.  Moreover, we find it interesting that a mechanism resembling our construction actually emerged in the CIFAR-10 experiments (Section 5.1). Understanding when and why such constructions are learned in practice would be an interesting direction for future research.
>
> > “Permutation based representation dissimilarity…”
>
> We appreciate the comment and suspect the confusion may be due to unclear phrasing on our part.
> The permutation in 3.1 should be understood as a permutation, $\pi$, of the _label indices_, $\mathbf{g}’(y_i) = \mathbf{g}(y_{\pi (i)})$ with a corresponding shift of the embedding clusters; and _not as a permutation of the representation dimensions (i.e., the axes in the figure)_, $\mathbf{f}’(\mathbf{x}) = \mathbf{P}\mathbf{f}(\mathbf{x})$ for some permutation matrix $\mathbf{P}$, which would indeed be a linear transformation, as the reviewer points out. Unlike the latter, the permutation of label indices, and corresponding shift of embedding clusters, is not a linear transformation of the representations: In Table 1, we show that when constructing models in this way, the mean canonical correlation is close to 0, which means their representations are very far from being linear transformations of each other. The permutation of label indices also changes the model’s output distribution, since it alters the rank—by predicted probability—of every label except the top one for each input $\mathbf{x}$.  Conversely, permutations of embedding neurons  $\mathbf{f}’(\mathbf{x}) = \mathbf{P}\mathbf{f}(\mathbf{x})$ (and accordingly of unembedding ones) in a way that lead to the same probabilities will not alter the model’s output distribution and will thus yield equivalent model instances according to our identifiability perspective (models that have equal distributions are deemed equivalent). We will include the clarification above in the revised version of the paper.
>
>
> > “The authors argue based upon their defined measure that representational similarity increases with width, an arguably known result from past work”
>
> We believe this comment may reflect a misunderstanding of the primary novelty of the experimental findings in Section 5.2. That higher-capacity models tend to produce more similar representations is indeed a known result, which we acknowledge on page 2 (see Footnote 2). The experiments in Section 5.2 examine this phenomenon through the lens of our novel theoretical results. Specifically, they show that wider neural networks yield lower mean and variance of our distributional distance $d_\mathrm{LLV}$ (second plot from the right in Figure 3), and that this correlates with lower values of our representational dissimilarity measure $d_\mathrm{SVD}$ (rightmost plot in Figure 3)—both in the extreme regime where the bound from Theorem 4.6 is non-vacuous (thereby illustrating our theory) and, intriguingly, even beyond that regime, suggesting that a tighter bound may be attainable. These results are inherently novel, as they rely on our newly introduced dissimilarity measure $d_\mathrm{SVD}$ and distributional distance $d_\mathrm{LLV}$. Importantly, prior experimental studies have used different similarity measures—most notably those based on CCA. However, similarity under CCA does not imply similarity under $d_\mathrm{SVD}$, since CCA is invariant to arbitrary linear transformations, whereas $d_\mathrm{SVD}$ is only invariant to a restricted class of such transformations (see Appendix E.8 and our response to reviewer `Hbxo`).
>
> > “Empirical characterization”
>
> Our primary goal is to advance the mathematical understanding of representational similarity through the lens of identifiability. We appreciate that the reviewer considers lack of largeer scale experiments a “minor weakness,” and we agree that a more comprehensive empirical investigation is a valuable direction for future work. At the same time, we are encouraged by reviewer `r8r8`’s assessment that, “while not large scale, [our experiments] are well-designed to illustrate [our] argument.” In addition, we have conducted further experiments to strengthen the empirical validation of our approach; please see our response to reviewer `66wY` for details.
>
> > “offering a tool should ideally be grounded in [...] application”
>
> While our current work is primarily theoretical and only represents an initial step towards applications, we believe it is a significant and original one; in particular, it relaxes a strongly unrealistic assumption underlying many prior identifiability analyses—namely, the assumption of equal model distributions. We see this as an important step toward developing methods with empirical relevance and applicability. Encouragingly, several reviewers offered concrete suggestions for how this line of work could evolve toward practical applications (e.g., for distillation, as suggested by reviewer `r8r8`). We take this as a sign that our theoretical contribution opens up a promising avenue for future development with real-world impact.
>
> [1] Hyvärinen, A., & Pajunen, P. (1999). Nonlinear independent component analysis: Existence and uniqueness results. Neural networks, 12(3), 429-439.

---

> > ### Comment · Reviewer_TFNc · 2025-08-01
> >
> > The comments were very helpful in helping clarify my questions! I'll update my score accordingly.

---

### Decision · Program_Chairs · 2025-09-17

**Decision:**

Accept (poster)

**Comment:**

This work considers the problem of comparing the representations of models trained to perform the same task. The authors show that small statistical divergences between model distributions (specifically, the conditional distributions of the outputs given the inputs) does not imply that their representations are similar. The authors then propose a distance (a proper mathematical metric) on the model distributions that has the property that bounds how similar the model representations are. They illustrate their theory empirically and show that larger intermediate dimensions lead to more similar representations. Reviewers stated that the problem is interesting, the theoretical results were impressive, and the paper was well written; though they felt that the empirical tests could be stronger. All of the reviewers ultimately gave the paper positive reviews, so I recommend acceptance.